CERN-TH-2024-134

# Matrix models for extremal and integrated correlators of higher rank

**Alba Grassi and Cristoforo Iossa**

*Section de Mathématiques, Université de Genève, 1211 Genève 4, Switzerland*

*Theoretical Physics Department, CERN, 1211 Geneva 23, Switzerland*

*E-mail:* Alba.Grassi@unige.ch, Cristoforo.Iossa@unige.ch

ABSTRACT: We study extremal and integrated correlators of half-BPS operators in four-dimensional $\mathcal{N} = 2$ SQCD and $\mathcal{N} = 4$ SYM with $SU(3)$ gauge group. We focus on the large R-charge sector where the number of operators insertions becomes very large. In this regime, we demonstrate that the correlators are described by a combination of Wishart and Jacobi matrix models, coupled in a non-trivial way. The size of the matrices in each model corresponds to the maximal number of insertions for each of the two single trace generators. This dual matrix model representation allows us to extract the behavior of the correlators at weak and strong coupling in a 't Hooft-like double scaling limit, including nonperturbative corrections. Although this work focuses on $SU(3)$, we expect that our techniques can be extended to $SU(N)$ for $N > 3$ as well.

# 1 Introduction

Strongly coupled systems present significant challenges due to their complex behavior. However, certain aspects of these theories become more manageable when focusing on sectors where specific quantum numbers are taken to be large. Well known examples include the large $N$ limit, where $N$ denotes the rank of the gauge group [1, 2], the large spin limit [3–6], and the large R-charge limit [7]. More recently, a new approach based on effective field theory (EFT) has gained importance for studying sectors of a theory where a given global charge is considered to be very large [8–10], see [11] for a review and a more exhaustive list of references.

Natural settings to rigorously understand and test these ideas are provided by $\mathcal{N} = 2$ SQCD and $\mathcal{N} = 4$ SYM in four dimensions, where localization techniques [12] offer a concrete handle on many observables. Such techniques are particularly useful for computing extremal [13] and integrated [14] correlators of 1/2 BPS operators. These operators are charged under a global $R$-symmetry, with the charge proportional to the number of operators insertions. Therefore, studying the large $R$-charge regime corresponds to analyzing correlation functions with a very large number of operators insertions.

For $SU(2)$, $\mathcal{N} = 2$ SQCD, an EFT approach was proposed in [15, 16], predicting that the behavior of extremal correlators of Coulomb branch operators, in the limit of large $R$-charge, essentially follows the form of a Gamma function with argument $R$. These predictions were subsequently demonstrated in [17]. The key insight of this approach is that we can write extremal correlators as Wishart matrix models, where the size of the matrices corresponds to the number of operators insertions, i.e. the $R$-charge[1]. Such matrix model representation provides a systematic method for computing the large $R$ expansion and gives an analytic prediction for the non-perturbative corrections to the EFT. In addition, it explains why in this limit, extremal correlators obey a non-trivial double scaling limit, as conjectured empirically in [20], see also [21, 22].

The objective of this work is to extend [17] to higher-rank theories, where EFT predictions are currently unavailable. We focus on the $SU(3)$ gauge group, though we expect that many of the structures can be generalized to $SU(N)$ for $N > 3$ as well. We will first study extremal correlators in $\mathcal{N} = 2$ SQCD and $\mathcal{N} = 4$ SYM and, then we apply our results to integrated correlators in $\mathcal{N} = 4$ SYM. Let us also mention that, for correlation functions of the so-called "maximal-trace operators" in $SU(N)$, $\mathcal{N} = 2$ SQCD, a matrix model approach was proposed in [23]. Although this approach works well for integrated correlators [24], it does not seem to accurately reproduce flat space extremal correlators in SQCD. The discrepancy begins at six loops and is due to the complex mixing structure of SQCD, which we discuss in detail in section 3.

---

[1]It is important to emphasize that our matrix models are fundamentally different from those typically encountered in the context of supersymmetric localization, where the size of the matrices corresponds to the rank of the gauge group and the models are hermitians. For a recent discussion on these other "Pestun"-type models, see for instance [18, 19].

This paper is structured as follows. In section 2 we review the definition and the computation of extremal correlators of Coulomb branch operators in $\mathbb{R}^4$ following [25]. This set of operators, endowed with the OPE, forms a freely generated ring. For $SU(3)$ gauge theories this ring is generated by

$$\phi_2 = \operatorname{Tr}\varphi^2, \quad \phi_3 = \operatorname{Tr}\varphi^3 , \tag{1.1}$$

where $\varphi$ is the complex scalar in the $\mathcal{N} = 2$ vector multiplet. The computation of extremal correlators in $\mathbb{R}^4$ can be then reduced to the computation of a minimal set of correlation functions, which we denote by

$$G_n^m(\tau,\overline{\tau}), \quad m, n \geq 0 , \tag{1.2}$$

where $\tau = \frac{4\pi i}{g_{\mathrm{YM}}^2} + \frac{\theta}{2\pi}$ is the marginal coupling. Roughly speaking, the index $n$ controls the maximal number of insertions of $\phi_2$, and $m$ controls the maximal number of insertions of $\phi_3$ into the correlation functions. See (2.39) for the definition of $G_n^m(\tau,\overline{\tau})$ in $\mathcal{N} = 4$ SYM, and (2.72) for the analogous definition in $\mathcal{N} = 2$ SQCD. For fixed values of $m$ and $n$, an algorithm was proposed in [25] to compute (1.2) starting from correlation functions on $S^4$, which can be computed using symmetric localization. However, extracting the large $m$ and/or $n$ behaviour from this prescription is far from trivial and, to explore this asymptotic regime, we need to rewrite the algorithm of [25] in the language of matrix models.

To achieve this, a key step is to understand the structure of the operator mixing, which we analyse in section 3. For $\mathcal{N} = 2$ SQCD, we will thoroughly examine such structure in the double scaling limit where $m$ and/or $n$ are taken to be large and, simultaneously, we also take $\mathrm{Im}\tau$ to be large in such a way that

$$\lambda = \frac{m}{2\pi\mathrm{Im}\tau}, \quad \kappa = \frac{n}{2\pi\mathrm{Im}\tau}, \tag{1.3}$$

are kept fixed. We often refer to this as a 't Hooft limit, following the matrix model terminology. However, in this context, we are specifically studying $SU(3)$ gauge theories, and the large parameter is not the rank of the gauge group but the R-charge $R = 2(3m+2n)$. In this limit we find that the mixing structure in $\mathcal{N} = 2$ SQCD behaves as the one of $\mathcal{N} = 4$ SYM and hence greatly simplifies. We expect a similar simplification to occur also if $\mathrm{Im}\tau$ is kept fixed, but we leave this for further investigation.

After making the mixing structure explicit, in section 4 and section 5 we show that the correlation functions (1.2) are computed by a combination of two matrix models: a Wishart matrix model, which controls the index $n$, and a Jacobi matrix model, which controls the index $m$. More precisely, in section 4 we focus on $\mathcal{N} = 4$ SYM, and we find an exact expression for extremal correlators in term of Wishart and Jacobi models (4.14):

$$(G_n^m(\tau,\overline{\tau}))^{\mathcal{N}=4} = \frac{6^{-m-1}\sqrt{3}}{Z_{S^4}} \frac{Z_{\mathrm{J}}^{(\sigma_m)}\left(1 + \left\lfloor\frac{m}{2}\right\rfloor\right)}{Z_{\mathrm{J}}^{(\sigma_m)}\left(\left\lfloor\frac{m}{2}\right\rfloor\right)} \frac{Z^{(m)}(n+1)}{Z^{(m)}(n)} , \tag{1.4}$$

where

$$Z^{(m)}(n) = \frac{1}{n!} \int_{\mathbb{R}_+^n} \mathrm{d}^n y \prod_{i<j} (y_i - y_j)^2 \prod_{i=1}^{n} \mathrm{e}^{-2\pi \mathrm{Im}\tau y_i} y_i^{3m+3} \tag{1.5}$$

is a Wishart-Laguerre matrix model and

$$Z_{\mathrm{J}}^{(\sigma_m)}(k) = \frac{1}{k!} \int_{[0,1]^k} \mathrm{d}^k x \prod_{1 \le i < j \le k} (x_i - x_j)^2 \prod_{i=1}^{k} \sqrt{\frac{1}{x_i} - 1} \, x^{\sigma_m} \,, \quad \sigma_m = m \mod 2 \tag{1.6}$$

is a Jacobi matrix model, see Appendix A for more details on these models. Let us stress that (1.4) is exact and holds even at finite values of $m$, $n$ and $\mathrm{Im}\tau$.

In section 5, we demonstrate that, similarly to the rank 1 case [17], the extremal correlators of $\mathcal{N} = 2$ SQCD in the large charge regime are equivalent to the expectation values of the so-called one-loop partition function $Z_{\mathrm{G}}(x, y)$ (2.63) within the $\mathcal{N} = 4$ SYM matrix models (1.4). The presence of the one-loop partition function $Z_{\mathrm{G}}(x, y)$ couples the two matrix models (1.5) and (1.6) in a non-trivial way. These results enable us to study the large $m$ and/or $n$ regime of extremal correlators, as well as the associated non-perturbative effects. Interestingly, if we keep $n$ finite and take $m$ to be large, we find a description involving only a Jacobi matrix model, that is

$$\frac{(G_0^m(\tau, \overline{\tau}))^{\mathcal{N}=2}}{(G_0^m(\tau, \overline{\tau}))^{\mathcal{N}=4}} \simeq \frac{\langle \mathcal{Z}_G(x, 3\lambda) \rangle_J^{(\lfloor \frac{m}{2} \rfloor + 1)}}{\langle \mathcal{Z}_G(x, 3\lambda) \rangle_J^{(\lfloor \frac{m}{2} \rfloor)}} \,, \tag{1.7}$$

where $\langle \, \cdot \, \rangle_J^{(k)}$ denote the expectation value in the Jacobi matrix model, see equations (5.8) and (5.10). In this case the behavior closely resembles that of $SU(2)$ SQCD, even though the matrix model involved is a Jacobi model rather than a Wishart model. Therefore, it seems plausible that an EFT description similar to the one proposed in [15] exists in this special sector of the $SU(3)$ theory. However, if we take also $n$ to be large, new structures emerge and the description is richer, see subsection 5.2 and subsection 5.3. For example, in the 't Hooft limit (1.3) with $\beta = \frac{n}{m}$ fixed, we find that the leading non-perturbative effect, in the regime $\lambda, \kappa \gg 1$, is

$$\mathrm{e}^{-A_1(\beta)\sqrt{\lambda}}, \quad \text{with} \quad A_1(\beta) = 2\pi \frac{\sqrt{6}}{\sqrt{2\beta + 2\sqrt{\beta(\beta + 3)} + 3}}, \quad \beta = \frac{n}{m}. \tag{1.8}$$

In particular in the limit $\beta \to \infty$ we have

$$A_1(\beta)\sqrt{\lambda} = \frac{\sqrt{6}\pi\sqrt{\kappa}}{\beta} + O\left(\frac{1}{\beta^2}\right). \tag{1.9}$$

This means that the leading instanton action vanishes in this limit and a new perturbative series at large $\kappa$ emerges, as we discuss in subsection 5.3.

In section 6 we apply our matrix model techniques to study integrated correlators in $\mathcal{N} = 4$ SYM. This analysis is technically much simpler than the one of extremal correlators

in $\mathcal{N} = 2$ SQCD, owing to the simpler mixing structure of $\mathcal{N} = 4$ SYM. One important difference is that, for $\mathcal{N} = 2$ SQCD, subleading corrections to the 't Hooft limit take contribution from subleading corrections to the $\mathcal{N} = 2$ mixing structure. Conversely, in the context of integrated correlators in $\mathcal{N} = 4$ SYM, our matrix models also capture all subleading corrections to the 't Hooft limit, as well as the large $m, n$ limits at fixed $\mathrm{Im}\tau$, including subleading and non-perturbative effects.

Integrated correlators are defined by a certain spacetime integral of [14]

$$\langle \Psi_1^0(x_1, y_1) \Psi_1^0(x_2, y_2) \Psi_n^m(x_3, y_3) \Psi_n^m(x_4, y_4) \rangle_{\mathbb{R}^4}$$

where $\Psi_j^i(x, y)$ are half-BPS operators in $\mathcal{N} = 4$ SYM, see discussion around (6.8). For a particular choice of the polarization vectors $y_i$, these reduced to extremal correlarors in $\mathcal{N} = 4$ SYM, but for a generic polarization they have a richer structure. Similar to the case of extremal correlators, we observe the emergence of two distinct matrix models: a Jacobi matrix model that governs the index $m$, and a Wishart matrix model that governs the index $n$. For $m = 0$ and $n$ large our results agree with [24, 26] and we have a description in terms of a Wishart model only. In addition we can perform a systematic analysis for any fixed $m$, see subsubsection 6.2.1. On the other hand, when taking $n$ finite and $m$ to be large, we find a description involving only a Jacobi matrix model, see subsubsection 6.2.2. Interestingly, in this regime, the behaviour is almost identical to the one of rank one case. Finally, the analysis of the large $m, n$ regime involve a combination of both type of matrix models, see subsubsection 6.2.3. Similar to the example of extremal correlators in SQCD, in the 't Hooft limit (1.3), and for $\lambda, \kappa \gg 1$, we find the leading instanton effect to be of the form

$$\mathrm{e}^{-A_1(\beta)\sqrt{\lambda}}, \quad \text{with} \quad A_1(\beta) = \frac{6\pi\sqrt{2}}{\sqrt{2\beta + 2\sqrt{\beta(\beta + 3)} + 3}}, \quad \beta = \frac{n}{m} . \tag{1.10}$$

In particular for $\beta \to \infty$ this instanton action vanishes causing a new perurbative serie at large $\kappa$ to emerge, see subsubsection 6.2.4.

We conclude in section 7 with a discussion on some open questions. There are also four appendices. In Appendix A, we collect background materials on matrix models. In Appendix B, we provide some numerical tests of (1.8). Appendix C contains technical details on resurgence. Finally, in Appendix D, we summarize the conventions for the different operators appearing in the paper.

## 2  Extremal correlators

In this paper, we study extremal correlators in four-dimensional $\mathcal{N} = 2$ superconformal field theories (SCFTs). The underlying superconformal algebra is $SU(2, 2|2)$, which extends the standard Poincaré and conformal generators by including additional supersymmetry generators. Specifically, this algebra has eight Poincaré supercharges, denoted as $Q_\alpha^a$ and $\overline{Q}_{\dot\alpha}^a$, and eight conformal supercharges, denoted as $S_\alpha^a$ and $\overline{S}_{\dot\alpha}^a$ where $a = 1, 2$ and $\alpha, \dot\alpha = 1, \ldots, 4$.

Furthermore, the algebra incorporates an $\mathfrak{su}(2)_R \times \mathfrak{u}(1)_R$ R-symmetry. See [25, Sec. 1] for a concise review and a list of references. Here we focus on a particular class of operators known as (anti-) chiral primary operators, or Coulomb branch operators. Specifically, chiral primary operators $\mathcal{O}_I$ are those annihilated by $S, \overline{S}$, and $Q$, while anti-chiral primary operators $\overline{\mathcal{O}}_I$ are annihilated by $S, \overline{S}$, and $\overline{Q}$. These operators are Lorentz scalars and $\mathfrak{su}(2)_R$ singlets. It follows from unitarity and supersymmetry that the scaling dimension $\Delta$ and the $\mathfrak{u}(1)_R$ R-charge $R$ of these operators are related by

$$
\begin{aligned}
\Delta &= \frac{R}{2}, \qquad \text{for chiral }, \\
\Delta &= -\frac{R}{2}, \qquad \text{for anti-chiral }.
\end{aligned}
\tag{2.1}
$$

An important property of these operators is the fact that their OPE is non-singular

$$
\mathcal{O}_I(x)\mathcal{O}_I(0) = \sum_K C_{IJ}^K \mathcal{O}_K(0) + \cdots
\tag{2.2}
$$

where $\cdots$ denote regular terms when $x \to 0$. Therefore (anti-) chiral primaries endowed with the OPE form a ring, often dubbed chiral ring, which is commutative and freely generated[2]. Thanks to this property it is always possible to choose a basis of operators and a normalization such that the structure constants are trivial and the OPE reads

$$
\mathcal{O}_I(x)\mathcal{O}_J(x) = \mathcal{O}_I\mathcal{O}_J(x) .
\tag{2.3}
$$

Within this normalization the two point function of chiral and anti-chiral operators becomes however non-trivial

$$
\langle \mathcal{O}_I(x)\overline{\mathcal{O}}_J(0)\rangle_{\mathbb{R}_4} = \frac{G_{IJ}}{|x|^{2\Delta_I}}\delta_{\Delta_I \Delta_J},
\tag{2.4}
$$

where $G_{IJ}$ is a non-trivial function of the marginal couplings.

Here we focus on two specific $\mathcal{N} = 2$ four-dimensional SCFT:

1. $\mathcal{N} = 4$ SYM with gauge group $SU(N)$ whose matter content consists in a $\mathcal{N} = 2$ vector multiplet together with one (massless) adjoint hypermultiplet.

2. $\mathcal{N} = 2$ $SU(N)$ SQCD whose matter content consists in a $\mathcal{N} = 2$ vector multiplet togethre with $N_f = 2N$ (massless) fundamental hypermultiplets.

These theories have a marginal coupling

$$
\tau = \frac{4\pi \mathrm{i}}{g_{\text{YM}}^2} + \frac{\theta}{2\pi}, \quad \theta, g_{\text{YM}} \in \mathbb{R}
\tag{2.5}
$$

which parametrizes the conformal manifold. The chiral ring is generated by

$$
\phi_k = \text{Tr}(\varphi^k), \quad k = 2, \cdots, N
\tag{2.6}
$$

---

[2]For non-Lagrangian theories this is still conjectural, but in this paper we focus on Lagrangian theories.

where $\varphi$ is the complex adjoint scalars in the $\mathcal{N} = 2$ vector multiplet. The operators (2.6) have dimension

$$\Delta(\phi_k) = k. \tag{2.7}$$

In addition the $\phi_k$'s are charged under the global $\mathtt{u(1)_R}$ symmetry, with the corresponding $R$ charge given by

$$R = 2\Delta(\phi_k) = 2k . \tag{2.8}$$

It is convenient to denote "elementary" coulomb branch operator by

$$\mathcal{O}_{\boldsymbol{n}} = \prod_{k=2}^{N} (\phi_k)^{n_k}, \quad \boldsymbol{n} = \{n_2, \ldots, n_N\} \tag{2.9}$$

whose dimension is

$$\Delta(\Phi_{\boldsymbol{n}}) = \sum_{k=2}^{N} n_k k. \tag{2.10}$$

For these operators the OPE (2.3) simply reads

$$\mathcal{O}_{\boldsymbol{n_1}}(x)\mathcal{O}_{\boldsymbol{n_2}}(x) = \mathcal{O}_{\boldsymbol{n_1 + n_2}}(x) . \tag{2.11}$$

The main object of study in this paper are extremal correlators of Coulomb branch operators. These are correlations function involving one anti-chiral primary $\overline{\mathcal{O}}_J$ with an arbitrary number of chiral primaries $\mathcal{O}_{I_k}$, namely

$$\left\langle \prod_{\ell=1}^{n} \mathcal{O}_{I_\ell}(x_\ell)\overline{\mathcal{O}_J}(y) \right\rangle_{\mathbb{R}^4} \tag{2.12}$$

where the dimensions $\Delta$ of the operators are subject to the constraint

$$\Delta(\overline{\mathcal{O}}_J) = \sum_{\ell=1}^{n} \Delta(\mathcal{O}_{I_\ell}). \tag{2.13}$$

For a comprehensive review of the topic and additional references, we refer to [13, 25]. An interesting characteristic of such correlation functions is the fact that their space-time dependence is very simple, specifically we have [27]

$$\left\langle \prod_{\ell=1}^{n} \mathcal{O}_{I_\ell}(x_\ell)\overline{\mathcal{O}_J}(y) \right\rangle_{\mathbb{R}^4} = G_{I_1,\ldots,I_n}(\tau, \bar{\tau}) \prod_{\ell=1}^{n} |y - x_\ell|^{-2\Delta(\mathcal{O}_{I_\ell})} . \tag{2.14}$$

Moreover, no spacetime singularity is encountered in the limit $x_\ell \to x_j$, and accordingly no singular terms appears in the OPE. Therefore we can apply (2.11) repeatedly and reduce any extremal correlators (2.14) to two point functions

$$\left\langle \mathcal{O}_I(0)\overline{\mathcal{O}_J}(\infty) \right\rangle_{\mathbb{R}^4} = G_{IJ}(\tau, \bar{\tau}) , \tag{2.15}$$

where we use $\mathcal{O}_I(\infty) = \lim_{x\to\infty} |x|^{2\Delta(\mathcal{O}_I)}\mathcal{O}_I(x)$. In particular, if we determine all the two-point functions (2.15) of the elements of a chiral ring basis, then we can reconstruct all extremal correlators. For sake of notation we will often omit the $\tau, \overline{\tau}$ dependence and simply note

$$G_{IJ} \equiv G_{IJ}(\tau, \overline{\tau}) . \tag{2.16}$$

Extremal correlators in four-dimensional $\mathcal{N} = 2$ theories have been extensively studied in recent years. In particular in [25] a systematic algorithm was found to compute such correlators on $\mathbb{R}^4$. The strategy adopted in [25] is to first compute these correlators on $S^4$, using localization techniques [28–32], and then move on to $\mathbb{R}^4$. For instance, two-point functions on $S^4$ of operators of the form (2.9) are given by [25, 32, 33][3]

$$\langle \mathcal{O}_{\boldsymbol{n}}(N)\overline{\mathcal{O}_{\boldsymbol{m}}(S)}\rangle_{S^4} = \frac{\prod_{i=2}^N (-\mathrm{i}\pi^{i/2})^{-n_i}(\mathrm{i}\pi^{i/2})^{-m_i}}{Z_{S^4}(\tau, \overline{\tau}; \boldsymbol{0}, \boldsymbol{0})} \partial_\tau^{n_1}\partial_{\overline{\tau}}^{m_1} \prod_{i=3}^N \partial_{\tau^i}^{n_i}\partial_{\overline{\tau}^i}^{m_i} Z_{S^4}(\tau, \overline{\tau}; \tau^A, \overline{\tau}^A)\Big|_{\tau^A = \overline{\tau}^A = 0} \tag{2.17}$$

where $Z_{S^4}(\tau, \overline{\tau}; \tau^A, \overline{\tau}^A)$ is the $SU(N)$ partition function of the theory of interest, with $N$ and $S$ in (2.17) denoting insertion at the North and South pole, respectively, and $\tau^A$, $A = 3, \ldots, N$ are the sources. Any other operator in the chiral ring can be written in a unique way as a polynomial in the $\mathcal{O}_{\boldsymbol{m}}$'s and therefore the corresponding two point functions follows from (2.17) and (2.11). We provide concrete examples below.

An important point is the fact that, although $\mathbb{R}^4$ and $S^4$ are conformally equivalent, the dictionary between the correlation functions on $S^4$ and the correlation functions on $\mathbb{R}^4$ is highly nontrivial because of conformal anomalies leading to operator mixing on the sphere [25]. Because of that, (2.17) can not be directly interpreted as a correlation function on $\mathbb{R}^4$. Indeed, on the sphere, a Coulomb branch operator $\mathcal{O}_\Delta$ of dimension $\Delta$, can mix with any other operator of dimension $\Delta - 2k$, $k = 1, 2, 3, ..$ because of the presence of a scale, i.e. the radius of the sphere. Hence we have[4]

$$\mathcal{O}_\Delta \;\to\; \mathcal{O}_\Delta + \alpha_1 R\mathcal{O}_{\Delta-2} + \alpha_2 R^2\mathcal{O}_{\Delta-4} + \cdots + \alpha_{\Delta_0/2}R^{\Delta_0/2}\mathbb{I} . \tag{2.18}$$

where $R$ is the Ricci scalar and $\alpha_i$ some dimensionless coefficients, so that each term on the rhs of (2.18) has the same dimension. Hence, to compute correlations functions on $\mathbb{R}^4$ starting from $S^4$, we need to find a way of dealing with this mixing. In [25] it was proposed that this issue can be resolved via the Gram-Schmidt (GS) procedure by finding a new orthogonal basis of operators $\{\mathcal{O}'_J\}$ on the sphere such that [25]

$$\langle \mathcal{O}'_J(N)\overline{\mathcal{O}'_I(S)}\rangle_{S^4} = 0 \quad \text{if} \quad \Delta(\mathcal{O}'_J) < \Delta(\mathcal{O}'_I) . \tag{2.19}$$

Then, schematically, we have

$$\langle \mathcal{O}_J(N)\overline{\mathcal{O}_J(S)}\rangle_{\mathbb{R}^4} = \langle \mathcal{O}'_J(N)\overline{\mathcal{O}'_J(S)}\rangle_{S^4} . \tag{2.20}$$

---

[3]Because of supersymmetry chiral operators are inserted at the North pole while anti-chiral operators are inserted at the South pole, see [25] and reference therein.

[4]Note that chiral operator can only mix among themselves [25].

We will review this procedure below.

For sake of notation in the rest of the paper we will omit the position of insertion points, that is

$$\langle \mathcal{O}_I \overline{\mathcal{O}_J} \rangle_{S^4} \equiv \langle \mathcal{O}_I(N) \overline{\mathcal{O}_J}(S) \rangle_{S^4} \ ,$$
$$\langle \mathcal{O}_I \overline{\mathcal{O}_J} \rangle_{\mathbb{R}^4} \equiv \langle \mathcal{O}_I(0) \overline{\mathcal{O}_J}(\infty) \rangle_{\mathbb{R}^4} \ . \tag{2.21}$$

We will also note

$$Z_{\mathrm{S}^4} \equiv Z_{S^4}(\tau, \overline{\tau}; 0, 0) \ . \tag{2.22}$$

## 2.1 Rank 1 theories

When the theory has rank one the chiral ring has only one generator. For $SU(2)$, $\mathcal{N} = 2$ SQCD and $SU(2)$, $\mathcal{N} = 4$ SYM this is

$$\mathrm{Tr}(\varphi^2) \tag{2.23}$$

and Coulomb branch operators are of the form $\mathcal{O}_n = (\mathrm{Tr}(\varphi^2))^n$. The CFT data simply consist of the two point functions

$$G_n = \langle \mathcal{O}_n \overline{\mathcal{O}_n} \rangle_{\mathbb{R}^4}. \tag{2.24}$$

In [25], the correlation functions (2.24) are computed starting from the two-point correlations on the sphere as follows. Let $M^{(n)}$ denote the $n \times n$ matrix whose elements are

$$M_{i,j} = \langle \mathcal{O}_i \overline{\mathcal{O}_j} \rangle_{S^4}, \quad i,j = 0, \ldots n - 1 \tag{2.25}$$

where

$$\langle \mathcal{O}_i \overline{\mathcal{O}_j} \rangle_{S^4} = \frac{1}{Z_{S^4}} ((-i\pi)^{-1} \partial_\tau)^i ((i\pi)^{-1} \partial_{\overline{\tau}})^j Z_{S^4} \ . \tag{2.26}$$

Since the operators $\mathcal{O}_n$ are not orthogonal, that is

$$\langle \mathcal{O}_i \overline{\mathcal{O}_j} \rangle_{S^4} \neq 0 \tag{2.27}$$

they undergo operator mixing. This mixing can be resolved by using the following orthogonal basis of operators

$$\mathcal{O}'_n = \mathcal{O}_n + \sum_{j=0}^{n-1} (-1)^{j+n+1} \frac{\det R^{(j)}}{\det R^{(n)}} \mathcal{O}_j \tag{2.28}$$

where $R^{(j)}$ is obtained from $M^{(n+1)}$ by erasing the $j+1$ column and the $n+1$ row. Equation (2.28) is nothing else that the usual GS procedure where the scalar product is given by the two-point function on $S^4$. Then we obtain [25]

$$\langle \mathcal{O}_n \overline{\mathcal{O}_n} \rangle_{\mathbb{R}^4} = \langle \mathcal{O}'_n \overline{\mathcal{O}'_n} \rangle_{S^4} = \frac{\det M^{(n+1)}}{\det M^{(n)}}, \tag{2.29}$$

that is

$$G_n = \frac{\det M^{(n+1)}}{\det M^{(n)}} \ . \tag{2.30}$$

The result (2.30) is very powerful because it provides a systematic solution to the operator mixing and makes it manifest the link between the rank 1 correlators $G_n$ and the Toda equations [25, 34–36]. In addition, as we will discuss later, representing the correlators as a ratio of determinants also makes it manifest the emergence of matrix models.

## 2.2 $SU(N)$, $\mathcal{N} = 4$ SYM

The $\tau$ dependence in the partition function of $\mathcal{N} = 4$ SYM is particularly simple as it is tree-level exact:

$$Z_{S^4}(\tau, \overline{\tau}; \tau^A, \overline{\tau}^A) = \int_{\mathbb{R}^{N-1}} \prod_{i=1}^{N-1} da_i \left( \prod_{1 \leq i < j \leq N} (a_i - a_j)^2 \right) e^{-2\pi \text{Im}(\tau)\text{Tr}(\varphi^2) - 2\sum_{A=3}^{N} \pi^{A/2} \text{Im}(\tau^A)\text{Tr}(\varphi^A)} \tag{2.31}$$

where

$$\varphi = \text{diag}(a_1, a_2, \cdots a_N), \quad \sum_{i=1}^{N} a_i = 0 \ . \tag{2.32}$$

In this case one can resolve the mixing following [25, Sec. (3.2.1)].

We start with a set of coulomb branch operators which do not contain $\phi_2$, i.e. operators of the form

$$\prod_{k=3}^{N} (\phi_k)^{n_k} \ . \tag{2.33}$$

We order them in such a way that their dimension $\Delta$ is growing. We note the ordered set of operator by $\{B_m\}_{m \geq 0}$ where by costruction $\Delta(B_m) \leq \Delta(B_{m+1})$. Hence $B_0 = \mathbb{I}$, $B_1 = \phi_3$, $\ldots$. We then construct another set of operators $\mathcal{O}_0^m$ which are built from $B_m$ by employing the GS procedure on $S^4$ with operators of the same dimension which contains $B_{m'}$ as long as $\Delta(B_{m'}) < \Delta(B_m)$. In this way we costruct a family of operators which are independent on $\tau$ and such that

$$\left\langle \mathcal{O}_0^m \overline{\mathcal{O}_0^{m'}} \right\rangle_{S^4}^{\mathcal{N}=4} = 0 \quad \text{if} \quad m \neq m' \ . \tag{2.34}$$

Starting from $\mathcal{O}_0^m$ we built a tower of operators defined by

$$\mathcal{O}_m^n = (\phi_2)^m \mathcal{O}_0^n \ . \tag{2.35}$$

Since $\mathcal{O}_0^m$ are $\tau$-independent, because of the particular form of the $\mathcal{N} = 4$ partition function (2.31), we also have the important implication

$$\left\langle \mathcal{O}_0^m \overline{\mathcal{O}_0^{m'}} \right\rangle_{S^4}^{\mathcal{N}=4} = 0 \quad \Longrightarrow \quad \left\langle \phi_2^a \mathcal{O}_0^m \overline{\phi_2^b \mathcal{O}_0^{m'}} \right\rangle_{S^4}^{\mathcal{N}=4} = 0 \tag{2.36}$$

when $m \neq m'$. To obtain an orthogonal family, we simply need to do a GS procedure on each tower constructed from a given $\mathcal{O}_0^m$ by acting with $\phi_2$, parallel to to the rank 1 example. Therefore we define $K_m^{(n)}$ to be the $n \times n$ matrix whose elements are

$$K_{i,j} = \left\langle \mathcal{O}_i^m \overline{\mathcal{O}_j^m} \right\rangle_{S^4}^{\mathcal{N}=4} \quad i, j = 0, \ldots, n-1 \ . \tag{2.37}$$

Then the orthogonal operators on the sphere are[5]

$$(\mathcal{O}_n^m)' = \mathcal{O}_n^m + \sum_{j=0}^{n-1}(-1)^{j+n+1}\frac{\det R^{(j)}}{\det R^{(n)}}\mathcal{O}_j^m, \tag{2.38}$$

where $R^{(j)}$ is obtained from $K_m^{(n+1)}$ by erasing the $j+1$ column and the $n+1$ row. We have

$$(G_n^m)^{\mathcal{N}=4} = \left\langle \mathcal{O}_n^m \overline{\mathcal{O}_n^m}\right\rangle_{\mathbb{R}^4}^{\mathcal{N}=4} = \left\langle (\mathcal{O}_n^m)'\overline{(\mathcal{O}_n^m)'}\right\rangle_{S4}^{\mathcal{N}=4} = \frac{\det K_m^{(n+1)}}{\det K_m^{(n)}}. \tag{2.39}$$

It follows that index $n$ in (2.39) is governed by a semi-infinite Toda Chain. This integrable structure was exploited in [25] to derive the following expression [25, eq.(3.39)]

$$\left(\frac{G_n^m}{G_0^m}\right)^{\mathcal{N}=4} = \frac{4^n n!}{(\mathrm{Im}\tau)^{2n}}\left(\frac{N^2-1}{2} + \Delta(B_m)\right)_n \tag{2.40}$$

where $(\cdot)_n$ is the Pochhammer symbol. However, since the index $m$ is not governed by the Toda chain, one can not fix $G_0^m$ in this way.

### 2.2.1 The $SU(3)$ case

For our propose we write the $SU(3)$ $\mathcal{N}=4$ partition function (2.31) using the variables

$$y = \phi_2 \in \mathbb{R}_+, \quad x = \frac{6\phi_3^2}{\phi_2^3} \in [0,1]. \tag{2.41}$$

As we discuss later, this change of variable is crucial to see the emergence of matrix models. In these new variables we get

$$Z_{S4}(\tau,\overline{\tau},\tau_3,\overline{\tau_3}) = \int_{\mathbb{R}_+} \mathrm{d}y \int_0^1 \mathrm{d}x \, \frac{y^3}{2\sqrt{3}}\sqrt{\frac{1}{x}-1} \, e^{-2\pi\mathrm{Im}\tau y - \frac{2\pi^{3/2}}{6^{1/2}}\mathrm{Im}\tau_3\sqrt{xy^3}} \tag{2.42}$$

and

$$Z_{S4} = \frac{\sqrt{3}}{32\pi^3\mathrm{Im}\tau^4}. \tag{2.43}$$

We take as staring point $B_m = \phi_3^m$, $m \geq 0$. The $\alpha \times \alpha$ matrix $M_m^{(\alpha)}$ has elements

$$M_{i,j} = \left\langle \left(\frac{\phi_3^2}{\phi_2^3}\right)^i \phi_3^{\sigma m}\phi_2^{3\lfloor m/2\rfloor}\overline{\left(\frac{\phi_3^2}{\phi_2^3}\right)^j \phi_3^{\sigma m}\phi_2^{3\lfloor m/2\rfloor}}\right\rangle_{S4}^{\mathcal{N}=4}, \quad i,j = 0,\dots,\alpha-1, \tag{2.44}$$

---

[5]Note that since $\mathcal{O}_0^m$ are already orthogonal on the sphere we simply have $(\mathcal{O}_0^m)' = \mathcal{O}_0^m$

where $\lfloor \cdot \rfloor$ is the floor function and $\sigma_m = m \bmod 2$. Assuming $m$ even, the spectrum of operators is organized as follows

$$\Delta = 3m: \qquad \mathcal{O}_0^m, \quad \mathcal{O}_3^{m-2}, \quad \mathcal{O}_6^{m-4}, \quad \ldots, \mathcal{O}_{\frac{3}{2}m}^0, \tag{2.45}$$

$$\Delta = 3m + 2: \qquad \mathcal{O}_1^m, \quad \mathcal{O}_4^{m-2}, \quad \mathcal{O}_5^{m-4}, \quad \ldots, \mathcal{O}_{\frac{3}{2}m+1}^0, \tag{2.46}$$

$$\Delta = 3m + 3: \qquad \mathcal{O}_0^{m+1}, \quad \mathcal{O}_3^{m-1}, \quad \mathcal{O}_6^{m-3}, \quad \ldots, \mathcal{O}_{\frac{3}{2}m}^1, \tag{2.47}$$

$$\Delta = 3m + 4: \qquad \mathcal{O}_2^m, \quad \mathcal{O}_5^{m-2}, \quad \mathcal{O}_8^{m-6}, \quad \ldots, \mathcal{O}_{\frac{3}{2}m+2}^0, \tag{2.48}$$

$$\Delta = 3m + 5: \qquad \mathcal{O}_1^{m+1}, \quad \mathcal{O}_4^{m-1}, \quad \mathcal{O}_7^{m-3}, \quad \ldots, \mathcal{O}_{\frac{3}{2}m+1}^1, \tag{2.49}$$

$$\vdots \tag{2.50}$$

Similar for the case of $m$ odd. The $\mathcal{O}_n^m$ operators in this $SU(3)$ example then reads

$$\mathcal{O}_0^m = \phi_3^m + \sum_{j=0}^{\lfloor m/2 \rfloor - 1} (-1)^{j + \lfloor m/2 \rfloor} \frac{\det R^{(j)}}{\det R^{\lfloor m/2 \rfloor}} \phi_3^{2j + \sigma_m} \phi_2^{3(\lfloor m/2 \rfloor - j)}$$

$$\mathcal{O}_n^m = \phi_2^n \mathcal{O}_0^m \tag{2.51}$$

where $R_3^{(k)}$ is obtained from $M_m^{\lfloor m/2 \rfloor + 1}$ by erasing the $k^{\text{th}} + 1$ column and the $\lfloor m/2 \rfloor + 1$ row. This gives

$$\mathcal{O}_0^1 = \phi_3$$

$$\mathcal{O}_0^2 = \phi_3^2 - \frac{\langle \phi_2^3 \phi_3^2 \rangle_{S^4}}{\langle \phi_2^6 \rangle_{S^4}} \phi_2^3 = \phi_3^2 - \frac{1}{24} \phi_2^3$$

$$\mathcal{O}_0^3 = \phi_3^3 - \frac{\langle \phi_3^4 \phi_2^3 \rangle_{S^4}}{\langle \phi_2^6 \phi_3^2 \rangle_{S^4}} \phi_3 \phi_2^3 = \phi_3^3 - \frac{1}{12} \phi_3 \phi_2^3$$

$$\mathcal{O}_0^4 = \phi_3^4 - \frac{1}{8} \phi_2^3 \phi_3^2 + \frac{1}{576} \phi_2^6 \tag{2.52}$$

$$\mathcal{O}_0^5 = \phi_3^5 + \frac{\phi_2^6 \phi_3}{192} - \frac{1}{6} \phi_2^3 \phi_3^3$$

$$\mathcal{O}_0^6 = \phi_3^6 - \frac{5}{24} \phi_2^3 \phi_3^4 + \frac{\phi_2^6 \phi_3^2}{96} - \frac{\phi_2^9}{13824}$$

$$\vdots$$

One can also write (2.51) in a more compact form similarly to [25], that is

$$\mathcal{O}_0^m = \phi_3^m - \sum_{j=1}^m \frac{\left\langle \phi_3^m \overline{\mathcal{O}_0^{m-2j}} \right\rangle_{S^4}^{\mathcal{N}=4}}{\left\langle \mathcal{O}_{3j}^{m-2j} \overline{\mathcal{O}_0^{m-2j}} \right\rangle_{S^4}^{\mathcal{N}=4}} \mathcal{O}_{3j}^{m-2j}. \tag{2.53}$$

Using (2.11), the OPE for these operators can be recursively derived from the simple relations

$$\mathcal{O}_0^1 \mathcal{O}_0^m = \mathcal{O}_0^{m+1} + \frac{1}{24} \mathcal{O}_3^{m-1},$$

$$\mathcal{O}_{n_1}^0 \mathcal{O}_{n_2}^m = \mathcal{O}_{n_1+n_2}^m. \tag{2.54}$$

For example

$$\mathcal{O}_0^2 \mathcal{O}_0^m = \left(\mathcal{O}_0^1 \mathcal{O}_0^1 - \frac{1}{24}\mathcal{O}_3^{(0)}\right)\mathcal{O}_0^m = \mathcal{O}_0^{m+2} + \frac{1}{24}\mathcal{O}_3^m + \left(\frac{1}{24}\right)^2 \mathcal{O}_6^{m-2}. \qquad (2.55)$$

One can easily check that the $\mathcal{O}_0^m$ operators are indeed orthogonal, that is

$$\langle \mathcal{O}_0^m \overline{\mathcal{O}_0^{m'}}\rangle_{S^4} = 0, \quad m \neq m', \qquad (2.56)$$

and we have

$$(G_0^m)^{\mathcal{N}=4} = \langle \mathcal{O}_0^m \overline{\mathcal{O}_0^m}\rangle_{\mathbb{R}^4} = \langle \mathcal{O}_0^m \overline{\mathcal{O}_0^m}\rangle_{S^4}. \qquad (2.57)$$

By explicitly computing the $S^4$ correlators we get

$$(G_0^m)^{\mathcal{N}=4} = Z_{S^4}^{-1} \frac{\det_{k,\ell=0,\dots,\lfloor\frac{m}{2}\rfloor}\int_{\mathbb{R}_+}\mathrm{d}y\, y^{3+3m}\mathrm{e}^{-2\pi\mathrm{Im}\tau y}\int_0^1 \frac{\mathrm{d}x}{2\sqrt{3}}\sqrt{\frac{1}{x}-1}\left(\frac{x}{6}\right)^{n+\ell+\sigma_m}}{\det_{k,\ell=0,\dots,\lfloor\frac{m}{2}\rfloor-1}\int_{\mathbb{R}_+}\mathrm{d}y\, y^{3+3m}\mathrm{e}^{-2\pi\mathrm{Im}\tau y}\int_0^1 \frac{\mathrm{d}x}{2\sqrt{3}}\sqrt{\frac{1}{x}-1}\left(\frac{x}{6}\right)^{n+\ell+\sigma_m}}$$
$$= \frac{\det M_m^{(\lfloor m/2\rfloor+1)}}{\det M_m^{(\lfloor m/2\rfloor)}} \qquad (2.58)$$

where $M_m^{(\alpha)}$ is defined in (2.44).

More generically if we consider $n \neq 0$ we need to take into account an additional GS orthogonalization, namely (2.38). This lead to

$$\langle \mathcal{O}_n^m \overline{\mathcal{O}_n^m}\rangle_{\mathbb{R}^4}^{\mathcal{N}=4} = \left\langle (\mathcal{O}_n^m)'\overline{(\mathcal{O}_n^m)}'\right\rangle_{S^4}^{\mathcal{N}=4} = Z_{S^4}^{-1} \frac{\det_{i,j=0,\dots,n}(-\partial_{2\pi\mathrm{Im}\tau})^{i+j}\left(Z_{S^4}\frac{\det M_m^{(\lfloor m/2\rfloor+1)}}{\det M_m^{(\lfloor m/2\rfloor)}}\right)}{\det_{i,j=0,\dots,n-1}(-\partial_{2\pi\mathrm{Im}\tau})^{i+j}\left(Z_{S^4}\frac{\det M_m^{(\lfloor m/2\rfloor+1)}}{\det M_m^{(\lfloor m/2\rfloor)}}\right)}. \qquad (2.59)$$

For instance we have

$$\left\langle\mathcal{O}_0^2\overline{\mathcal{O}_0^2}\right\rangle_{\mathbb{R}^4}^{\mathcal{N}=4} = \frac{105}{(2\pi\mathrm{Im}\tau)^6}, \qquad \left\langle\mathcal{O}_2^2\overline{\mathcal{O}_2^2}\right\rangle_{\mathbb{R}^4}^{\mathcal{N}=4} = \frac{23100}{(2\pi\mathrm{Im}\tau)^{10}},$$
$$\left\langle\mathcal{O}_0^5\overline{\mathcal{O}_0^5}\right\rangle_{\mathbb{R}^4}^{\mathcal{N}=4} = \frac{134008875}{(2\pi\mathrm{Im}\tau)^{15}}, \qquad \left\langle\mathcal{O}_1^5\overline{\mathcal{O}_1^5}\right\rangle_{\mathbb{R}^4}^{\mathcal{N}=4} = \frac{2546168625}{(2\pi\mathrm{Im}\tau)^{17}}. \qquad (2.60)$$

## 2.3 $SU(N)$ $\mathcal{N}=2$ **SQCD with** $N_f = 2N$

The $S^4$ partition function of $SU(N)$ $\mathcal{N}=2$ SQCD with $N_f = 2N$ has the following structure [32]

$$Z_{S^4}(\tau,\overline{\tau};\tau^A,\overline{\tau}^A) = \int_{\mathbb{R}^{N-1}} \prod_{i=1}^{N-1}\mathrm{d}a_i \left(\prod_{1\leq i<j\leq N}(a_i-a_j)^2\right)Z_{\mathrm{G}}(a_1,\dots,a_{N-1},\tau)$$
$$|Z_{\mathrm{inst}}(a_1,\dots,a_{N-1},\tau)|^2 \mathrm{e}^{-2\pi\mathrm{Im}(\tau)\mathrm{Tr}(\varphi^2)-2\sum_{A=3}^N \pi^{A/2}\mathrm{Im}(\tau^A)\mathrm{Tr}(\varphi^A)} \qquad (2.61)$$

where

$$\varphi = \mathrm{diag}(a_1, a_2, \cdots a_N), \quad \sum_{i=1}^N a_i = 0, \qquad (2.62)$$

$$Z_{\rm G}(a_1, \ldots, a_{N-1}) = \prod_{i \neq j} H(\mathrm{i}(a_i - a_j)) \prod_{i=1}^{N} H(\mathrm{i}a_i)^{-2N}, \quad H(x) = G(1+x)G(1-x) \quad (2.63)$$

$G$ being the Barnes G functions. In the context of localization we usually refer to (2.63) as the one-loop partition function. Moreover $Z_{\rm inst}(a_1, \ldots, a_{N-1}, \tau) = 1 + \mathcal{O}(\mathrm{e}^{2\pi\mathrm{i}\tau})$ is the instanton partition function in the $\epsilon_1 = \epsilon_2 = 1$ phase of the $\Omega$ background [28–31][6]. When considering extremal correlators with a very large number of insertions, we can neglect such instanton term[7]. The loop expansion of (2.61) can thus be obtain by simply Taylor expanding $Z_{\rm G}(a_1, \ldots, a_{N-1})$ around $a_i = 0$.

### 2.3.1 The SU(3) case

Let us work out some details for the example of $SU(3)$, $\mathcal{N} = 2$ SQCD with $N_f = 6$ flavours. The explicit expression for $Z_{\rm G}$ is

$$\begin{aligned}
Z_{\rm G}(a_1, a_2) = &\frac{H(ia_1 - ia_2)H(2ia_1 + ia_2))H(ia_1 + 2ia_2)H(-ia_1 + ia_2)}{(H(ia_1)H(ia_2)H(-ia_1 - ia_2))^6} \\
&\times H(-2ia_1 - ia_2))H(-ia_1 - 2ia_2).
\end{aligned} \quad (2.64)$$

The loop expansion of the $Z_{S^4}$ partition function (2.61) reads

$$\begin{aligned}
Z_{S^4} = &-\frac{\sqrt{3}}{32\pi^3 \mathrm{Im}\tau^4} + \frac{15\sqrt{3}\zeta(3)}{32\pi^5 \mathrm{Im}\tau^6} - \frac{425\zeta(5)}{64(\sqrt{3}\pi^6)\mathrm{Im}\tau^7} - \frac{35(324\zeta(3)^2 - 511\zeta(7))}{512(\sqrt{3}\pi^7)\mathrm{Im}\tau^8} \\
&+ O\left(\left(\frac{1}{\mathrm{Im}\tau}\right)^9\right) + \mathcal{O}(\mathrm{e}^{2\pi\mathrm{i}\tau}) + \mathcal{O}(\mathrm{e}^{2\pi\mathrm{i}\bar\tau}).
\end{aligned} \quad (2.65)$$

To compute correlation functions on $\mathbb{R}^4$ we need to disentangle the mixing with operators of the lower dimension [25]. Let us illustrate this in one example, more are given in [25, Sec. 3.3]. Let us compute $\langle \phi_3^2 \overline{\phi_3^2} \rangle_{\mathbb{R}^4}$. The results of [25] is that

$$\langle \phi_3^2 \overline{\phi_3^2} \rangle_{\mathbb{R}^4}^{\mathcal{N}=2} = \langle (\phi_3^2)' \overline{(\phi_3^2)'} \rangle_{S^4}^{\mathcal{N}=2} \quad (2.66)$$

where $(\phi_3^2)'$ is obtained starting from $\phi_3^2$ and by doing GS on the sphere w.r.t. operators of lower dimension[8], which in this case are $\phi_2^2, \phi_2, 1$. The scalar product in this GS procedure

---

[6]There are some subtleties related to the so-called $U(1)$ factor, but this will not be important here.

[7]Strictly speaking this is fully justified in the so-called double scaling limit, where we also take $\mathrm{Im}(\tau) \to \infty$. In the pure large charge limit such approximation require a careful treatment, a detailed discussion will appear in [37].

[8]Recall that the dimensions of the operators involved must differs by 2, see (2.18).

is the two-point function of the theory on $S^4$. More precisely:

$$(\phi_3^2)' = \frac{\det\begin{pmatrix} \langle 1 \rangle_{S^4}^{\mathcal{N}=2} & \langle \phi_2 \rangle_{S^4}^{\mathcal{N}=2} & \langle \phi_2^2 \rangle_{S^4}^{\mathcal{N}=2} & \langle \phi_3^2 \rangle_{S^4}^{\mathcal{N}=2} \\ \langle \overline{\phi_2} \rangle_{S^4}^{\mathcal{N}=2} & \langle \overline{\phi_2}\phi_2 \rangle_{S^4}^{\mathcal{N}=2} & \langle \overline{\phi_2}\phi_2^2 \rangle_{S^4}^{\mathcal{N}=2} & \langle \overline{\phi_2}\phi_3^2 \rangle_{S^4}^{\mathcal{N}=2} \\ \langle \overline{\phi_2^2} \rangle_{S^4}^{\mathcal{N}=2} & \langle \overline{\phi_2^2}\phi_2 \rangle_{S^4}^{\mathcal{N}=2} & \langle \overline{\phi_2^2}\phi_2^2 \rangle_{S^4}^{\mathcal{N}=2} & \langle \overline{\phi_2^2}\phi_3^2 \rangle_{S^4}^{\mathcal{N}=2} \\ 1 & \phi_2 & \phi_2^2 & \phi_3^2 \end{pmatrix}}{\det\begin{pmatrix} \langle 1 \rangle_{S^4}^{\mathcal{N}=2} & \langle \phi_2 \rangle_{S^4}^{\mathcal{N}=2} & \langle \phi_2^2 \rangle_{S^4}^{\mathcal{N}=2} \\ \langle \overline{\phi_2} \rangle_{S^4}^{\mathcal{N}=2} & \langle \overline{\phi_2}\phi_2 \rangle_{S^4}^{\mathcal{N}=2} & \langle \overline{\phi_2}\phi_2^2 \rangle_{S^4}^{\mathcal{N}=2} \\ \langle \overline{\phi_2^2} \rangle_{S^4}^{\mathcal{N}=2} & \langle \overline{\phi_2^2}\phi_2 \rangle_{S^4}^{\mathcal{N}=2} & \langle \overline{\phi_2^2}\phi_2^2 \rangle_{S^4}^{\mathcal{N}=2} \end{pmatrix}} \tag{2.67}$$

This gives

$$\langle \phi_3^2 \overline{\phi_3^2} \rangle_{\mathbb{R}^4}^{\mathcal{N}=2} = \frac{425}{256\pi^6 \mathrm{Im}\tau^6} - \frac{57645\zeta(3)}{512\pi^8 \mathrm{Im}\tau^8} + \frac{1688875\zeta(5)}{1536\pi^9 \mathrm{Im}\tau^9} + \frac{5(14749776\zeta(3)^2 - 25878125\zeta(7))}{12288\pi^{10}\mathrm{Im}\tau^{10}}$$

$$+ \frac{175(1285211\zeta(9) - 1608930\zeta(3)\zeta(5))}{2048\pi^{11}\mathrm{Im}\tau^{11}}$$

$$+ \frac{5(7516566435\zeta(7)\zeta(3) - 1490674752\zeta(3)^3 + 4068688250\zeta(5)^2 - 6201901090\zeta(11))}{24576\pi^{12}\mathrm{Im}\tau^{12}}$$

$$+ O\left(\left(\frac{1}{\mathrm{Im}\tau}\right)^{13}\right). \tag{2.68}$$

The same procedure can be applied to all other two-point functions.

For our purposes, it is convenient to work in a new basis of the chiral ring. Specifically, we define a new set of operators $\{\Theta_n^m\}_{m,n\geq 0}$ with dimension $\Delta = 3m + 2n$ as follows. The $\Theta_n^m$ operator is constructed starting from $\phi_2^n\phi_3^m$ and then applying GS orthogonalization on $\mathbb{R}^4$ with respect to all operators $\phi_3^j\phi_2^k$ such that $3j + 2k = 3m + 2n$ and $j < m$. That is

$$\Theta_n^m = \phi_2^n\phi_3^m + \sum_{j,k} c_{j,k}\phi_3^j\phi_2^k \tag{2.69}$$

where the coefficients are determined by the GS procedure on $\mathbb{R}_4$ and the sum is subject to the constraint $3j + 2k = 3m + 2n$ with $j < m$. For example we have

$$\Theta_n^0 = \phi_2^n$$

$$\Theta_0^1 = \phi_3$$

$$\Theta_1^1 = \phi_3\phi_2$$

$$\Theta_0^2 = \frac{\det\begin{pmatrix} \langle \phi_2^3\overline{\phi_2^3} \rangle_{\mathbb{R}^4}^{\mathcal{N}=2} & \langle \phi_2^3\overline{\phi_3^2} \rangle_{\mathbb{R}^4}^{\mathcal{N}=2} \\ \phi_2^3 & \phi_3^2 \end{pmatrix}}{\langle \phi_2^3\overline{\phi_2^3} \rangle_{\mathbb{R}^4}^{\mathcal{N}=2}} = \phi_3^2 - \frac{\langle \phi_2^3\overline{\phi_3^2} \rangle_{\mathbb{R}^4}^{\mathcal{N}=2}}{\langle \phi_2^3\overline{\phi_2^3} \rangle_{\mathbb{R}^4}^{\mathcal{N}=2}}\phi_2^3$$

$$\Theta_1^2 = \frac{\det\begin{pmatrix} \langle \phi_2^4\overline{\phi_2^4} \rangle_{\mathbb{R}^4}^{\mathcal{N}=2} & \langle \phi_2^4\overline{\phi_3^2\phi_2} \rangle_{\mathbb{R}^4}^{\mathcal{N}=2} \\ \phi_2^4 & \phi_3^2\phi_2 \end{pmatrix}}{\langle \phi_2^3\overline{\phi_2^3} \rangle_{\mathbb{R}^4}^{\mathcal{N}=2}} = \phi_3^2\phi_2 - \frac{\langle \phi_2^4\overline{\phi_3^2\phi_2} \rangle_{\mathbb{R}^4}^{\mathcal{N}=2}}{\langle \phi_2^4\overline{\phi_2^4} \rangle_{\mathbb{R}^4}^{\mathcal{N}=2}}\phi_2^4 \tag{2.70}$$

$$\vdots$$

By construction these operators are orthogonal on $\mathbb{R}^4$, that is

$$\langle \Theta_n^m \Theta_{n'}^{m'} \rangle_{\mathbb{R}_4}^{\mathcal{N}=2} = \delta_{nn'} \delta_{mm'} \langle \Theta_n^m \Theta_n^m \rangle_{\mathbb{R}_4}^{\mathcal{N}=2}. \tag{2.71}$$

Hence we define

$$(G_n^m)^{\mathcal{N}=2} = \langle \Theta_n^m \Theta_n^m \rangle_{\mathbb{R}^4}^{\mathcal{N}=2}. \tag{2.72}$$

In practice, to compute these correlation we go on $S^4$ and do a GS procedure on $S^4$ where we orthogonalize w.r.t. operator of the same and lower dimensions. More precisely let us define $(\Theta_n^m)'$ as the operator constructed starting from $\phi_2^n \phi_3^m$ and then applying GS orthogonalization on $S^4$ with respect to all operators $\phi_3^j \phi_2^k$ such that $3j + 2k \leq 3m + 2n$ and $j < m$. Then it follows from [25] that

$$\langle \Theta_n^m \overline{\Theta_n^m} \rangle_{\mathbb{R}^4}^{\mathcal{N}=2} = \langle (\Theta_n^m)' \overline{(\Theta_n^m)'} \rangle_{S^4}^{\mathcal{N}=2} , \tag{2.73}$$

which is the analogous of (2.29) for higher rank. To summarise, the key equation we will use is

$$(G_n^m)^{\mathcal{N}=2} = \langle \Theta_n^m \overline{\Theta_n^m} \rangle_{\mathbb{R}^4}^{\mathcal{N}=2} = \langle (\Theta_n^m)' \overline{(\Theta_n^m)'} \rangle_{S^4}^{\mathcal{N}=2} \tag{2.74}$$

Let us work out some examples. For the first few operators we have

$$\begin{aligned}
(\Theta_0^0)' &= 1 = \Theta_0^0 \,, \\
(\Theta_1^0)' &= \phi_2 - \langle \phi_2 \rangle_{S^4}^{\mathcal{N}=2} 1 = \Theta_1^0 - \langle \phi_2 \rangle_{S^4}^{\mathcal{N}=2} \Theta_0^0 \,, \\
(\Theta_0^1)' &= \phi_3 = \Theta_0^1 \,.
\end{aligned} \tag{2.75}$$

The first non-trivial example is $(\Theta_0^2)'$. In our construction this operator is obtained by applying GS on $S^4$ and w.r.t. the family $\boldsymbol{v} = \{1, \phi_2, \phi_2^2, \phi_2^3, \phi_3^2\}$[9]:

$$(\Theta_0^2)' = \cfrac{\begin{pmatrix} \langle 1 \rangle_{S^4} & \langle \phi_2 \rangle_{S^4} & \langle \phi_2^2 \rangle_{S^4} & \langle \phi_2^3 \rangle_{S^4} & \langle \phi_3^2 \rangle_{S^4} \\ \langle \overline{\phi_2} \rangle_{S^4} & \langle \overline{\phi_2}\phi_2 \rangle_{S^4} & \langle \overline{\phi_2}\phi_2^2 \rangle_{S^4} & \langle \overline{\phi_2}\phi_2^3 \rangle_{S^4} & \langle \overline{\phi_2}\phi_3^2 \rangle_{S^4} \\ \langle \overline{\phi_2^2} \rangle_{S^4} & \langle \overline{\phi_2^2}\phi_2 \rangle_{S^4} & \langle \overline{\phi_2^2}\phi_2^2 \rangle_{S^4} & \langle \overline{\phi_2^2}\phi_2^3 \rangle_{S^4} & \langle \overline{\phi_2^2}\phi_3^2 \rangle_{S^4} \\ \langle \overline{\phi_2^3} \rangle_{S^4} & \langle \overline{\phi_2^3}\phi_2 \rangle_{S^4} & \langle \overline{\phi_2^3}\phi_2^2 \rangle_{S^4} & \langle \overline{\phi_2^3}\phi_2^3 \rangle_{S^4} & \langle \overline{\phi_2^3}\phi_3^2 \rangle_{S^4} \\ 1 & \phi_2 & \phi_2^2 & \phi_2^3 & \phi_3^2 \end{pmatrix}}{\begin{pmatrix} \langle 1 \rangle_{S^4} & \langle \phi_2 \rangle_{S^4} & \langle \phi_2^2 \rangle_{S^4} & \langle \phi_2^3 \rangle_{S^4} \\ \langle \overline{\phi_2} \rangle_{S^4} & \langle \overline{\phi_2}\phi_2 \rangle_{S^4} & \langle \overline{\phi_2}\phi_2^2 \rangle_{S^4} & \langle \overline{\phi_2}\phi_2^3 \rangle_{S^4} \\ \langle \overline{\phi_2^2} \rangle_{S^4} & \langle \overline{\phi_2^2}\phi_2 \rangle_{S^4} & \langle \overline{\phi_2^2}\phi_2^2 \rangle_{S^4} & \langle \overline{\phi_2^2}\phi_2^3 \rangle_{S^4} \\ \langle \overline{\phi_2^3} \rangle_{S^4} & \langle \overline{\phi_2^3}\phi_2 \rangle_{S^4} & \langle \overline{\phi_2^3}\phi_2^2 \rangle_{S^4} & \langle \overline{\phi_2^3}\phi_2^3 \rangle_{S^4} \end{pmatrix}}$$
$$= \phi_3^2 + \sum_{j=1}^4 (-1)^{j+1} c_j \phi_2^{j-1} \tag{2.76}$$

---

[9]All the vevs in (2.76) are taken in the $\mathcal{N} = 2$ theory. We omit the superscript to light the notation.

where

$$
\begin{aligned}
c_1 =& -\frac{875\zeta(5)}{8\pi^6\mathrm{Im}\tau^6} + \frac{6125\zeta(7)}{2\pi^7\mathrm{Im}\tau^7} + \frac{18375(72\zeta(3)\zeta(5) - 451\zeta(9))}{128\pi^8\mathrm{Im}\tau^8} \\
& - \frac{1225(12075\zeta(5)^2 + 27720\zeta(3)\zeta(7) - 98857\zeta(11))}{96\pi^9\mathrm{Im}\tau^9} + \dots \\
c_2 =& -\frac{1575\zeta(5)}{8\pi^5\mathrm{Im}\tau^5} + \frac{42875\zeta(7)}{8\pi^6\mathrm{Im}\tau^6} + \frac{86625(6\zeta(3)\zeta(5) - 41\zeta(9))}{32\pi^7\mathrm{Im}\tau^7} \\
& - \frac{3675(34100\zeta(5)^2 + 77280\zeta(3)\zeta(7) - 296571\zeta(11))}{512\pi^8\mathrm{Im}\tau^8} + \dots \\
c_3 =& -\frac{1575\zeta(5)}{16\pi^4\mathrm{Im}\tau^4} + \frac{5145\zeta(7)}{2\pi^5\mathrm{Im}\tau^5} + \frac{525(414\zeta(3)\zeta(5) - 3157\zeta(9))}{32\pi^6\mathrm{Im}\tau^6} \\
& - \frac{105(31550\zeta(5)^2 + 70560\zeta(3)\zeta(7) - 296571\zeta(11))}{32\pi^7\mathrm{Im}\tau^7} + \dots \\
c_4 =& \frac{1}{24} - \frac{175\zeta(5)}{12\pi^3\mathrm{Im}\tau^3} + \frac{8575\zeta(7)}{24\pi^4\mathrm{Im}\tau^4} + \frac{35(360\zeta(3)\zeta(5) - 3157\zeta(9))}{16\pi^5\mathrm{Im}\tau^5} \\
& - \frac{35(199600\zeta(5)^2 + 441000\zeta(3)\zeta(7) - 2075997\zeta(11))}{576\pi^6\mathrm{Im}\tau^6} + \dots
\end{aligned}
\tag{2.77}
$$

Note that the $c_j$'s are $\tau$-dependent, therefore derivatives with respect to $\tau$ of $Z_{S^4}$ with $(\Theta_n^m)'$ insertions do not correspond to insertions of $\phi_2$. As a result (2.36) is not valid in $\mathcal{N} = 2$ theories. As we will discuss in some detail in the next section, the $\tau$-dependence only appears from 3 loops order on $S^4$ and at 6 loops on $\mathbb{R}^4$. By using (2.74) and (2.76), we find

$$
\begin{aligned}
\left(G_0^2\right)^{\mathcal{N}=2} =& \frac{105}{64\pi^6\mathrm{Im}\tau^6} - \frac{14175\zeta(3)}{128\pi^8\mathrm{Im}\tau^8} + \frac{139125\zeta(5)}{128\pi^9\mathrm{Im}\tau^9} + \frac{1575(477\zeta(3)^2 - 854\zeta(7))}{128\pi^{10}\mathrm{Im}\tau^{10}} \\
& - \frac{1575(21975\zeta(3)\zeta(5) - 18067\zeta(9))}{256\pi^{11}\mathrm{Im}\tau^{11}} \\
& + \frac{175(-3446712\zeta(3)^3 + 17781624\zeta(7)\zeta(3) + 9583250\zeta(5)^2 - 15220359\zeta(11))}{2048\pi^{12}\mathrm{Im}\tau^{12}} \\
& + O\left(\left(\frac{1}{\mathrm{Im}\tau}\right)^{13}\right).
\end{aligned}
\tag{2.78}
$$

# 3   A close-up on mixing in $SU(3)$ $\mathcal{N} = 2$ SQCD

We now investigate in detail the differences between mixing patterns in $\mathcal{N} = 2$ and $\mathcal{N} = 4$ with $SU(3)$ gauge group. As discussed in the previous section, in $\mathcal{N} = 4$ there is a class of operators $\mathcal{O}_0^m$ of dimension $\Delta(\mathcal{O}_0^m) = 3m$ with the following properties:

1. $\mathcal{O}_0^m$ can be obtained performing a Gram Schmidt procedure only with operators of the same dimensions,

2. $\mathcal{O}_0^m$ is $\tau$-independent and orthogonal to all operators with $\Delta < \Delta(\mathcal{O}_0^m)$ on the sphere.

For these operators we simply have

$$\langle \mathcal{O}_0^m \overline{\mathcal{O}_0^m} \rangle_{\mathbb{R}^4} = \langle \mathcal{O}_0^m \overline{\mathcal{O}_0^m} \rangle_{S^4} , \tag{3.1}$$

and from $\tau$ independence it follows that families $\{\mathcal{O}_n^m\}_{n \geq 0} = \{\phi_2^n \mathcal{O}_0^m\}_{n \geq 0}$ will only mix between themselves on the sphere. In the $\mathcal{N} = 2$ case we don't have a class of operators with these properties. In fact expanding (2.64) at 3 loops in terms of $\mathcal{N} = 4$ operators we find

$$Z_{\rm G}(a_1, a_2) = 1 - 3\zeta(3)\mathcal{O}_2^0 - \frac{20}{3}\zeta(5)\left(\mathcal{O}_0^2 - \frac{17}{24}\mathcal{O}_3^0\right) + \dots \tag{3.2}$$

The 2 loop term is proportional to $\mathcal{O}_2^0$ and doesn't create any mixing problem since it just amounts to take derivatives with respect to $\tau$. However the appearance of $\mathcal{O}_0^2$ at 3 loops spoils (2.36), and qualitatively changes the mixing pattern of $\mathcal{N} = 2$ operators on the sphere. For example from (2.54)

$$\langle \mathcal{O}_0^2 \mathcal{O}_2^0 \rangle_{S^4}^{\mathcal{N}=2} \simeq -\frac{20}{3}\zeta(5)\langle \mathcal{O}_0^2 \mathcal{O}_2^2 \rangle_{S^4}^{\mathcal{N}=4} . \tag{3.3}$$

Similarly $\mathcal{O}_0^2$ mixes on the sphere with $\mathcal{O}_0^0, \mathcal{O}_1^0$, and a non trivial orthogonalization with operators with smaller dimensions is necessary to find a well defined flat space correlator.

## 3.1 Mixing at six loops

Although the mixing pattern on the sphere changes at 3 loops, it will only produce 6 loops effects on $\mathbb{R}^4$. To see how this happens, it is convenient to write the Gram-Schmidt determinant that gives the flat space correlator in the $\mathcal{N} = 4$ basis of operators $\mathcal{O}_n^m$. Since the two point function $\langle \mathcal{O}_n^m \mathcal{O}_{n'}^{m'} \rangle_{S^4}^{\mathcal{N}=2}$ is different from zero only from three loop order on, if $m \neq m'$ we have

$$(G_n^m)^{\mathcal{N}=2} = \frac{\det \begin{pmatrix} \left(\langle \overline{\mathcal{O}_\ell^0} \mathcal{O}_{\ell'}^0 \rangle_{S^4}^{\mathcal{N}=2}\right)_{\ell,\ell'=0,\dots,n} & \mathcal{O}(3\mathrm{L}) & \dots \\ \mathcal{O}(3\mathrm{L}) & \left(\langle \overline{\mathcal{O}_\ell^2} \mathcal{O}_{\ell'}^2 \rangle_{S^4}^{\mathcal{N}=2}\right)_{\ell,\ell'=0,\dots,n} & \dots \\ \vdots & & \\ \mathcal{O}(3\mathrm{L}) & \dots & \left(\langle \overline{\mathcal{O}_\ell^m} \mathcal{O}_{\ell'}^m \rangle_{S^4}^{\mathcal{N}=2}\right)_{\ell,\ell'=0,\dots,n} \end{pmatrix}}{\det \begin{pmatrix} \left(\langle \overline{\mathcal{O}_\ell^0} \mathcal{O}_{\ell'}^0 \rangle_{S^4}^{\mathcal{N}=2}\right)_{\ell,\ell'=0,\dots,n} & \mathcal{O}(3\mathrm{L}) & \dots \\ \mathcal{O}(3\mathrm{L}) & \left(\langle \overline{\mathcal{O}_\ell^2} \mathcal{O}_{\ell'}^2 \rangle_{S^4}^{\mathcal{N}=2}\right)_{\ell,\ell'=0,\dots,n} & \dots \\ \vdots & & \\ \mathcal{O}(3\mathrm{L}) & \dots & \left(\langle \overline{\mathcal{O}_\ell^m} \mathcal{O}_{\ell'}^m \rangle_{S^4}^{\mathcal{N}=2}\right)_{\ell,\ell'=0,\dots,n-1} \end{pmatrix}} . \tag{3.4}$$

This immediately gives

$$(G_n^m)^{\mathcal{N}=2} = \frac{\det\left(\langle \overline{\mathcal{O}_\ell^m} \mathcal{O}_{\ell'}^m \rangle_{S^4}^{\mathcal{N}=2}\right)_{\ell,\ell'=0,\dots,n}}{\det\left(\langle \overline{\mathcal{O}_\ell^m} \mathcal{O}_{\ell'}^m \rangle_{S^4}^{\mathcal{N}=2}\right)_{\ell,\ell'=0,\dots,n-1}} \left(1 + \mathcal{O}\left(\mathrm{Im}\tau^{-6}\right)\right). \tag{3.5}$$

Consider now the $n = 0$ case. Let $\widetilde{\Theta}_0^m$ be the operator obtained by performing a GS procedure on the sphere only with operators of the same dimensions, that is (assuming $m$ even for simplicity)

$$
\widetilde{\Theta}_0^m = \frac{\det\left( \begin{array}{c} \left(\langle \overline{\phi_2^{3m-3n}\phi_3^{2n}}\phi_2^{3m-3n'}\phi_3^{2n'}\rangle_{S^4}^{\mathcal{N}=2}\right)_{\substack{n=0,\dots,m,\\ n'=0,\dots,m-1}} \\ \left(\phi_2^{3m-3n}\phi_3^{2n}\right)_{n=0,\dots,m} \end{array} \right)}{\det\left( \langle \overline{\phi_2^{3m-3n}\phi_3^{2n}}\phi_2^{3m-3n'}\phi_3^{2n'}\rangle_{S^4}^{\mathcal{N}=2}\right)_{\substack{n=0,\dots,m-1,\\ n'=0,\dots,m-1}}} \, . \tag{3.6}
$$

Expanding $\widetilde{\Theta}_0^m$ in the basis $\mathcal{O}_n^m$ one finds

$$
\widetilde{\Theta}_0^m = \mathcal{O}_0^{(m)} + \sum_{\ell=1}^{\frac{m}{2}} \mathcal{O}\left(\frac{1}{\mathrm{Im}\tau^3}\right)\mathcal{O}_{n+3\ell}^{(m-2\ell)} \, , \tag{3.7}
$$

that gives

$$
\langle \overline{\widetilde{\Theta}_0^m}\widetilde{\Theta}_0^m\rangle_{S^4}^{\mathcal{N}=2} = \langle \overline{\mathcal{O}_0^m}\mathcal{O}_0^m\rangle_{S^4}^{\mathcal{N}=2}\left(1 + \mathcal{O}\left(\mathrm{Im}\tau^{-6}\right)\right) \, . \tag{3.8}
$$

Therefore we can compute $(G_0^m)^{\mathcal{N}=2}$ by performing a GS procedure that only involves operators of the same dimension in $\mathcal{N} = 2$, that is

$$
(G_n^m)^{\mathcal{N}=2} = \frac{\det\left( \langle \overline{\phi_2^{3m-3n}\phi_3^{2n}}\phi_2^{3m-3n'}\phi_3^{2n'}\rangle_{S^4}^{\mathcal{N}=2}\right)_{\substack{n=0,\dots,m,\\ n'=0,\dots,m}}}{\det\left( \langle \overline{\phi_2^{3m-3n}\phi_3^{2n}}\phi_2^{3m-3n'}\phi_3^{2n'}\rangle_{S^4}^{\mathcal{N}=2}\right)_{\substack{n=0,\dots,m-1,\\ n'=0,\dots,m-1}}}\left(1 + \mathcal{O}\left(\mathrm{Im}\tau^{-6}\right)\right) \, . \tag{3.9}
$$

Defining

$$
\widetilde{\Theta}_n^m = \phi_2^n\widetilde{\Theta}_0^m \, , \tag{3.10}
$$

by taking derivatives with respect to $\tau$ of (3.8) we find that in general

$$
\langle \overline{\widetilde{\Theta}_a^m}\widetilde{\Theta}_b^\ell\rangle_{S^4}^{\mathcal{N}=2} = \delta_{m\ell}\langle \overline{\mathcal{O}_a^m}\mathcal{O}_b^m\rangle_{S^4}^{\mathcal{N}=2}\left(1 + \mathcal{O}\left(\mathrm{Im}\tau^{-6}\right)\right) \, . \tag{3.11}
$$

Therefore the difference between $\mathcal{N} = 2$ and $\mathcal{N} = 4$ mixing will only produce six loops effect on flat space correlators. The $(G_0^m)^{\mathcal{N}=2}$ correlators can be computed by performing a GS procedure only with operators of the same dimensions on the sphere, and mixing between $\Theta_n^m$ families with different $m$'s will only affect flat space correlators from 6 loops on. In particular we have

$$
(G_n^m)^{\mathcal{N}=2} = Z_{S^4}^{-1}\frac{\det_{i,j=0,\dots,n}\partial_\tau^i\partial_{\bar\tau}^j\left(Z_{S^4}(G_0^m)^{\mathcal{N}=2}\right)}{\det_{i,j=0,\dots,n-1}\partial_\tau^i\partial_{\bar\tau}^j\left(Z_{S^4}(G_0^m)^{\mathcal{N}=2}\right)}\left(1 + \mathcal{O}\left(\mathrm{Im}\tau^{-6}\right)\right) \, , \tag{3.12}
$$

in complete analogy with the $\mathcal{N} = 4$ case (2.59).

## 3.2 Mixing at large charge

The $\mathcal{N} = 2$ operators $\Theta_n^m$ are characterized be the quantum numbers $m, n$ that parametrize their scaling dimension and $R$ charge as

$$R = 2\Delta = 2(3m + 2n) \,. \tag{3.13}$$

We now study the $\mathcal{N} = 2$ correlator $G_n^m$ and the fate of the $\Theta_n^m$ mixing pattern in the double scaling large charge limit

$$R \to \infty \,, \ \mathrm{Im}\tau \to \infty \,, \ \frac{R}{\mathrm{Im}\tau} \ \text{fixed.} \tag{3.14}$$

Since the $R$ charge is controlled by the two independent parameters $m, n$, we will consider three different cases:

$$m \to \infty \,, \ \mathrm{Im}\tau \to \infty \,, \ \lambda = \frac{m}{2\pi\mathrm{Im}\tau} \,, \ n \ \text{fixed,} \tag{3.15}$$

$$m, n \to \infty \,, \ \mathrm{Im}\tau \to \infty \,, \ \lambda = \frac{m}{2\pi\mathrm{Im}\tau} \,, \ \kappa = \frac{n}{2\pi\mathrm{Im}\tau} \ \text{fixed,} \tag{3.16}$$

$$n \to \infty \,, \ \mathrm{Im}\tau \to \infty \,, \ \kappa = \frac{n}{2\pi\mathrm{Im}\tau} \,, \ m \ \text{fixed.} \tag{3.17}$$

We can check order by order in the loop expansion that all these limits exist, that is[10]

$$\frac{(G_n^m)^{\mathcal{N}=2}}{(G_n^m)^{\mathcal{N}=4}} = f(\lambda, \kappa) + \frac{g(\lambda, \kappa)}{m} + \frac{h(\lambda, \kappa)}{n} + \mathcal{O}\left(\frac{1}{m^2}\right) + \mathcal{O}\left(\frac{1}{n^2}\right). \tag{3.18}$$

Note in convention: if two quantities differ by subleading correction in the double scaling limits above, that is

$$A(m, n, \mathrm{Im}\tau) = B(m, n, \mathrm{Im}\tau) + \sum_{i \geq 1} \frac{g_i(\lambda, \kappa)}{m^i} + \frac{h_i(\lambda, \kappa)}{n^i} \tag{3.19}$$

we simply note

$$A(m, n, \mathrm{Im}\tau) \simeq B(m, n, \mathrm{Im}\tau) \,. \tag{3.20}$$

The aim of the following analysis will be to compare correlators of the genuine $\mathcal{N} = 2$ operators $\Theta_n^m$ with the ones of $\widetilde{\Theta}_n^m$ defined in (3.6) and (3.10). Recall that $\widetilde{\Theta}_n^m$ enjoy the same mixing pattern as their $\mathcal{N} = 4$ analogue, therefore in every regime in which

$$G_n^m \simeq \langle \overline{\widetilde{\Theta}_n^m} \widetilde{\Theta}_n^m \rangle_{S^4}^{\mathcal{N}=2} \tag{3.21}$$

the $\mathcal{N} = 2$ mixing pattern mimics the $\mathcal{N} = 4$ one.

Let us start from the first case (3.15). The $n$ dependence is subleading in $m$, but we will fix $n = 0$ for simplicity. In the previous section we have shown that up to six loops

---

[10]Note that in the full scaling limit (3.16) $h$ can be reabsorbed into $g$.

$(G_0^m)^{\mathcal{N}=2}$ can be computed by a GS procedure that only involves operators of the same scaling dimensions on the sphere, as it happens for $(G_0^m)^{\mathcal{N}=4}$. We now argue that in the limit (3.15) equation (3.21) holds true. The first nontrivial term to check is the six loop term that comes from multiplying together three loop terms in the determinant (3.4). It will be proportional to $\zeta(5)^2$ since three loop terms are always proportional to $\zeta(5)$. The strategy to compute these terms is the following: we compute the full correlatots $(G_0^m)^{\mathcal{N}=2}$ for a sufficiently large number of cases and we find the interpolating sequence using the `FindSequenceFunction` Mathematica command. Restricting to terms that only contain $\zeta(5)$ factors we find at six loops

$$
\begin{aligned}
\left.\frac{(G_0^m)^{\mathcal{N}=2}}{(G_0^m)^{\mathcal{N}=4}}\right|_{\zeta(5)} &= 1 + \frac{5}{6}\frac{18m^3 + 90m^2 + 148m - 5}{\mathrm{Im}\tau^3\pi^3}\zeta(5)+ \\
&+ \frac{24}{144}\frac{648m^6 + 8424m^5 + 45000m^4 + 120240m^3 + \frac{332659}{2}m^2 + \frac{221817}{2}m - 5120}{\mathrm{Im}\tau^6\pi^6}\zeta(5)^2 \quad (3.22)\\
&+ \mathcal{O}\left(\frac{1}{\mathrm{Im}\tau^7}\right),
\end{aligned}
$$

which in the double scaling limit (3.15) gives at 6 loops

$$
\begin{aligned}
\left.\frac{(G_0^m)^{\mathcal{N}=2}}{(G_0^m)^{\mathcal{N}=4}}\right|_{\zeta(5)} &= \left(1 + 120\zeta(5)\lambda^3 + 7200\zeta(5)^2\lambda^6\right) + \frac{1}{m}\left(600\zeta(5)\lambda^3 + 93600\zeta(5)^2\lambda^6\right) \\
&\quad + \mathcal{O}\left(\frac{1}{m^2}\right).
\end{aligned} \quad (3.23)
$$

Let us now perform GS only on operator of the same dimension, and compute the two point function $\langle\overline{\widetilde{\Theta}_0^m}\widetilde{\Theta}_0^m\rangle$ of the operators defined in (3.6) We find

$$
\begin{aligned}
\left.\frac{\langle\overline{\widetilde{\Theta}_0^m}\widetilde{\Theta}_0^m\rangle}{(G_0^m)^{\mathcal{N}=4}}\right|_{\zeta(5)} &= 1 + \frac{5}{6}\frac{18m^3 + 90m^2 + 148m - 5}{\mathrm{Im}\tau^3\pi^3}\zeta(5)+ \\
&+ \frac{24}{144}\frac{648m^6 + \frac{135027}{16}m^5 + \frac{180567}{4}m^4 + \frac{1932255}{16}m^3 + \frac{1335685}{8}m^2 + \frac{222281}{2}m - 5120}{\mathrm{Im}\tau^6\pi^6}\zeta(5)^2 \quad (3.24)\\
&+ \mathcal{O}\left(\frac{1}{\mathrm{Im}\tau^7}\right),
\end{aligned}
$$

that in the double scaling limit (3.15) gives

$$
\begin{aligned}
\left.\frac{\langle\overline{\widetilde{\Theta}_0^m}\widetilde{\Theta}_0^m\rangle}{(G_0^m)^{\mathcal{N}=4}}\right|_{\zeta(5)} &= \left(1 + 120\zeta(5)\lambda^3 + 7200\zeta(5)^2\lambda^6\right) + \frac{1}{m}\left(600\zeta(5)\lambda^3 + \frac{375075}{4}\zeta(5)^2\lambda^6\right) \\
&\quad + \mathcal{O}\left(\frac{1}{m^2}\right).
\end{aligned} \quad (3.25)
$$

The difference between (3.23) and (3.25) starts at six loops and is subleading in $m$ in the double scaling limit. Even though we don't have an all loop argument to support this claim,

we are led to conjecture that the $\mathcal{N} = 2$ operators $\Theta_0^m$ follow the same mixing pattern of the $\mathcal{N} = 4$ $\mathcal{O}_0^{(m)}$ at leading order in $m$ and all orders in $\lambda$ in the double scaling limit (3.15). At leading order in $m$ we have

$$\log \left. \frac{(G_0^m)^{\mathcal{N}=2}}{(G_0^m)^{\mathcal{N}=4}} \right|_{\zeta(5)} \simeq 120\zeta(5)\lambda^3 + \mathcal{O}(m^{-1}). \tag{3.26}$$

The dependence on $\zeta(5)$ exponentiates, meaning that $\zeta(5)$ only appears linearly in the logarithm. The same will happen for all $\zeta-$numbers. The same behavior has been observed in $SU(2)$ correlator in the double scaling limit [17, 20–22].

Let us discuss what happens if we turn on $n$ as in (3.16). We now argue that at leading order in $m, n$ and at all orders in $\lambda$ and $\kappa$ the $\mathcal{N} = 2$ operators $\Theta_n^m$ enjoy the same mixing patterns as their $\mathcal{N} = 4$ counterpart and (3.21) is satisfied.

For concreteness let us specify to the case $m = 2\ell$, $n = 3\ell$, which gives $\lambda = \frac{2}{3}\kappa$. Following the same strategy as before we find

$$\left. \frac{\langle \overline{\widetilde{\Theta}_{3\ell}^{2\ell}} \widetilde{\Theta}_{3\ell}^{2\ell} \rangle}{\left(G_{3\ell}^{2\ell}\right)^{\mathcal{N}=4}} \right|_{\zeta(5)} = 1 + \frac{5}{6} \frac{2448\ell^3 + 2304\ell^2 + 736\ell - 5}{\mathrm{Im}\tau^3 \pi^3} \zeta(5) + \frac{25}{576} \frac{1}{\mathrm{Im}\tau^6 \pi^6} \zeta(5)^2 \times$$

$$\frac{1}{36\ell^3 + 133\ell^2 + 191\ell + 84} \left( 1725898752\ell^9 + 11544952842\ell^8 + 32396774337\ell^7 \right.$$

$$+ 49752465327\ell^6 + 45784538253\ell^5 + 26031617883\ell^4 + 9103663738\ell^3 + 1866708868\ell^2 +$$

$$\left. + 183208400\ell - 1720320 \right), \tag{3.27}$$

which in the double scaling limit simplifies to

$$\left. \frac{\langle \overline{\widetilde{\Theta}_{3\ell}^{2\ell}} \widetilde{\Theta}_{3\ell}^{2\ell} \rangle}{\left(G_{3\ell}^{2\ell}\right)^{\mathcal{N}=4}} \right|_{\zeta(5)} = \left( 1 + 2040\zeta(5)\lambda^3 + 2080800\zeta(5)^2\lambda^6 \right) + \\ + \frac{1}{\ell} \left( 1920\zeta(5)\lambda^3 + \frac{716258925}{128}\zeta(5)^2\lambda^6 \right). \tag{3.28}$$

Although we did not find a closed form expression for $\left(G_{3\ell}^{2\ell}\right)^{\mathcal{N}=2}$ as a function of $\ell$, in figure 1 we provide numerical evidence that (3.21) is satisfied in this case as well. Moreover

$$\log \left. \frac{\langle \overline{\widetilde{\Theta}_{3\ell}^{2\ell}} \widetilde{\Theta}_{3\ell}^{2\ell} \rangle}{\left(G_{3\ell}^{2\ell}\right)^{\mathcal{N}=4}} \right|_{\zeta(5)} = 2040\zeta(5)\lambda^3 + \mathcal{O}(\ell^{-1}), \tag{3.29}$$

that is the dependence on $\zeta(5)$ exponentiates here as well,

We finally consider the last case (3.17) with $m = 0$. Proceeding as before we find

$$\left. \frac{(G_n^0)^{\mathcal{N}=2}}{(G_n^0)^{\mathcal{N}=4}} \right|_{\zeta(5)} = 1 + \frac{425}{36\pi^3 \mathrm{Im}\tau^3} \left( 11n + 6n^2 + n^3 \right) \zeta(5) +$$

$$+ \frac{125\zeta(5)^2}{2592\pi^6 \mathrm{Im}\tau^6} \left( 464154n + 668553n^2 + 448038n^3 + 149665n^4 + 23970n^6 + 1450n^6 \right), \tag{3.30}$$

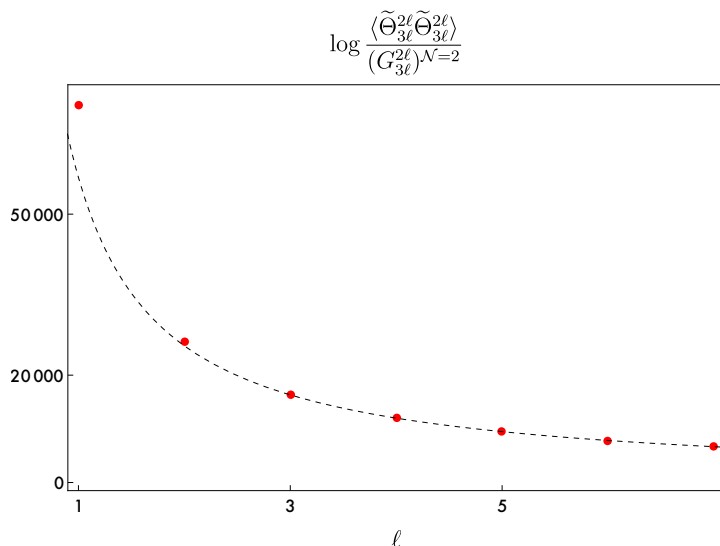

**Figure 1**. Red dots are coefficients of $\zeta(5)^2$ in $\log \frac{\langle \widetilde{\Theta}_{3\ell}^{2\ell} \widetilde{\Theta}_{3\ell}^{2\ell} \rangle}{(G_{3\ell}^{2\ell})^{\mathcal{N}=2}}$ for various values of $\ell$. The dashed line is a numerical fit that confirms the $1/\ell$ behavior as $\ell$ gets larger. As $\ell \to \infty$ equation (3.21) gets satisfied.

that in the double scaling limit gives

$$\frac{(G_n^0)^{\mathcal{N}=2}}{(G_n^0)^{\mathcal{N}=4}}\bigg|_{\zeta(5)} = \left(1 + \frac{850}{9}\zeta(5)\kappa^3 + \frac{362500}{81}\zeta(5)^2\lambda^6\right) + \frac{1}{n}\left(\frac{1700}{3}\zeta(5)\lambda^3 + \frac{1997500}{27}\zeta(5)^2\lambda^6\right). \tag{3.31}$$

This has to be compared with

$$\frac{\langle \overline{\widetilde{\Theta}_n^0} \widetilde{\Theta}_n^0 \rangle}{(G_n^0)^{\mathcal{N}=4}}\bigg|_{\zeta(5)} = 1 + \frac{425}{36\pi^3 \mathrm{Im}\tau^3}\left(11n + 6n^2 + n^3\right)\zeta(5) +$$

$$+ \frac{29\zeta(5)^2}{10368\pi^6 \mathrm{Im}\tau^6}\left(9271248n + 13386304n^2 + 8959230n^3 + 2990665n^4 + 479622n^3 + 29131n^6\right) \tag{3.32}$$

which in the double scaling limit gives

$$\frac{\langle \overline{\widetilde{\Theta}_n^0} \widetilde{\Theta}_n^0 \rangle}{(G_n^0)^{\mathcal{N}=4}}\bigg|_{\zeta(5)} = \left(1 + \frac{850}{9}\zeta(5)\kappa^3 + \frac{728275}{162}\zeta(5)^2\lambda^6\right) + \frac{1}{n}\left(\frac{1700}{3}\zeta(5)\kappa^3 + \frac{1998425}{27}\zeta(5)^2\kappa^6\right). \tag{3.33}$$

Equation (3.33) is in agreement with [20, 23, 38]. Note however that it does not agree with (3.31) at leading order in the double scaling limit (3.17). Mixing of maximal trace operators $\phi_2^n$ with operators that contain an order $\mathcal{O}(n^0)$ number of $\phi_3'$s is not subleading in the double scaling limit (3.17). This means that the matrix model proposed in [23] computes (3.33) but it does not compute the correct flat space correlators in (3.31). In addition, note that when the number of $\phi_3$ inserted gets of order $\mathcal{O}(n)$ mixing between families $\widetilde{\Theta}_n^m$ with different $m'$s gets again subleading in the double scaling limit.

# 4  Matrix models for extremal correlators: $\mathcal{N} = 4$ SYM

It was observed in [17] that extremal correlators in $SU(2)$ $\mathcal{N} = 4$ SYM can be expressed as a Wishart matrix model where the size of the matrices is related to the number of operator insertions. The key tool used in this analysis is the Andréief–Gram–Hein identity which can be summarized as follows. Let

$$\mu_n = \int_U x^n w(x)\mathrm{d}x, \quad D_n = \det(\mu_{i+j})_{i,j=0}^{n-1} . \tag{4.1}$$

Then $D_n$ has the following matrix model representation

$$D_n = \frac{1}{n!} \int_{U^n} \mathrm{d}^n x \prod_{1 \le i,j \le n} (x_i - x_j)^2 \prod_{i=1}^n w(x_i). \tag{4.2}$$

Our claim is that, if we consider $SU(N)$ $\mathcal{N} = 4$ SYM, it is still possible to represent extremal correlators (2.39) using matrix models. However, if $N > 2$ it becomes necessary to employ multiple matrix models with possible multi-cut phases starting at $N = 4$. For instance in the $SU(3)$ case we find a description which involve a Wishart and a Jacobi matrix model, as detailed below.

In a Wishart matrix model we integrate over Hermitian, non-negative $n \times n$ matrices $W$. After some manipulations, this integral can be reduced to a multidimensional integral over the eigenvalues $y_i \in \mathbb{R}_+$ of $W$. In this section, the relevant Wishart model is given by

$$Z^{(m)}(n) = \frac{1}{n!} \int_{\mathbb{R}_+^n} \mathrm{d}^n y \prod_{i<j} (y_i - y_j)^2 \prod_{i=1}^n \mathrm{e}^{-2\pi \mathrm{Im}\tau y_i} y_i^{3m+3} , \tag{4.3}$$

For more details and references, see Appendix A.

On the other hand, in a Jacobi matrix model, we integrate over $n \times n$ Hermitian matrices $J$, which are non-negative and bounded above by the identity matrix. This matrix integral can be reduced to a multidimensional integral over the eigenvalues $x_i \in [0,1]$ of $J$. In this section, the relevant Jacobi model is given by

$$Z_{\mathrm{J}}^{(\sigma_m)}(n) = \frac{1}{n!} \int_{[0,1]^n} \mathrm{d}^n x \prod_{1 \le i<j \le n} (x_i - x_j)^2 \prod_{i=1}^n \sqrt{\frac{1}{x_i} - 1}\, x_i^{\sigma_m}, \tag{4.4}$$

where $\sigma_m = m \mod 2$. For more details and references on Jacobi model, see Appendix A.

Let us first consider the correlators (2.39) for $n = 0$, that is

$$(G_0^m)^{\mathcal{N}=4} = \left\langle \mathcal{O}_0^m \overline{\mathcal{O}_0^m} \right\rangle_{\mathbb{R}^4} . \tag{4.5}$$

As explained around (2.58) we can write

$$(G_0^m)^{\mathcal{N}=4} = \frac{\det M_m^{(\lfloor m/2 \rfloor + 1)}}{\det M_m^{(\lfloor m/2 \rfloor)}} \tag{4.6}$$

where $M_m^{(\alpha)}$ is defined in (2.44). Hence, by using (4.1) and (4.2), we immediately get

$$(G_0^m)^{\mathcal{N}=4} = \left( \frac{\sqrt{3}}{Z_{S^4}} \frac{6^{-m-1}}{(2\pi \mathrm{Im}(\tau))^{4+3m}} \Gamma(4+3m) \right) \frac{Z_{\mathrm{J}}^{(\sigma_m)}(\lfloor \frac{m}{2} \rfloor + 1)}{Z_{\mathrm{J}}^{(\sigma_m)}(\lfloor \frac{m}{2} \rfloor)} \tag{4.7}$$

where we used $2\lfloor \frac{m}{2} \rfloor + \sigma_m = m$ and $Z_{\mathrm{J}}^{(\sigma_m)}(n)$ is the Jacobi matrix model in (4.4). This matrix model is exactly solvable, see (A.21), and it gives

$$(G_0^m)^{\mathcal{N}=4} = \frac{1}{(2\pi \mathrm{Im}(\tau))^{3m}} \frac{3^{-m-1} 8^{-m}}{2} \Gamma(4+3m) . \tag{4.8}$$

Let us now consider the generic correlator with $m, n \neq 0$, that is

$$\langle \mathcal{O}_{n'}^m \mathcal{O}_n^m \rangle_{\mathbb{R}^4}^{\mathcal{N}=4}, \qquad \mathcal{O}_n^m = \phi_2^n \mathcal{O}_0^m . \tag{4.9}$$

Since adding $\phi_2$ still preserve orthogonality between the $\mathcal{O}_0^m$, see (2.36), we can obtain the correlatos (4.9) simply by performing one more time the GS on the indices $n, n'$. We get

$$\langle \mathcal{O}_n^m \mathcal{O}_{n'}^{m'} \rangle_{\mathbb{R}^4}^{\mathcal{N}=4} = \delta_{n,n'} \delta_{m,m'} \frac{\det K_m^{(n+1)}}{\det K_m^{(n)}} \tag{4.10}$$

where $K_m^{(n)}$ is the $n \times n$ matrix whose elements are

$$K_{i,j} = \frac{(-2\pi)^{-i-j}}{Z_{S^4}} \partial_{\mathrm{Im}\tau}^{i+j} \left( Z_{S^4}(\tau, 0) \frac{\det M_m^{(\lfloor m/2 \rfloor + 1)}}{\det M_m^{(\lfloor m/2 \rfloor)}} \right) \quad i, j = 0, \ldots, n-1 \tag{4.11}$$

see (2.37) and also (2.59). Since the $\tau$−dependence is tree level exact we have

$$K_{i,j} = \frac{\sqrt{3} 6^{-\sigma_m - 2\lfloor \frac{m}{2} \rfloor - 1}}{Z_S^4} \frac{Z_{\mathrm{J}}^{(\sigma_m)}(1 + \lfloor \frac{m}{2} \rfloor)}{Z_{\mathrm{J}}^{(\sigma_m)}(\lfloor \frac{m}{2} \rfloor)} \left( \int_0^\infty \exp(-2\pi \mathrm{Im}\tau y) y^{i+j} y^{3m+3} \, dy \right) . \tag{4.12}$$

Hence by using the identities (4.1) and (4.2) we obtain a second matrix model, that is

$$\det K^{(n)} = \left( \frac{\sqrt{3} 6^{-\sigma_m - 2\lfloor \frac{m}{2} \rfloor - 1}}{Z_{S^4}} \frac{Z_{\mathrm{J}}^{(\sigma_m)}(1 + \lfloor \frac{m}{2} \rfloor)}{Z_{\mathrm{J}}^{(\sigma_m)}(\lfloor \frac{m}{2} \rfloor)} \right)^n Z^{(m)}(n) \tag{4.13}$$

where $Z^{(m)}(n)$ is the Wishart-Laguerre matrix model given in (4.3). In summary we obtain

$$(G_n^m)^{\mathcal{N}=4} = \langle \mathcal{O}_n^m \mathcal{O}_n^m \rangle_{\mathbb{R}^4}^{\mathcal{N}=4} = \frac{6^{-m-1} \sqrt{3}}{Z_{S^4}} \frac{Z_{\mathrm{J}}^{(\sigma_m)}(1 + \lfloor \frac{m}{2} \rfloor)}{Z_{\mathrm{J}}^{(\sigma_m)}(\lfloor \frac{m}{2} \rfloor)} \frac{Z^{(m)}(n+1)}{Z^{(m)}(n)} , \tag{4.14}$$

where the $\tau, \bar{\tau}$ dependence enters via $Z^{(m)}(n)$, see (4.3).

By further using (A.4), (A.6) and (A.21), we obtain the more explicit expression

$$(G_n^m)^{\mathcal{N}=4} = \langle \mathcal{O}_n^m \mathcal{O}_n^m \rangle_{\mathbb{R}^4}^{\mathcal{N}=4} = \frac{n! \Gamma(3m+n+4)}{3^{m+1} 2^{6m+2n+1} \pi^{3m+2n} \mathrm{Im}\tau^{3m+2n}} \tag{4.15}$$

as well as

$$\frac{\langle \mathcal{O}_n^m \mathcal{O}_n^m \rangle_{\mathbb{R}^4}^{\mathcal{N}=4}}{\langle \mathcal{O}_0^m \mathcal{O}_0^m \rangle_{\mathbb{R}^4}^{\mathcal{N}=4}} = \left( \frac{1}{2\pi \mathrm{Im}\tau} \right)^{2n} \Gamma(1+n)(3m+4)_n \tag{4.16}$$

in agreement with (2.40). If we take $m, n$ large such that $\frac{n}{m} = \beta$ is fixed, we get

$$\begin{aligned}
\log(G_n^m)^{\mathcal{N}=4} = {}& 3(\beta+2)m\log(m) \\
& - m\Big( (2\beta+3)(\log(2\pi\mathrm{Im}\tau)+1) + \log(24) - \beta\log(\beta) - (\beta+3)\log(\beta+3) \Big) \\
& + \frac{15\log(m)}{2} + 4\log(\beta) + \log\left(\frac{\pi}{3}\right) + \frac{74\beta+3}{12\beta^2 m + 36\beta m} + \mathcal{O}(m^{-2}) \, .
\end{aligned} \tag{4.17}$$

It would be interesting to interpret this expansion from the point of view of a large charge EFT, but we leave that for future work.

Let us conclude this section by noting that, parallel to the rank 1 case, the expression for the correlators in terms of matrix models (4.14) holds also at finite $n, m$. Moreover, the two matrix models involved in the correlators (4.14) "factorize" (even tough the dependence on $m$ enters in both of them). This will not hold true for $\mathcal{N} = 2$ SQCD or for the integrated correlators in $\mathcal{N} = 4$ SYM, as we discuss below.

## 5 Matrix models for extremal correlators: $\mathcal{N} = 2$ SQCD

We are interested in computing extremal correlators of $\mathcal{N} = 2$, $N_f = 2N$, $SU(N)$ SQCD in the regime where we have a large number of insertions, i.e. a large R-charge.

We first recall that in the rank 1 example, the correlators (2.30) of $\mathcal{N} = 2$ SQCD are, in the large $R$ regime, equivalent to expectation values of $Z_{\mathrm{G}}$ within the $\mathcal{N} = 4$ SYM matrix model, see [17]. This prompts the question of whether this structure also extends to higher ranks. In this section, we show that indeed, this persists at higher ranks, at least at leading order in the 't Hooft expansion[11]. In the rest of the section we will focus on the example of $SU(3)$ SQCD with $N_f = 6$ flavors but we expect an analogous behaviour for $SU(N)$ as well.

### 5.1 The $G_0^m$ correlators

Let us start by studying at the correlators (2.72) when $n = 0$, that is

$$(G_0^m)^{\mathcal{N}=2} = \langle \Theta_0^m \Theta_0^m \rangle_{\mathbb{R}^4}^{\mathcal{N}=2}, \tag{5.1}$$

in the following double scaling limit

$$m, \mathrm{Im}\tau \to \infty \quad \text{s.t.} \quad \lambda = \frac{m}{2\pi\mathrm{Im}\tau}, \quad \text{fixed} \, . \tag{5.2}$$

---

[11]It may holds beyond this, but in this paper we focus only on such regime.

As illustrated in section 3, the mixing structure in this limit simplify as mixing with operators of lower dimension is subleading. In practice this means that, in the scaling limit (5.2) we have

$$(G_0^m)^{\mathcal{N}=2} \simeq \frac{\det \mathtt{M}_m^{(\lfloor m/2 \rfloor+1)}}{\det \mathtt{M}_m^{(\lfloor m/2 \rfloor)}} \tag{5.3}$$

where $\mathtt{M}_\mathtt{m}^{(\mathtt{k})}$ is the $k \times k$ matrix whose elements are

$$\mathtt{M}_{i,j} = \left\langle \left( \frac{\phi_3^2}{\phi_2^3} \right)^i \phi_3^{\sigma_m} \phi_2^{3\lfloor m/2 \rfloor} \overline{\left( \frac{\phi_3^2}{\phi_2^3} \right)^j \phi_3^{\sigma_m} \phi_2^{3\lfloor m/2 \rfloor}} \right\rangle_{S^4}^{\mathcal{N}=2}, \quad i,j = 0,\dots,k-1 . \tag{5.4}$$

This is analogous to (2.44) except that the vev is now taken in $\mathcal{N}=2$ SQCD. We now show that (5.3) is in fact a ratio of matrix models expectation value.

Let us define

$$\mathtt{f}_m(x, \mathrm{Im}(\tau)) = \int_{\mathbb{R}_+} \mathrm{d}y \; y^{3+3m} \mathrm{e}^{-2\pi \mathrm{Im}\tau y} \mathcal{Z}_\mathrm{G}(x,y) . \tag{5.5}$$

with

$$\mathcal{Z}_\mathrm{G}(x,y) = Z_\mathrm{G}(a_1(x,y), a_2(x,y)) \tag{5.6}$$

where we are using the change of variable (2.41) and $Z_\mathrm{G}(a_1, a_2)$ is given in (2.64). By using the Andréief-Gram-Hein identity (4.2), it is easy to see that

$$\det \mathtt{M}_m^{(k)} = \frac{Z_\mathrm{J}^{(\sigma_m)}(k)\sqrt{3}^k}{6^{k^2+k\sigma_m}(Z_{S^4})^k} \langle \mathtt{f}_m(x, \mathrm{Im}\tau) \rangle_\mathrm{J}^{(k)}, \tag{5.7}$$

where the expectation value is w.r.t. the Jacobi model (4.4), namely

$$\langle \mathtt{f}_m(x, \mathrm{Im}\tau)) \rangle_\mathrm{J}^{(k)} = \frac{\frac{1}{k!} \int_{[0,1]^n} \mathrm{d}^k x_i \prod_{i<j}(x_i - x_j)^2 (x_i-1)^{1/2} x_i^{\frac{1}{2}+\sigma_m} \mathtt{f}_m(x, \mathrm{Im}\tau)}{Z_\mathrm{J}^{(\sigma_m)}(k)} . \tag{5.8}$$

This already show that (5.3) is indeed a ratio of expectation values in the $\mathcal{N}=4$ Jacobi model. However, let us massage this expression a but more. It easy to see [12] that in the 't Hooft limit (5.23) we have

$$\mathtt{f}_m(x, \mathrm{Im}\tau)) = \int_{\mathbb{R}^+} \mathrm{d}y \mathrm{e}^{-2\pi \mathrm{Im}\tau y} y^{3m+3} \mathcal{Z}_\mathrm{G}(x,y) \simeq \frac{\sqrt{2\pi}\mathrm{e}^{-3m}(3\lambda)^{3m+4}}{\sqrt{3m}} \mathcal{Z}_\mathrm{G}(x,3\lambda). \tag{5.9}$$

Note that the second equality in (5.9) also holds if we replace $\mathcal{Z}_\mathrm{G}(x,y)$ by another function which admit a Laurent expansion around $y = 0$.

By combining (5.7) and (5.9) we arrive at the following simple formula

$$\frac{(G_0^m)^{\mathcal{N}=2}}{(G_0^m)^{\mathcal{N}=4}} \simeq \frac{\langle \mathcal{Z}_G(x,3\lambda) \rangle_J^{(\lfloor \frac{m}{2} \rfloor+1)}}{\langle \mathcal{Z}_G(x,3\lambda) \rangle_J^{(\lfloor \frac{m}{2} \rfloor)}} , \tag{5.10}$$

---

[12]Either by doing a saddle point or by using the Laurent expansion of the Barnes G-funtions

where the expectation value of $\mathcal{Z}_G(x, y)$ is taken in the $\mathcal{N} = 4$ Jacobi matrix model (4.4), parallel to (5.8) and, $(G_0^m)^{\mathcal{N}=4}$ is given in (4.7). From now on it will be convenient to use

$$\Delta G_0^m = \frac{(G_0^m)^{\mathcal{N}=2}}{(G_0^m)^{\mathcal{N}=4}} \ . \tag{5.11}$$

One advantage of writing correlators as matrix models is that one can easily extract the large $m$ expansion. This is done as follows. First we note that $\mathcal{Z}_G(x, 3\lambda)$ is subleading w.r.t. the eigenvalue repulsion term in the Jacobi matrix model (4.4). Therefore

$$\log \langle \mathcal{Z}_G(x, 3\lambda) \rangle_J^{(k)} = k \int_0^1 \mathrm{d}x \sigma_J(x) \log(\mathcal{Z}_G(x, 3\lambda)) + \mathcal{O}(k^0), \tag{5.12}$$

where $\sigma_J(x)$ is the Jacobi density

$$\sigma_J(x) = \frac{1}{\pi\sqrt{x(1-x)}}, \quad x \in [0, 1], \tag{5.13}$$

see also Appendix A. Hence we simply have

$$\log \Delta G_0^m \simeq \int_0^1 \mathrm{d}x \sigma_J(x) \log(\mathcal{Z}_G(x, 3\lambda)) \ , \tag{5.14}$$

where $\simeq$ has the same meaning as in (3.20). If we further expand (5.14) at weak coupling $\lambda$ we find the following all order expansion

$$\log \Delta G_0^m \simeq \sum_{k=1}^\infty \frac{9(-1)^k 2^{k+2}\left(3^k - 1\right)\lambda^{k+1}\zeta(2k+1)\Gamma\left(k + \frac{3}{2}\right)}{\sqrt{\pi}(k+1)^2 k!}. \tag{5.15}$$

We note that (5.15) naturally split in two series:

$$\begin{aligned}
&\sum_{k=1}^\infty \frac{9(-1)^k 2^{k+2}3^k \lambda^{k+1}\zeta(2k+1)\Gamma\left(k + \frac{3}{2}\right)}{\sqrt{\pi}(k+1)^2 k!} \\
&\sum_{k=1}^\infty \frac{9(-1)^k 2^{k+2}\lambda^{k+1}\zeta(2k+1)\Gamma\left(k + \frac{3}{2}\right)}{\sqrt{\pi}(k+1)^2 k!} \ .
\end{aligned} \tag{5.16}$$

The radii of convergence of the first and second series are $\lambda_c^{(1)} = 1/6$ and $\lambda_c^{(1)} = 1/2$, respectively. It would be interesting to understand what this means physically but we leave this for further investigation.

The strong coupling expansion of (5.14) can also be computed in a straightforward way. Parallel to [17] we first write (5.15) as an integral over Bessel J function:

$$\log \Delta G_0^m \simeq \int_0^\infty \frac{6e^t}{t(e^t - 1)^2}\left(6J_0(t\sqrt{2\lambda}) - 2J_0(t\sqrt{6\lambda}) - 4\right)\mathrm{d}t \ . \tag{5.17}$$

By expanding at large $\lambda$, this leads to the following strong coupling expansion:

$$\log \Delta G_0^m \simeq -2 - 9\lambda \log 3 - \frac{1}{2} \log 12 + 24 \log A + \log \lambda - \mathcal{C}_{np}(\sqrt{6\lambda}) + 3\,\mathcal{C}_{np}(\sqrt{2\lambda})\,. \quad (5.18)$$

with

$$\mathcal{C}_{np}(\sqrt{\lambda}) = \sum_{n \geq 1} \frac{6}{n^2 \pi^2} \Big( K_0(2n\pi\sqrt{\lambda}) + 2n\pi\sqrt{\lambda} K_1(2n\pi\sqrt{\lambda}) \Big)$$

$$= \mathrm{e}^{-2\pi\sqrt{\lambda}} \left( \frac{6\sqrt[4]{\lambda}}{\pi} + \frac{33\sqrt[4]{\frac{1}{\lambda}}}{8\pi^2} - \frac{93\left(\frac{1}{\lambda}\right)^{3/4}}{256\pi^3} + \mathcal{O}\left(\left(\frac{1}{\lambda}\right)^{5/4}\right) \right), \quad (5.19)$$

where $A$ is the Glaisher constant and $K_i$ are the modified Bessel functions of second kind. In particular $\mathcal{C}_{np} \sim \mathcal{O}(\mathrm{e}^{-2\pi\sqrt{\lambda}})$ is purely non-perturbative at strong coupling and we have

$$\log \Delta G_0^m \simeq -2 - 9\lambda \log 3 - \frac{1}{2} \log 12 + 24 \log A + \log \lambda +$$

$$\mathrm{e}^{-2\sqrt{2}\pi\sqrt{\lambda}} \left( \frac{18\sqrt[4]{2}\sqrt[4]{\lambda}}{\pi} + \mathcal{O}(\lambda^{-1/4}) \right) - \mathrm{e}^{-2\sqrt{6}\pi\sqrt{\lambda}} \left( \frac{6\sqrt[4]{6}\sqrt[4]{\lambda}}{\pi} + \mathcal{O}(\lambda^{-1/4}) \right). \quad (5.20)$$

This regime is structurally very similar to the rank 1 case studied in [17] where the perturbative expansion at strong coupling truncated after few terms and the non-perurbative effects are expressed as a sum of Bessel K functions. The instantons actions are integer multiple of the following two leading actions[13]

$$A_1 = 4\pi \sqrt{\frac{1}{2}}, \qquad A_2 = 4\pi \sqrt{\frac{3}{2}}. \quad (5.21)$$

Note that $A_i = 2\pi \sqrt{1/\lambda_c^{(i)}}$, where $\lambda_c^{(i)}$ are the radii of convergence of the two sums in the weak coupling expansion (5.19). Physically, we expect such non-perturbative effects to be interpreted as the worldline of BPS particles, parallel to the rank 1 case [17, 39, 40]. In particular, it seems that in this specific sector of the $SU(3)$ theory, the EFT capturing the large charge expansion may not be too much different from the one proposed in the rank 1 case [15].

We conclude the section by noting that turning on a finite $n$ would only produce subleading effects in the 't Hooft limit, since $\partial_\tau \sim m^{-1}\partial_\lambda$.

## 5.2 The $G_n^m$ correlators at $\beta = \frac{n}{m}$ fixed

Let us now consider the full correlators (2.72) where we allow both $m, n$ to be large, that is

$$(G_n^m)^{\mathcal{N}=2} = \langle \Theta_n^m \overline{\Theta_n^m} \rangle_{\mathbb{R}^4}^{\mathcal{N}=2}\,. \quad (5.22)$$

---

[13]A naïve saddle point analysis seem to indicate that in this regime we the vev on the complex scalar is such that $a_1 \sim a_2$ and these two instanton effects should be in correspondence with the mass of $W$ bosons and the hypers.

in the regime

$$m, n, \text{Im}\tau \to \infty \quad \text{s.t.} \quad \kappa = \frac{n}{2\pi\text{Im}\tau}, \quad \lambda = \frac{m}{2\pi\text{Im}\tau}, \quad \text{fixed}. \tag{5.23}$$

It is convenient to introduce

$$\beta = \frac{n}{m}. \tag{5.24}$$

As illustrated in section 3, the mixing structure in this limit simplify. In practice this means that, in the scaling limit (5.23) we have

$$(G_n^m)^{\mathcal{N}=2} \simeq \frac{\det \mathtt{K}_m^{(n+1)}}{\det \mathtt{K}_m^{(n)}} \tag{5.25}$$

where $\mathtt{K}_m^{(n)}$ is the $n \times n$ matrix whose elements are

$$\mathtt{K}_{i,j} = \frac{(-2\pi)^{-i-j}}{Z_{S^4}} \partial_{\text{Im}\tau}^{i+j} \left( Z_{S^4} \frac{\det \mathtt{M}_m^{(\lfloor m/2 \rfloor + 1)}}{\det \mathtt{M}_m^{(\lfloor m/2 \rfloor)}} \right) \quad i, j = 0, \dots, n-1 \tag{5.26}$$

and $\mathtt{M}_m^{(k)}$ is the $k \times k$ matrix whose elements are given in (5.4). Let us stress that, in view of the discussion in subsection 3.1 we also have

$$(G_n^m)^{\mathcal{N}=2}\Big|_{5 \text{ loops}} = \frac{\det \mathtt{K}_m^{(n+1)}}{\det \mathtt{K}_m^{(n)}}\Big|_{5 \text{ loops}} \tag{5.27}$$

which holds for finite values of $m$ and $n$ as well.

In the rest of the section we will show how to compute (5.25) explicitly using matrix models. Using the results of subsection 5.1 we can write

$$\log \left( Z_{S^4} \frac{\det \mathtt{M}_m^{(\lfloor \frac{m}{2} \rfloor + 1)}}{\det \mathtt{M}_m^{(\lfloor \frac{m}{2} \rfloor)}} \right) \simeq \log \left( Z_{S^4} (G_0^m)^{\mathcal{N}=4} \frac{\langle \mathcal{Z}_G(x, 3\lambda) \rangle_J^{(\lfloor \frac{m}{2} \rfloor + 1)}}{\langle \mathcal{Z}_G(x, 3\lambda) \rangle_J^{(\lfloor \frac{m}{2} \rfloor)}} \right)$$
$$\simeq \log \left( Z_{S^4} (G_0^m)^{\mathcal{N}=4} \right) + \int_0^1 dx \sigma_J(x) \log(Z_G(x, 3\lambda)). \tag{5.28}$$

It is useful to use the trick in the second equality of (5.9) again and write (5.28) as

$$\left( Z_{S^4} \frac{\det \mathtt{M}_m^{(\lfloor \frac{m}{2} \rfloor + 1)}}{\det \mathtt{M}_m^{(\lfloor \frac{m}{2} \rfloor)}} \right) \simeq \left( Z_{S^4} (G_0^m)^{\mathcal{N}=4} \right) e^{\int_0^1 dx \sigma_J(x) \log(Z_G(x, 3\lambda))}$$
$$\simeq \frac{\sqrt{3m} Z_{S^4} (G_0^m)^{\mathcal{N}=4}}{\sqrt{2\pi} e^{-3m} (3\lambda)^{3m+4}} \int_{\mathbb{R}_+} dy e^{-2\pi\text{Im}\tau y} y^{3m+3} e^{\int_0^1 dx \sigma_J(x) \log(Z_G(x,y))}$$
$$\simeq \frac{\sqrt{3}}{6^{m+1}} \frac{Z_J^{(\sigma_m)}(\lfloor \frac{m}{2} \rfloor + 1)}{Z_J^{(\sigma_m)}(\lfloor \frac{m}{2} \rfloor)} \int_{\mathbb{R}_+} dy e^{-2\pi\text{Im}\tau y} y^{3m+3} e^{\int_0^1 dx \sigma_J(x) \log(Z_G(x,y))}. \tag{5.29}$$

This leads to

$$\mathrm{K}_{i,j} \simeq \frac{\sqrt{3}}{Z_{\mathrm{S}^4}6^{m+1}} \frac{Z_{\mathrm{J}}^{(\sigma_m)}(\lfloor\frac{m}{2}\rfloor+1)}{Z_{\mathrm{J}}^{(\sigma_m)}(\lfloor\frac{m}{2}\rfloor)} \int_{\mathbb{R}_+} \mathrm{d}y \mathrm{e}^{-2\pi\mathrm{Im}\tau y} y^{3m+3+i+j} \mathrm{e}^{\int_0^1 \mathrm{d}x\sigma_{\mathrm{J}}(x)\log(Z_{\mathrm{G}}(x,y))} \tag{5.30}$$

Hence

$$\det \mathrm{K}_m^{(n)} \simeq \left(\frac{\sqrt{3}}{Z_{\mathrm{S}^4}6^{m+1}} \frac{Z_{\mathrm{J}}^{(\sigma_m)}(\lfloor\frac{m}{2}\rfloor+1)}{Z_{\mathrm{J}}^{(\sigma_m)}(\lfloor\frac{m}{2}\rfloor)}\right)^n \frac{1}{n!} \int_{\mathbb{R}_+} \mathrm{d}^n y \mathrm{e}^{-2\pi\mathrm{Im}\tau y} \prod_{i<j}(y_i-y_j)^2 \prod_{i=1}^n y_i^{3m+3} \mathrm{e}^{\int_0^{1/6} \mathrm{d}x\sigma_{\mathrm{J}}(x)\log(\mathcal{Z}_{\mathrm{G}}(x,y_i))}$$

$$\simeq \left(\frac{\sqrt{3}}{Z_{\mathrm{S}^4}6^{m+1}} \frac{Z_{\mathrm{J}}^{(\sigma_m)}(\lfloor\frac{m}{2}\rfloor+1)}{Z_{\mathrm{J}}^{(\sigma_m)}(\lfloor\frac{m}{2}\rfloor)}\right)^n Z^{(m)}(n) \mathrm{e}^{n\int_a^b \mathrm{d}z\rho_{\mathrm{MP}}(z)\int_0^1 \mathrm{d}x\sigma_{\mathrm{J}}(x)\log(\mathcal{Z}_{\mathrm{G}}(x,\kappa z))(1+\mathcal{O}(n^0))}$$

$$\tag{5.31}$$

where $\sigma_{\mathrm{J}}(x)$ is given in (5.13), $Z^{(m)}(n)$ is the Wishart-Laguerre matrix model (4.3) and $\rho_{\mathrm{MP}}(z)$ is the Marčenko-Pastur distribution

$$\begin{aligned} \rho_{\mathrm{MP}}(z) =&\frac{1}{2\pi z}\sqrt{(b-z)(z-a)}, \quad z\in[a,b], \\ a =&2+3\beta^{-1}-2\sqrt{1+3\beta^{-1}}, \\ b =&2+3\beta^{-1}+2\sqrt{1+3\beta^{-1}} \end{aligned} \tag{5.32}$$

where $\beta = \frac{n}{m}$, see also Appendix A. Hence we obtain

$$\log\Delta G_{\beta m}^m \simeq \frac{\mathrm{d}}{\mathrm{d}n}\left(n\int_a^b \mathrm{d}y\rho_{\mathrm{MP}}(y)\int_0^1 \mathrm{d}x\sigma_{\mathrm{J}}(x)\log(\mathcal{Z}_{\mathrm{G}}(x,\lambda y))\right), \tag{5.33}$$

where we use

$$\Delta G_n^m = \frac{\langle\Theta_n^m\overline{\Theta_n^m}\rangle_{\mathbb{R}^4}^{\mathcal{N}=2}}{\langle\mathcal{O}_n^m\overline{\mathcal{O}_n^m}\rangle_{\mathbb{R}^4}^{\mathcal{N}=4}} . \tag{5.34}$$

By making the $n$-dependence in $\beta$ and $\kappa$ explicit, it is easy to see that we can write (5.33) in a more explicit form, namely

$$\log\Delta G_{\beta m}^m \simeq (1+\kappa\partial_\kappa+\beta\partial_\beta)\left(\int_a^b \mathrm{d}y\rho_{\mathrm{MP}}(y)\int_0^1 \mathrm{d}x\sigma_{\mathrm{J}}(x)\log(\mathcal{Z}_{\mathrm{G}}(x,\kappa y))\right) \tag{5.35}$$

where the dependence on $\beta$ is inside $a,b$ and $\rho_{\mathrm{MP}}$, see (5.32).

The expression (5.35) is exact in the 't Hooft couplings $\kappa = \frac{n}{2\pi\mathrm{Im}\tau}$, $\lambda = \frac{m}{2\pi\mathrm{Im}\tau}$. If we expand it at weak coupling, i.e. $\lambda, \kappa$ small with $\beta = \frac{n}{m}$ fixed, we find the following all order expansion

$$\log\Delta G_{\beta m}^m = \sum_{k\geq 1} \frac{9(-1)^k}{\sqrt{\pi}} \frac{2^{2+k}(3^k-1)}{(1+k)(k+1)!}\Gamma\left(\frac{3}{2}+k\right)\zeta(2k+1)_2F_1\left(-1-k,2+k,1,-\frac{\beta}{3}\right)\lambda^{k+1} \tag{5.36}$$

The series (5.36) naturally split into two components

$$\sum_{k\geq 1}\frac{9(-1)^k}{\sqrt{\pi}}\frac{2^{2+k}3^k}{(1+k)(k+1)!}\Gamma\left(\frac{3}{2}+k\right)\zeta(2k+1)_2F_1\left(-1-k,2+k,1,-\frac{\beta}{3}\right)\lambda^{k+1}$$

$$\sum_{k\geq 1}\frac{9(-1)^k}{\sqrt{\pi}}\frac{2^{2+k}}{(1+k)(k+1)!}\Gamma\left(\frac{3}{2}+k\right)\zeta(2k+1)_2F_1\left(-1-k,2+k,1,-\frac{\beta}{3}\right)\lambda^{k+1}$$

(5.37)

These series, non-surprisingly, have a finite radius of convergence which depends on $\beta$, that is

$$\lambda_c^{(1)}(\beta)=\frac{1}{18}\left(2\beta-2\sqrt{\beta(\beta+3)}+3\right),\qquad \lambda_c^{(2)}(\beta)=3\lambda_c^{(1)}(\beta),$$

(5.38)

where we used

$$\lim_{n\to\infty}\frac{_2F_1\left(-n,n+1;1;-\frac{\beta}{3}\right)}{_2F_1\left(-n-1,n+2;1;-\frac{\beta}{3}\right)}=\frac{1}{3}\left(2\beta-2\sqrt{\beta(\beta+3)}+3\right).$$

(5.39)

Note that $\lambda_c^{(1)}(\beta)=\frac{\beta}{18}a$, where $a$ is the endpoint of the cut in the Marčenko-Pastur distribution (5.32).

To obtain the strong $\lambda$ coupling expansion it is useful to recast the sum in (5.36) into its Mellin-Barnes representation, that is

$$\log\Delta G_{\beta m}^m\simeq\frac{9}{2\pi i}\int_{i\mathbb{R}+\epsilon}ds\frac{2^{s+2}(3^s-1)\zeta(2s+1)\Gamma(-s)\Gamma\left(s+\frac{3}{2}\right)_2F_1\left(-s-1,s+2;1;-\frac{\beta}{3}\right)(\lambda)^{s+1}}{\sqrt{\pi}(s+1)^2}$$

(5.40)

where $\epsilon$ is a small positive number. If we close the contour in (5.40) on the rhs we recover the small $\lambda$ (5.36), while if we close the contour on the lhs we make contanct with the perturbative expansion at large $\lambda$. Such perturbative expansion however truncates after a few terms and we get[14]

$$\log\Delta G_{\beta m}^m\simeq-3(2\beta+3)\lambda\log(3)+\log\left(\frac{\beta}{3}+1\right)+24\log(A)$$

$$+\log(\lambda)-2-\frac{\log(12)}{2}+\mathcal{O}\left(e^{-A_1(\beta)\lambda^{1/2}}\right),$$

(5.41)

where $A_1(\beta)$ is the leading instanton action. For a generic value of $\beta$, we find numerically in Appendix B, that the leading instanton action gives

$$A_1(\beta)\sqrt{\lambda}=\frac{2\sqrt{6}\pi}{\sqrt{2\beta+2\sqrt{\beta(\beta+3)}+3}}\sqrt{\lambda}.$$

(5.42)

In particular in the limit $\beta\to\infty$ we have

$$A_1(\beta)\sqrt{\lambda}=\frac{\sqrt{6}\pi\sqrt{\kappa}}{\beta}+O\left(\left(\frac{1}{\beta}\right)^2\right).$$

(5.43)

---

[14]Note that we could have done exactly the same procedure by keeping $\kappa$ instead of $\lambda$ and the same conclusions hold as long as we keep $\beta$ fixed.

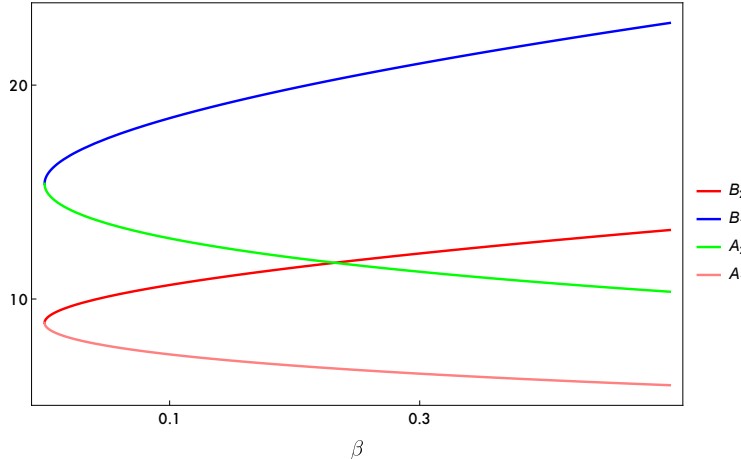

**Figure 2.** The four independent instantons actions in (5.42), (5.44), (5.45) as a function of $\beta$. Other actions will be an integer multiple of these four.

Therefore this instanton action vanishes in the limit $\beta \to \infty$. Consequently, we anticipate a reorganization of (5.41) and the emergence of a new perturbative series in this limit. As we discuss in subsection 5.3, this is indeed the case.

In addition, due to the splitting structure in (5.37), it follows that there is also another instanton action which is

$$A_2(\beta)\sqrt{\lambda} = \frac{2\sqrt{18}\pi}{\sqrt{2\beta + 2\sqrt{\beta(\beta+3)} + 3}}\sqrt{\lambda} \; . \tag{5.44}$$

It is natural to conjecture that there are also two other independent instanton actions which are subleading w.r.t (5.42). These are

$$B_i(\beta)\lambda^{1/2} = 2\pi\sqrt{\lambda/\lambda_c^{(i)}(\beta)}, \quad i = 1, 2 \tag{5.45}$$

where $\lambda_c^{(i)}(\beta)$ is the radius of convergence of the weak coupling expansion (5.38). These two actions naturally interpolate between the two independent actions at $\beta = 0$, namely (5.21), and the two independent actions at $\beta \to \infty$, namely (5.56). We plot the four actions on Figure 2. We notice that there is an interesting crossing between $A_2(\beta)$ and $B_2(\beta)$ happening at $\beta = \sqrt{3} - \frac{3}{2}$. A more detailed discussion on these non-perturbative effects and their physical meaning will appear elsewhere.

As a final remark, let us note that taking the $\beta = \frac{n}{m} \to 0$ limit is also equivalent to setting $n = 0$ from the very beginning

$$\log \Delta G_{\beta m}^m \Big|_{\beta=0} \simeq \log \Delta G_0^m \; . \tag{5.46}$$

## 5.3 The $G_n^m$ correlators at $\beta = \frac{n}{m} \to \infty$

Let us now consider (5.35) in the limit $\beta \to \infty$, i.e. $\kappa \gg \lambda$. We have

$$\lim_{\beta \to \infty} \log \Delta G_{\beta m}^m \simeq \sum_{k \geq 1} \frac{(-1)^k 2^{3k+4} 3^{1-k} \left(3^k - 1\right) \kappa^{k+1} \zeta(2k+1) \Gamma\left(k + \frac{3}{2}\right)^2}{\pi(k+1)^3 \Gamma(k+1)^2}. \tag{5.47}$$

As before, it is natural to split this sum into

$$\begin{aligned}
\sum_{k \geq 1} \frac{(-1)^k 2^{3k+4} 3^{1-k} \kappa^{k+1} \zeta(2k+1) \Gamma\left(k + \frac{3}{2}\right)^2}{\pi(k+1)^3 \Gamma(k+1)^2} \\
\sum_{k \geq 1} \frac{(-1)^k 2^{3k+4} 3^{1-k} 3^k \kappa^{k+1} \zeta(2k+1) \Gamma\left(k + \frac{3}{2}\right)^2}{\pi(k+1)^3 \Gamma(k+1)^2}
\end{aligned} \tag{5.48}$$

which have radius of convergence $\kappa_c^{(1)} = 1/8$ and $\kappa_c^{(2)} = 3/8$ respectively.

Note that $\beta \to \infty$ means $n \gg m \gg 1$. This is different from taking $m$ to be small in the correlators $G_n^m$. Indeed our matrix model derivation of (5.35) necessarily requires $m$ to be large, while $n$ can also be either fixed or large.

From the weak coupling expansion (5.47), one can easily see that

$$\lim_{\beta \to \infty} \log \Delta G_{\beta m}^m \simeq -12 \int_0^\infty \frac{e^x}{x(e^x - 1)^2} \left(2 + J_0(x\sqrt{2\kappa})^2 - 3J_0\left(x\sqrt{\frac{2\kappa}{3}}\right)^2\right) \tag{5.49}$$

However, extracting the strong coupling expansion from (5.49) is not straightforward. We find more convenient to use the Mellin-Barnes transformation of (5.47) which reads

$$\lim_{\beta \to \infty} \log \Delta G_{\beta m}^m \simeq \frac{1}{i} \int_{i\mathbb{R} + \epsilon} \frac{2^{3s+3} 3^{1-s} (3^s - 1) \kappa^{s+1} \zeta(2s+1) \Gamma(-s) \Gamma\left(s + \frac{3}{2}\right)^2}{\pi^2 (s+1)^3 \Gamma(s+1)} ds . \tag{5.50}$$

If we close the contour in (5.50) on the r.h.s. we recover the weak coupling expansion (5.47) while to get the strong coupling expansion we have to close it on the l.h.s. In the latter case we have non vanishing residue at

$$s = 0, -1, \quad s = -\frac{2n+1}{2}, \quad n \geq 1 \tag{5.51}$$

leading to

$$\lim_{\beta \to \infty} \log \Delta G_{\beta m}^m \simeq 24 \log(A) + \log\left(\frac{\kappa}{6\sqrt{3}}\right) - 6\kappa \log(3) - 2 + F^p(\kappa) + \mathcal{O}(e^{-A_1 \kappa^{1/2}}), \tag{5.52}$$

where $A_1 > 0$ is the leading instanton action which we will determine below (see (5.56)) and

$$F^p(\kappa) = \sum_{n \geq 0} \frac{3 \, 2^{-3n - \frac{1}{2}} \left(3^{n + \frac{3}{2}} - 1\right)(n+1) \pi^{-2n - \frac{9}{2}} \kappa^{-n - \frac{1}{2}} \zeta(2n+3) \Gamma\left(n + \frac{1}{2}\right)^3}{\Gamma(n+1)} . \tag{5.53}$$

We see that in this regime there is an important difference w.r.t. the behaviour in (5.41) and (5.18): the perturbative expansion at strong coupling does not truncate. Indeed the poles at $s = 0, 1$ are analogous of what we have in (5.41) and (5.18), while the poles at $s = \frac{2n+1}{2}$ produces a new, infinite series of perturbative corrections in $1/\kappa$. Moreover the series expansion in the second line of (5.52) is factorially divergent. A similar phenomenon was also observed in the study of integrated correlators [26], we will comment more on this in section 6.

Let us study the resurgence properties of the series in (5.53) and see how does it compare to the exact answer (5.49). The first important observation is that $F^{\mathrm{p}}(\kappa)$ is not Borel summable along the real axis since there is a singularity in the Borel plane located at

$$\left(4\pi\sqrt{\frac{2}{3}}\right)^2,\tag{5.54}$$

see Appendix C. However we can easily perform median Borel summation and we can easily check numerically with very high precision that median summation of $F^{\mathrm{p}}(\kappa)$ matches precisely the exact answer in (5.49)

$$\frac{1}{\sqrt{\kappa}}s_F(\sqrt{\kappa}) = -12\int_0^\infty \frac{e^x}{x(e^x-1)^2}\left(2 + J_0(x\sqrt{2\kappa})^2 - 3J_0\left(x\sqrt{\frac{2\kappa}{3}}\right)^2\right)$$
$$- 24\log(A) + 6\kappa\log(3) - \log(\kappa) + 2 + \frac{\log(3)}{2} + \log(6),\tag{5.55}$$

where $A$ is the Glaisher constant and $s_F(\sqrt{\kappa})$ denotes the median Borel summation of $F^{\mathrm{p}}$, see (C.4). We show the details in appendix Appendix C.

Therefore, in this limit, the resurgence analysis of the strong coupling expansion capture the all the relevant non-perturbative effects. As a consequence, in this limit, we can extract analytically the full non-perturbative structure from the large order behaviour of the coefficients in the $F^{\mathrm{p}}(\kappa)$. The analysis is shown in Appendix C. We find that the instantons actions are integer multiple of the following two independent actions

$$A_1\sqrt{\kappa} = 4\pi\sqrt{2/3}\sqrt{\kappa}, \qquad A_2\sqrt{\kappa} = 4\pi\sqrt{2}\sqrt{\kappa}.\tag{5.56}$$

Also in this case the two instantons actions are $2\pi\sqrt{\frac{1}{\kappa_c^{(i)}}}$ where $\kappa_c^{(i)}$ are the radii of convergence of the two weak coupling series (5.48).

# 6 Integrated correlators in $\mathcal{N} = 4$ SYM

One interesting generalization of extremal correlators in $\mathcal{N} = 4$ are the so called integrated correlators of half-BPS operators, see for instance [14, 41] and reference therein. In this section we apply our matrix model techniques to study such correlators in the $SU(3)$ gauge theory. Let us start by reviewing the standard notation. We follow [26, 42] and denote

half-BPS operators in terms of the six real scalars of the theory $\Phi^I$, $I = 1, \cdots, 6$. [15]. We have two independent single trace operators

$$\Phi_2(x, y) = y_{I_1} y_{I_2} \mathrm{Tr}\big(\Phi^{I_1}(x)\Phi^{I_2}(x)\big) \,,$$
$$\Phi_3(x, y) = y_{I_1} y_{I_2} y_{I_3} \mathrm{Tr}\big(\Phi^{I_1}(x)\Phi^{I_2}(x)\Phi^{I_3}(x)\big) \,, \tag{6.1}$$

where $y_I$ are $SO(6)$ null polarization vectors obeying $y^2 = y_I y^I = 0$. Note that it is only for a particular choice of the polarization vectors $y$ that these operator are the $\phi_2$ and $\phi_3$ operators in (2.6), see for instance [43]. Multi-traces operators can be obtain from product of the form

$$\Phi_2^{n_2}\Phi_3^{n_3}(x, y) = \Phi_2^{n_2}(x, y)\Phi_3^{n_3}(x, y) \tag{6.2}$$

where

$$\Delta(\Phi_2^{n_2}\Phi_3^{n_3}) = 2n_2 + 3n_3 \,. \tag{6.3}$$

Since these are half-BPS conformal primaries, their two-point functions are three level exact. Let us take

$$\Delta = 2n + 3n_3 = 2m + 3m_3 \,. \tag{6.4}$$

We have

$$\langle\Phi_2^n\Phi_3^{n_3}(x_1, y_1)\Phi_2^m\Phi_3^{m_3}(x_2, y_2)\rangle_{\mathbb{R}^4}^{\mathcal{N}=4} = \left(\frac{(y_1 - y_2)^2}{(x_1 - x_2)^2}\right)^\Delta \frac{\Delta^2}{(4\pi\mathrm{Im}\tau)^\Delta} R_{n,m}^\Delta \tag{6.5}$$

where $R_{n,m}^\Delta$ are some numbers that generically depends on the rank of the gauge group, see e.g. [42, eq.(2.4)] and reference therein. In the $SU(3)$ example we have

$$R_{1,1}^2 = 4, \quad R_{0,0}^3 = \frac{40}{9}, \quad R_{2,2}^4 = 40, \quad R_{1,1}^5 = \frac{224}{5}, \quad \cdots \tag{6.6}$$

Since the $y$ dependence in the two-point function factorizes, we simply have

$$\langle\Phi_2^n\Phi_3^{n_3}(x_1, y_1)\Phi_2^m\Phi_3^{m_3}(x_2, y_2)\rangle_{\mathbb{R}^4}^{\mathcal{N}=4} = \left(\frac{(y_1 - y_2)^2}{(x_1 - x_2)^2}\right)^\Delta \left\langle\phi_2^n\phi_3^{n_3}\overline{\phi_2^m\phi_3^{m_3}}\right\rangle_{\mathbb{R}^4}^{\mathcal{N}=4} \tag{6.7}$$

where $\phi_{2,3}$ are the operators in (2.6). The relation (6.7) is specific to two-point functions and does not hold in general. Integrated correlators, in particular, are generally not extremal and, therefore, cannot be reduced to two-point functions of the $\phi_k$'s.

For our purposes, it is convenient to work in a new basis of 1/2-BPS operators which we denote by $\{\Psi_n^m\}_{m,n \geq 0}$ and it is defined as follows. The $\Psi_n^m$ operator is constructed starting from $\Phi_2^n\Phi_3^m$ and then applying GS orthogonalization on $\mathbb{R}^4$ with respect to all operators $\Phi_3^j\Phi_2^k$ such that $3j + 2k = 3m + 2n$ and $j < m$. That is

$$\Psi_n^m(x, y) = \Phi_2^n\Phi_3^m(x, y) + \sum_{j,k} c_{j,k}\Phi_3^j\Phi_2^k(x, y) \tag{6.8}$$

---

[15]We would like to warn the reader that in this section, we are using the notation of the $\mathcal{N} = 4$ algebra, whereas in the previous sections, we used the notation of the $\mathcal{N} = 2$ algebra.

where the coefficients are determined by the GS procedure on $\mathbb{R}_4$ and the sum is subject to the constraint $3j + 2k = 3m + 2n$ with $j < m$. Let us stress that the scalar product used in this GS procedure is the two point function in the $\mathcal{N} = 4$ theory on $\mathbb{R}^4$. For example

$$\Psi_0^2 = \Phi_3^2(x,y) - \frac{\langle \Phi_3^2 \Phi_2^3 \rangle_{\mathbb{R}^4}^{\mathcal{N}=4}}{\langle \Phi_2^3 \Phi_2^3 \rangle_{\mathbb{R}^4}^{\mathcal{N}=4}} \Phi_2^3(x,y) = \Phi_3^2(x,y) - \frac{1}{24} \Phi_2^3(x,y) \ . \tag{6.9}$$

In this new basis two point correlations take the form (we take $m$ even for concreteness and to lighten the notation)

$$\langle \Psi_n^m \Psi_n^m \rangle_{\mathbb{R}^4} = \frac{\det_{\ell,k=0,\ldots,\frac{m}{2}} \left( \langle \Phi_2^{2m+2n-3k-3\ell} \Phi_3^{2k+2\ell} \rangle_{\mathbb{R}^4}^{\mathcal{N}=4} \right)}{\det_{\ell,k=0,\ldots,\frac{m}{2}-1} \left( \langle \Phi_2^{2m+2n-3k-3\ell} \Phi_3^{2k+2\ell} \rangle_{\mathbb{R}^4}^{\mathcal{N}=4} \right)}$$
$$= \frac{\det_{\ell,k=0,\ldots,\frac{m}{2}} \left( \langle \phi_2^{2m+2n-3k-3\ell} \phi_3^{2k+2\ell} \rangle_{\mathbb{R}^4}^{\mathcal{N}=4} \right)}{\det_{\ell,k=0,\ldots,\frac{m}{2}-1} \left( \langle \phi_2^{2m+2n-3k-3\ell} \phi_3^{2k+2\ell} \rangle_{\mathbb{R}^4}^{\mathcal{N}=4} \right)} \ . \tag{6.10}$$

The operators $\Psi_n^m$ correspond to the $\mathcal{O}_n^m$ operators discussed in the previous sections. Therefore from (4.14) and (4.15) we obtain

$$\langle \Psi_n^m \Psi_n^m \rangle_{\mathbb{R}^4} = \frac{n! \Gamma(3m + n + 4)}{3^{m+1} 2^{6m+2n+1} \pi^{3m+2n} \mathrm{Im}\tau^{3m+2n}} \ . \tag{6.11}$$

We are interested in 4 point functions of the form[16]

$$\langle \Psi_1^0(x_1,y_1) \Psi_1^0(x_2,y_2) \Psi_n^m(x_3,y_3) \Psi_n^m(x_4,y_4) \rangle_{\mathbb{R}^4} = \frac{y_{12}^2}{x_{12}^2} \frac{y_{12}^{3m+2n}}{x_{12}^{3m+2n}} \left( \mathcal{G}_{m,n}^{\mathrm{free}} + \mathcal{I} \mathcal{H}_{m,n}(u,v,\tau) \right), \tag{6.12}$$

where $x_{ij} = x_i - x_j$, and $u, v$ are the cross ratios

$$u = \frac{x_{12}^2 x_{34}^2}{x_{13}^2 x_{24}^2} \ , \quad v = \frac{x_{14}^2 x_{23}^2}{x_{13}^2 x_{24}^2} \ . \tag{6.13}$$

$\mathcal{G}_{m,n}^{\mathrm{free}}$ is the free theory correlator and $\mathcal{I}$ is fixed by superconformal Ward identity, see [42, eq. (2.10)] and reference therein. All the nontrivial $\tau$ dependence is contained in the *dynamical* term $\mathcal{H}_{m,n}(u,v,\tau)$. Our main focus will be the integrated correlator

$$\mathcal{G}_{m,n} = -\frac{2}{\pi} \int_0^\infty dr \int_0^\pi d\theta \frac{r^3 \sin^2 \theta}{u^2} \mathcal{H}_{m,n}(1 + r^2 - 2r\cos\theta, r^2, \tau) \ . \tag{6.14}$$

It was first shown in [14] that $\mathcal{G}_{m,n}$ can be computed from localization on $S^4$. The starting point is the $\mathcal{N} = 2^*$ $SU(3)$ sphere partition function, that is

$$Z_{S^4} = \int_{\mathbb{R}^2} da_1 da_2 \prod_{1 \leq i < j \leq 3} (a_i - a_j)^2 Z_G(a_1, a_2, \mu) |Z_{\mathrm{inst}}(\mu, a_1, a_2, \tau)|^2 e^{-2\pi \mathrm{Im}\tau \, \mathrm{Tr}\, a^2 - \frac{2\pi^{3/2}}{6^{1/2}} \mathrm{Im}\tau_3 \, \mathrm{Tr}\, a^3}, \tag{6.15}$$

---

[16]These correlators become extremal only when the polarizations vectors $y$ are aligned, see [14].

where $\mu$ is the mass of the adjoint hypermultiplet in the $\mathcal{N} = 2^*$ theory and we are implicitly using

$$a_3 = -a_1 - a_2, \quad \mathrm{Tr} a^2 = \sum_{i=1}^{3} a_i^2 \,. \tag{6.16}$$

The function $Z_G(a_1, a_2, \mu)$ for the $\mathcal{N} = 2^*$ $SU(3)$ theory is given by

$$Z_G(a_1, a_2, \mu) = \frac{1}{H(\mu)^3} \frac{(H(a_2 - a_1)H(a_3 - a_2)H(a_3 - a_1))^2}{\prod_{\pm} H(a_2 - a_1 \pm \mu)H(a_3 - a_2 \pm \mu)H(a_3 - a_1 \pm \mu)} \bigg|_{a_3 = -a_1 - a_2}, \tag{6.17}$$

where $H(x) = G(1+x)G(1-x)$, $G$ being the Barnes G-function. We noted by $Z_{\mathrm{inst}}(\mu, a_1, a_2, \tau) = 1 + \mathcal{O}(e^{2\pi i \tau})$ the instanton partition function of the $\mathcal{N} = 2^*$ theory in the $\epsilon_1 = \epsilon_2 = 1$ phase of the $\Omega$ background [28–31]. One important property of $Z_G$ and $Z_{\mathrm{inst}}$ is that they are even functions of $\mu$, and they satisfy

$$\partial_\mu Z_G(a_1, a_2, \mu)|_{\mu=0} = \partial_\mu Z_{\mathrm{inst}}(\mu)|_{\mu=0} = 0 \,. \tag{6.18}$$

In terms of the operators $(\mathcal{O}_n^m)'$ defined in (2.38) the prescription to compute $\mathcal{G}_{m,n}$ simply reads [17]

$$\frac{4\mathcal{G}_{m,n}}{(G_n^m)^{\mathcal{N}=4}} = \frac{\partial_\mu^2 \int_{\mathbb{R}^2} d^2a \prod_{1 \leq i < j \leq 3}(a_i - a_j)^2 Z_G(a_1, a_2, \mu) e^{-2\pi \mathrm{Im}\tau \, \mathrm{Tr}\, a^2} (\mathcal{O}_n^m)'(\mathcal{O}_n^m)'}{\int_{\mathbb{R}^2} d^2a \prod_{1 \leq i < j \leq 3}(a_i - a_j)^2 e^{-2\pi \mathrm{Im}\tau \, \mathrm{Tr}\, a^2} (\mathcal{O}_n^m)'(\mathcal{O}_n^m)'} \bigg|_{\mu=0}, \tag{6.19}$$

where we are using (6.16) and $(G_n^m)^{\mathcal{N}=4}$ is given in (4.15).

## 6.1 Matrix models for integrated correlators

The matrix model description of extremal correlators [17] can be straightforwardly generalized to integrated correlators as well. This generalization has been applied to the so-called "maximal-trace" family of operators in [24] using the approach of [23]. The resulting expression for this family of operators allows for the computation of the large $n$ expression of $\mathcal{G}_{m,n}$ at $m$ fixed, but it is not suitable to control the $m \to \infty$ limit. See also [26, 44–46] for other recent studies of the large $n$ limit. In this section we show how our matrix model for extremal correlators can be adapted to compute integrated correlators as well. Despite restricting our analysis to the $SU(3)$ case, this allows us to scale independently $m, n \to \infty$.

We closely follow the derivation of [24], and adapt the final result to our matrix model. The operators $(\mathcal{O}_n^m)'$ can be thought as orthogonal polynomials with respect to the measure

$$d\xi = \prod_{1 \leq i < j \leq 3}(a_i - a_j)^2 \big|_{a_3 = -a_2 - a_1} e^{-2\pi \mathrm{Im}\tau \, \mathrm{Tr}\, a^2} da_1 da_2 \,, \tag{6.20}$$

that is

$$\int_{\mathbb{R}^2} d\xi \, (\mathcal{O}_n^m)'(\mathcal{O}_\ell^k)' = (G_n^m)^{\mathcal{N}=4} \delta_{mk} \delta_{n\ell} \,. \tag{6.21}$$

---

[17]As before we set $Z_{\mathrm{inst}} = 1$, see footnote 7.

Consider the deformed measure

$$d\tilde{\xi}(\mu) = \prod_{1 \leq i < j \leq 3} (a_i - a_j)^2 \big|_{a_3 = -a_2 - a_1} e^{-2\pi \mathrm{Im}\tau \,\mathrm{Tr}\, a^2} Z_G(a_1, a_2, \mu) da_1 da_2 \,. \qquad (6.22)$$

The operators $(\mathcal{O}_n^m)'$ are no longer orthogonal w.r.t. (6.22). Hence let us define a new set of operator $\widetilde{\mathcal{O}}_n^m = \phi_2^n \widetilde{\mathcal{O}}_0^m$, where

$$\widetilde{\mathcal{O}}_0^m = \frac{\det\left(\begin{array}{c} \left(\langle \overline{\phi_2^{3m-3n}\phi_3^{2n}}\phi_2^{3m-3n'}\phi_3^{2n'}\rangle_{S^4}^{\mathcal{N}=2^*}\right)_{\substack{n=0,\ldots,m,\\ n'=0,\ldots,m-1}} \\ \left(\phi_2^{3m-3n}\phi_3^{2n}\right)_{n=0,\ldots,m} \end{array}\right)}{\det\left(\langle \overline{\phi_2^{3m-3n}\phi_3^{2n}}\phi_2^{3m-3n'}\phi_3^{2n'}\rangle_{S^4}^{\mathcal{N}=2^*}\right)_{\substack{n=0,\ldots,m-1,\\ n'=0,\ldots,m-1}}} \,, \qquad (6.23)$$

with

$$\langle \phi_2^a \phi_3^b \overline{\phi_2^c \phi_3^d}\rangle_{S^4}^{\mathcal{N}=2^*} = \frac{1}{Z_{S^4}} \int_{\mathbb{R}^2} \phi_2^a \phi_3^b \overline{\phi_2^c \phi_3^d} \; \mathrm{d}\tilde{\xi}(\mu) \,. \qquad (6.24)$$

This means that $\widetilde{\mathcal{O}}_0^m$ are obtained starting with $\phi_3^m$ and by doing GS orthogonalization wrt operators of the same dimension. It is important that in this procedure the scalar product we use in GS is the two point function of $\mathcal{N} = 2^*$ on $S^4$. Since $\widetilde{\mathcal{O}}_n^m$ are by construction orthogonal to operators of the same dimension, it follows that

$$\int_{\mathbb{R}^2} d\tilde{\xi}(\mu) \, \widetilde{\mathcal{O}}_{3\ell}^{m-2\ell} \widetilde{\mathcal{O}}_{3k}^{m-2k} \propto \delta_{k\ell} \,. \qquad (6.25)$$

This is in complete analogy with (3.6), the only difference being that the scalar product is taken with respect to the mass deformed measure (6.22). Now

$$\left(\partial_\mu^2 \int_{\mathbb{R}^2} d\tilde{\xi}(\mu) \, \widetilde{\mathcal{O}}_0^m \widetilde{\mathcal{O}}_0^m\right)\Big|_{\mu=0} = \int_{\mathbb{R}^2} \left(\partial_\mu^2 d\tilde{\xi}(\mu)|_{\mu=0}\right) (\mathcal{O}_0^m)(\mathcal{O}_0^m) + \\ + 2\int_{\mathbb{R}^2} d\xi \left(\partial_\mu \widetilde{\mathcal{O}}_0^m|_{\mu=0}\right)^2 + 2\int_{\mathbb{R}^2} d\xi \, \mathcal{O}_0^m \partial_\mu^2 \widetilde{\mathcal{O}}_0^m|_{\mu=0} \,. \qquad (6.26)$$

The second term is zero thanks to (6.18). Regarding the last term note that we have

$$\widetilde{\mathcal{O}}_0^m = \phi_3^m + \sum_{k \geq 1}^{\frac{m}{2}} c_k(\mu)\widetilde{\mathcal{O}}_{3k}^{m-2k} \implies \partial_\mu^2 \widetilde{\mathcal{O}}_0^m|_{\mu=0} = \sum_{k \geq 1}^{\frac{m}{2}} \partial_\mu^2 c_k(\mu)|_{\mu=0}\phi_2^{3k}\mathcal{O}_0^{m-2k} \,. \qquad (6.27)$$

Therefore the last term will vanish thanks to (6.21), and we simply have

$$\partial_\mu^2 \int_{\mathbb{R}^2} d\tilde{\xi}(\mu) \, \widetilde{\mathcal{O}}_0^m \widetilde{\mathcal{O}}_0^m|_{\mu=0} = \int_{\mathbb{R}^2} \left(\partial_\mu^2 d\tilde{\xi}(\mu)|_{\mu=0}\right) \mathcal{O}_0^m \mathcal{O}_0^m \,. \qquad (6.28)$$

We further introduce the operators $\left(\widetilde{\mathcal{O}}_n^m\right)'$ as

$$\left(\widetilde{\mathcal{O}}_n^m\right)' = \frac{\det\left(\begin{array}{c} \left(\langle \widetilde{\mathcal{O}}_k^m \widetilde{\mathcal{O}}_\ell^m\rangle_{S^4}^{\mathcal{N}=2^*}\right)_{\substack{k=0,\ldots,n,\\ \ell=0,\ldots,n-1}} \\ \left(\widetilde{\mathcal{O}}_\ell^m\right)_{\ell=0,\ldots,m} \end{array}\right)}{\det\left(\langle \widetilde{\mathcal{O}}_k^m \widetilde{\mathcal{O}}_\ell^m\rangle_{S^4}^{\mathcal{N}=2^*}\right)_{\substack{k=0,\ldots,n-1,\\ \ell=0,\ldots,n-1}}} \,. \qquad (6.29)$$

This is the analogous of (2.38) but in $\mathcal{N} = 2^*$. We also have $\left(\widetilde{\mathcal{O}}_0^m\right)' = \widetilde{\mathcal{O}}_0^m$. By repeating the same argument as above we find

$$\partial_\mu^2 \int_{\mathbb{R}^2} d\tilde{\xi}(\mu) \left(\widetilde{\mathcal{O}}_n^m\right)' \left(\widetilde{\mathcal{O}}_n^m\right)'|_{\mu=0} = \int_{\mathbb{R}^2} \left(\partial_\mu^2 d\tilde{\xi}(\mu)|_{\mu=0}\right) (\mathcal{O}_n^m)'(\mathcal{O}_n^m)'. \tag{6.30}$$

Finally

$$\mathcal{G}_{m,n} = \frac{(G_n^m)^{\mathcal{N}=4}}{4} \frac{\int_{\mathbb{R}^2} \left(\partial_\mu^2 d\tilde{\xi}(\mu)|_{\mu=0}\right) (\mathcal{O}_n^m)'(\mathcal{O}_n^m)'}{\int_{\mathbb{R}^2} d\xi \, (\mathcal{O}_n^m)'(\mathcal{O}_n^m)'} = \frac{(G_n^m)^{\mathcal{N}=4}}{4} \frac{\partial_\mu^2 \int_{\mathbb{R}^2} d\tilde{\xi}(\mu) \left(\widetilde{\mathcal{O}}_n^m\right)' \left(\widetilde{\mathcal{O}}_n^m\right)'|_{\mu=0}}{\int_{\mathbb{R}^2} d\xi \, (\mathcal{O}_n^m)'(\mathcal{O}_n^m)'}. \tag{6.31}$$

The crucial observation now is that $\left(\widetilde{\mathcal{O}}_n^m\right)'$ are precisely the operators whose two point functions are computed by the $\mathcal{N} = 2$ matrix models discussed in section 5. Setting

$$\mathcal{M}_{i,j}^{(\alpha)} = \left\langle \left(\frac{\phi_3^2}{\phi_2^3}\right)^i \phi_3^{\sigma_m} \phi_2^{3\lfloor m/2 \rfloor} \overline{\left(\frac{\phi_3^2}{\phi_2^3}\right)^j \phi_3^{\sigma_m} \phi_2^{3\lfloor m/2 \rfloor}} \right\rangle_{S^4}^{\mathcal{N}=2^*}, \quad i,j = 0,\dots,\alpha-1, \tag{6.32}$$

where now the vev is taken with respect to the $\mathcal{N} = 2^*$ theory, in analogy with (2.59). We have

$$\int_{\mathbb{R}^2} d\tilde{\xi}(\mu) \left(\widetilde{\mathcal{O}}_n^m\right)' \left(\widetilde{\mathcal{O}}_n^m\right)' = Z_{S^4}^{-1} \frac{\det_{i,j=0,\dots,n}(-\partial_{2\pi\mathrm{Im}\tau})^{i+j}\left(Z_{S^4} \frac{\det \mathcal{M}_m^{(\lfloor m/2 \rfloor+1)}}{\det \mathcal{M}_m^{(\lfloor m/2 \rfloor)}}\right)}{\det_{i,j=0,\dots,n-1}(-\partial_{2\pi\mathrm{Im}\tau})^{i+j}\left(Z_{S^4} \frac{\det \mathcal{M}_m^{(\lfloor m/2 \rfloor+1)}}{\det \mathcal{M}_m^{(\lfloor m/2 \rfloor)}}\right)} =$$

$$= \frac{\det \mathcal{K}_m^{(n+1)}}{\det \mathcal{K}_m^{(n)}}, \tag{6.33}$$

where $\mathcal{K}_m^{(\alpha)}$ is the $\alpha \times \alpha$ matrix whose elements are

$$\mathcal{K}_{i,j} = \frac{(-\partial_{2\pi\mathrm{Im}\tau})^{i+j}}{Z_{S^4}} \left(Z_{S^4} \frac{\det \mathcal{M}_m^{(\lfloor m/2 \rfloor+1)}}{\det \mathcal{M}_m^{(\lfloor m/2 \rfloor)}}\right). \tag{6.34}$$

In particular, parallel to subsection 5.1, we find the following matrix model representation for $\det \mathcal{M}$

$$\det \mathcal{M}_m^{(k)} = \frac{Z_{\mathrm{J}}^{(\sigma_m)}(k)\sqrt{3}^k}{6^{k^2+k\sigma_m}(Z_{S^4})^k} \langle \mathcal{F}_m(x, \mathrm{Im}\tau, \mu) \rangle_{\mathrm{J}}^{(k)}, \tag{6.35}$$

where the expectation value is w.r.t. the Jacobi model (4.4) and

$$\mathcal{F}_m(x, \mathrm{Im}\tau, \mu) = \int_0^\infty dy \, e^{-2\pi\mathrm{Im}\tau y} y^{3m+3} \mathcal{Z}_G(\mu, x, y) \tag{6.36}$$

with

$$\mathcal{Z}_G(\mu, x, y) = Z_G(\mu, a_1(x, y), a_2(x, y)) \tag{6.37}$$

and we are implicitly using the change of variable (2.41). The integrated correlator is now given as

$$\Delta\mathcal{G}_{m,n} = 4\frac{\mathcal{G}_{m,n}}{(G_n^m)^{\mathcal{N}=4}} = \left(\frac{\partial_\mu^2 \det \mathcal{K}_m^{(n)}}{\det \mathcal{K}_m^{(n)}} - \frac{\partial_\mu^2 \det \mathcal{K}_m^{(n-1)}}{\det \mathcal{K}_m^{(n-1)}}\right)\Bigg|_{\mu=0}. \tag{6.38}$$

## 6.2 Double scaling limits

We now study the integrated correlators in the double scaling limits

$$n \to \infty\,,\ \mathrm{Im}\tau \to \infty\,,\ \kappa = \frac{n}{2\pi\mathrm{Im}\tau}\,,\ m \text{ fixed.} \tag{6.39}$$

$$m \to \infty\,,\ \mathrm{Im}\tau \to \infty\,,\ \lambda = \frac{m}{2\pi\mathrm{Im}\tau}\,,\ n \text{ fixed,} \tag{6.40}$$

$$m, n \to \infty\,,\ \mathrm{Im}\tau \to \infty\,,\ \lambda = \frac{m}{2\pi\mathrm{Im}\tau}\,,\ \kappa = \frac{n}{2\pi\mathrm{Im}\tau} \text{ fixed.} \tag{6.41}$$

Note that for integrated correlators we can use our matrix model representation to study the limit (6.39) as well since, contrarily to the $\mathcal{N} = 2$ story, here the representation (6.38) holds for any $n, m, \mathrm{Im}\tau$.

### 6.2.1  $m$ fixed and $n \to \infty$

Let us start by taking $m = 0$, which is the example studied in [24, 26] and [44][18]. In this case we have

$$\partial_\mu^2 \det \mathcal{K}_0^{(n)} = \partial_\mu^2 \det \int_0^\infty dy\, e^{-2\pi\mathrm{Im}\tau y} y^3 \int_0^1 dx\, (x-1)^{\frac{1}{2}} x^{\frac{1}{2}} \mathcal{Z}_G(\mu, x, y) =$$
$$= \frac{\partial_\mu^2}{(n+1)!} \int_{\mathbb{R}_+^k} d^k y \prod_{i<j} (y_i - y_j)^2 y_j^3 e^{-\frac{n}{\kappa} y_i} \left( \int_0^1 dx\, (1-x)^{\frac{1}{2}} x^{\frac{1}{2}} \mathcal{Z}_G(\mu, x, y) \right). \tag{6.42}$$

This is again a Wishart matrix model as we saw before. Hence $\mathcal{G}_{0,n}$ can be simply written as ratio of matrix models of type (6.42) in complete agreement with [24]. We now study the large charge double scaling limit of the matrix models, that is

$$m = 0\,,\ n \to \infty\,,\ \mathrm{Im}\tau \to \infty\,,\ \kappa = \frac{n}{2\pi\mathrm{Im}\tau} \quad \text{fixed.} \tag{6.43}$$

Parallel to what we discussed in section 5, we can evaluate (6.42) in the large $n$ limit as

$$\frac{\partial_\mu^2 \det \mathcal{K}_0^{(n)}}{\det \mathcal{K}_0^{(n)}} \bigg|_{\mu=0} \simeq \partial_\mu^2 \frac{1}{n!} \exp\left( \int_0^4 dy\, \rho_{\mathrm{MP}}^{(0)}(y) \int_0^1 dx\, (1-x)^{\frac{1}{2}} x^{\frac{1}{2}} \mathcal{Z}_G(\mu, x, \kappa y) \right) \bigg|_{\mu=0}$$
$$\simeq \frac{1}{n!} \int_0^4 dy\, \rho_{\mathrm{MP}}^{(0)}(y) \int_0^1 dx\, (1-x)^{\frac{1}{2}} x^{\frac{1}{2}} \partial_\mu^2 \mathcal{Z}_G(\mu, x, \kappa y) \bigg|_{\mu=0}. \tag{6.44}$$

where $\rho_{\mathrm{MP}}^{(0)}(y)$ is the Marčenko-Pastur distribution with endpoints $a = 0, b = 4$, see (A.5), (A.9). This gives

$$\mathcal{G}'_{0,n} = (1 + \kappa \partial_\kappa) \int_0^4 dy\, \rho_{\mathrm{MP}}^{(0)}(y) \int_0^1 dx\, (1-x)^{\frac{1}{2}} x^{\frac{1}{2}} \partial_\mu^2 \mathcal{Z}_G(\mu, x, \kappa y) \bigg|_{\mu=0}. \tag{6.45}$$

---

[18]In [44] also cases with $m \neq 0$, but always of $\mathcal{O}(1)$, are taken into account.

Expanding (6.45) at weak coupling we find the following all order expression

$$\Delta \mathcal{G}_{0,n} \simeq \sum_{k \geq 1} 48 \frac{(-1)^{k+1} 2^{3k} \Gamma\left(\frac{3}{2} + k\right)^2}{\pi \Gamma(1+k)^2} \frac{6 + k + k^2}{(1+k)(2+k)(3+k)} \zeta(2k+1) \kappa^k \,, \tag{6.46}$$

in complete agreement with [24, 26].

Let us consider other examples where $m$ is fixed but $m \neq 0$. Let us start with $m = 2$. Now the $\det \mathcal{M}_2^2$ is the determinant of a $2 \times 2$ matrix and we get

$$Z_{S^4} \frac{\det \mathcal{M}_2^{(2)}}{\det \mathcal{M}_2^{(1)}} \simeq \frac{1728}{\pi} \left( \int_0^{\frac{\pi}{3}} \frac{d\theta}{288} \frac{\sin^4 3\theta}{\sin^2 \frac{3\theta}{2}} \mathcal{Z}_G(\mu, x(\theta), 3\kappa) - \frac{\left( \int_0^{\frac{\pi}{3}} \frac{d\theta}{12} \sin^2 3\theta \mathcal{Z}_G(\mu, x(\theta), 3\kappa) \right)^2}{\int_0^{\frac{\pi}{3}} 2 d\theta \sin^2 \frac{3\theta}{2} \mathcal{Z}_G(\mu, x(\theta), 3\kappa)} \right) \,, \tag{6.47}$$

where we used the trick (5.9) and performed the change of variables

$$x = \frac{\cos 3\theta + 1}{2} \,. \tag{6.48}$$

Setting for simplicity $q(3\kappa, \mu) \equiv Z_{S^4} \frac{\det \mathcal{M}_2^{(2)}}{\det \mathcal{M}_2^{(1)}}$ we find

$$\Delta \mathcal{G}_{2,n} \simeq (1 + \kappa \partial_\kappa) \int_0^4 dy \, \rho_{\mathrm{MP}}^{(0)}(y) \partial_\mu^2 q(y\kappa, \mu)|_{\mu=0} = $$
$$= \sum_{k \geq 1} 48 \frac{(-1)^{k+1} 2^{3k} \Gamma\left(\frac{3}{2} + k\right)^2}{\pi \Gamma(1+k)^2} p_2(k) \zeta(2k+1) \kappa^k \,, \tag{6.49}$$

with

$$p_2(k) = \frac{362880 + 341136k + 380844k^2 + 80116k^3 + 41809k^4 + 2104k^5 + 706k^6 + 4k^7 + k^8}{(1+k)_9} \,. \tag{6.50}$$

Empirically we find for generic $m$

$$\Delta \mathcal{G}_{m,n} \simeq \sum_{k \geq 1} 48 \frac{(-1)^{k+1} 2^{3k} \Gamma\left(\frac{3}{2} + k\right)^2}{\pi \Gamma(1+k)^2} p_m(k) \zeta(2k+1) \kappa^k \,, \tag{6.51}$$

$$p_m(k) = \frac{1}{(k+1)_{3m+3}} \sum_{j=1}^{3m+2} v_j^{(m)} k^j \,.$$

where $p_m(k)$ satisfies

$$p_m(k) = (2k+1)^{-1} \,, k = 1, \ldots, 3m+2 \,, \tag{6.52}$$

$$p_m(k) = \frac{1}{k} + \mathcal{O}\left(k^{-2}\right) \,. \tag{6.53}$$

In particular (6.53) gives $p_m(n)/p_m(n+1) \to 1$ as $n \to \infty$, therefore all the weak coupling series (6.51) will have the same radius of convergence for any fixed value of $m$, that is

$$\kappa^* = \frac{1}{8} \,. \tag{6.54}$$

Following the argument explained in section 7.1 of [47], we expect non-perturbative corrections at strong coupling with instanton action $2\pi\sqrt{1/\kappa^*}$.

Note that the conditions (6.52) and (6.53) are enough to determine all the $v_j^{(m)}$'s. From (6.53) we immediately find $v_{3m+2}^{(m)} = 1$, and (6.52) gives the set of constraints

$$\frac{1}{2n+1}(n+1)_{3m+3} - v_0^{(m)} - n^{3m+2} = \sum_{j=1}^{3m+1} v_j^{(m)} n^j, \quad n = 1, \ldots 3m+1,$$

$$v_0^{(m)} = (1)_{3m+3},$$

(6.55)

which can be solved efficiently for any give $m$. For completeness, one finds

$$v_{3m+1}^{(m)} = \frac{1}{2}(2+3m), \quad v_{3m}^{(m)} = \frac{1}{8}(2+3m)(24+77m+78m^2+27m^3), \ldots$$

(6.56)

Proceeding as before we can extract the large $\kappa$ behavior of (6.51) using the Mellin-Barnes transform. We find

$$\Delta\mathcal{G}_{m,n} \simeq K_m + 6\log\frac{\kappa}{2} + \sum_{k\geq 1}\frac{\Gamma\left(k+\frac{1}{2}\right)^3 \pi^{-2k-\frac{5}{2}} 2^{-\frac{1}{2}-3k}}{\Gamma(k+1)}\zeta(2k+1)\frac{q_m(k)}{\kappa^{k+\frac{1}{2}}} + \mathcal{O}\left(e^{-2\pi\sqrt{8\kappa}}\right),$$

$$q_m(k) = 48k^2\frac{\sum_{j=0}^{\lfloor\frac{2+3m}{2}\rfloor} w_j k^{2j}}{(-2^{2+3m})(2k-1)\left(\frac{3}{2}-k\right)_{2+3m}}, \quad q_m(k) = \begin{cases} 24k + \mathcal{O}((k^{-1})^0), & m \text{ even} \\ -423 + \mathcal{O}((k^{-1})^0), & m \text{ odd}, \end{cases}$$

$$K_m = 12 + \frac{2}{m+1} + 12\gamma_E.$$

(6.57)

For the first few $m$'s we find

$$\sum_{j=0}^{\lfloor\frac{2+3m}{2}\rfloor} w_j n^{2j}\bigg|_{m=0} = 23 + 4n^2,$$

$$\sum_{j=0}^{\lfloor\frac{2+3m}{2}\rfloor} w_j n^{2j}\bigg|_{m=1} = -12(1627 + 1160n^2 + 48n^4),$$

$$\sum_{j=0}^{\lfloor\frac{2+3m}{2}\rfloor} w_j n^{2j}\bigg|_{m=2} = 71697105 + 82281776n^2 + 10030944n^4 + 178944n^6 + 256n^8.$$

(6.58)

Let

$$a_n^{(m)} = \frac{\Gamma\left(n+\frac{1}{2}\right)^3 \pi^{-2n-\frac{5}{2}} 2^{-\frac{1}{2}-3n}}{\Gamma(n+1)}\zeta(2n+1)\frac{q_m(n)}{\kappa^{n+\frac{1}{2}}}$$

(6.59)

be the coefficients in the doubly factorially divergent series in (6.57). Then we can write it as

$$a_n^{(m)} = \frac{4}{\pi}\sum_{\ell\geq 1}\left(\ell 4\sqrt{2}\pi\right)^{-2n-1}\Gamma(2n+1)\left(-24 + \sum_{k\geq 1}\frac{c_k^{(m)}(4\sqrt{2}\pi)^k}{\prod_{i=1}^k(2n+1-i)}\right).$$

(6.60)

We can adapt the analysis in Appendix C to the coefficients (6.60). The series is not Borel summable on the real axis[19], and its discontinuity is given by

$$\text{disc}^{(m)}(\kappa) = \text{i} e^{-4\pi\sqrt{2\kappa}}\left(24 + \sum_{n\geq 1}\frac{c_n^{(m)}}{\kappa^{n/2}}\right) + \text{i}\sum_{\ell\geq 2}e^{-4\pi\ell\sqrt{2\kappa}}\left(24 + \sum_{n\geq 1}\frac{c_n^{(m)}}{\ell^n\kappa^{n/2}}\right). \tag{6.61}$$

Since the sums over $n$ on the r.h.s. are divergent, to make sense of them we need to replace them by their median Borel summation as in Appendix C. Note that we have only one instanton action

$$4\pi\sqrt{2} = 2\pi\sqrt{\frac{1}{\kappa^*}} \tag{6.62}$$

as opposed to the case of $\mathcal{N} = 2$ SQCD where we always find two independent sectors. However notice that the action (6.62) matches one of the two we find in the SQCD example (5.56). Furthermore, the instanton action is completely $m-$independent. The $c_n^{(m)}$ coefficients can be easily derived and the list is available upon request. For $m = 0$ they agree with [26, eq. (3.39)].

### 6.2.2   $n$ fixed and $m \to \infty$

We focus on $n = 0$ for concreteness, but the same analysis holds for any other fixed value of $n$. In fact since $\partial_\tau \sim m^{-1}\partial_\lambda$, the effect of considering $n \neq 0$ is subleading in the double scaling limit

$$m \to \infty\,, \;\; \text{Im}\tau \to \infty\,, \;\; \lambda = \frac{m}{2\pi\text{Im}\tau} \;\; \text{fixed.} \tag{6.63}$$

In this case we simply have

$$\Delta\mathcal{G}_{m,0} = \left(\frac{\partial_\mu^2 \det \mathcal{M}_m^{\lfloor m/2\rfloor+1}}{\det \mathcal{M}_m^{\lfloor m/2\rfloor+1}} - \frac{\partial_\mu^2 \det \mathcal{M}_m^{\lfloor m/2\rfloor}}{\det \mathcal{M}_m^{\lfloor m/2\rfloor}}\right)\Bigg|_{\mu=0}\,, \tag{6.64}$$

Hence we recover a representation only in terms of a Jacobi matrix model, see (6.35). Following the analysis in section 5.1, $\Delta\mathcal{G}_{m,0}$ can be evaluated as

$$\Delta\mathcal{G}_{m,0} \simeq \int_0^1 \sigma_\text{J}(x)\partial_\mu^2\mathcal{Z}_G(\mu, x, 3\lambda)|_{\mu=0}\,, \tag{6.65}$$

where $\sigma_\text{J}(x)$ is the Jacobi density (5.13). Expanding the integral we find the following all order expression

$$\Delta\mathcal{G}_{m,0} \simeq \sum_{k\geq 1}\frac{(-1)^{n+1}2^{2-n}3^{1+n}\Gamma(2n+2)}{\Gamma(n+1)^2}\zeta(2n+1)\lambda^n\,, \tag{6.66}$$

that has radius of convergence

$$\lambda^* = \frac{1}{6}\,. \tag{6.67}$$

---

[19]We nevertheless expect its median summation to agree with the Mellin-Barnes representation as it happens for $m = 0$, see [26].

This looks exactly the same as the $SU(2)$ expression derived in [24, eq.(5.130)], see also [26, 44]. This may seems strange as here we are considering the $SU(3)$ theory and we are taking the large $m$ limit. However, it could the that in this particular sector of the $SU(3)$ theory, the large charge expansion is captured by an EFT which is similar the one describing rank 1 theories discussed for instance in [15, 24, 48].

Equation (6.66) gives

$$\Delta\mathcal{G}_{m,0} \simeq 12 \int_0^\infty dx\, \frac{x}{\sinh^2 x}\left(1 - J_0(2x\sqrt{6\lambda})\right),\tag{6.68}$$

which evaluated at strong coupling reads

$$\Delta\mathcal{G}_{m,0} \simeq 12 + 12\gamma_E + 6\log\frac{3\lambda}{2} + 24\sum_{n\geq 1}\left(4\pi n\sqrt{3/2\lambda}K_1(4n\pi\sqrt{3/2\lambda}) - K_0(4n\pi\sqrt{3/2\lambda})\right).\tag{6.69}$$

Again we only have a single instanton action

$$4\pi\sqrt{\frac{3\lambda}{2}} = 2\pi\sqrt{\frac{1}{\lambda^*}}\tag{6.70}$$

as opposed to extremal correlator in $\mathcal{N} = 2$ SQCD. The action (6.70) matches one of the two we have in SQCD in the limit $n$ fixed $m \to \infty$ in (5.21).

### 6.2.3 $m, n \to \infty$ with $m/n$ fixed

We now consider the double scaling limit

$$m, n \to \infty,\ \ \mathrm{Im}\tau \to \infty,\ \ \lambda = \frac{m}{2\pi\mathrm{Im}\tau},\ \ \kappa = \frac{n}{2\pi\mathrm{Im}\tau}\ \ \text{fixed.}\tag{6.71}$$

Following the footprint laid out in section 5.2 we get to

$$\Delta\mathcal{G}_{m,\beta m} \simeq (1 + \kappa\partial_\kappa + \beta\partial_\beta)\left(\int_a^b dy\rho_{\mathrm{MP}}(y)\int_0^1 dx\sigma_{\mathrm{J}}(x)\log(\mathcal{Z}_{\mathrm{G}}(x, \kappa y))\right),\tag{6.72}$$

where $a$ and $b$ are as in equation (5.32) and as before $\beta = \kappa/\lambda = n/m$. Evaluating the integral we find

$$\Delta\mathcal{G}_{m,\beta m} \simeq \sum_{k\geq 1}\frac{(-1)^{k+1}2^{2-k}3^{1+k}\Gamma(2 + 2k)\zeta(2k+1)}{\Gamma(k+1)^2}\,{}_2F_1\left(-k, 1+k, 1, -\frac{\beta}{3}\right)\lambda^k.\tag{6.73}$$

The $\beta = 0$ result immediately reproduces equation (6.66). The radius of convergence of (A.17) is given by

$$\lambda^*(\beta) = \frac{1}{6}\lim_{n\to\infty}\frac{{}_2F_1\left(-n, n+1, 1, -\frac{\beta}{3}\right)}{{}_2F_1\left(-n-1, n+2, 1, -\frac{\beta}{3}\right)} = \frac{1}{6}\frac{3 + 2\beta - 2\sqrt{\beta(3+\beta)}}{3}\,.\tag{6.74}$$

Note that we have $\lambda^*(0) = 1/6$ in agreement with (6.67), and as $\beta \to \infty$

$$\beta \lambda^*(\beta) = \frac{1}{8} = \kappa^* \tag{6.75}$$

in agreement with (6.54). To obtain the strong $\lambda$ expansion we recast the sum into its Mellin-Barnes representation as in (5.40) and close the contour on the l.h.s. We find

$$\Delta \mathcal{G}_{m,\beta m} \simeq 12(1 + \gamma_E) + 6 \log \frac{3}{2} + 6 \log \left(1 + \frac{\beta}{3}\right) + 6 \log \lambda + \mathcal{O}\left(e^{-A(\beta)\sqrt{\lambda}}\right), \tag{6.76}$$

where $\gamma_E$ is the Euler gamma and $A(\beta)$ is the leading instanton action. Following the numerical approach described in Appendix B, we find the leading instanton action to be

$$A(\beta) = \frac{6\pi}{\sqrt{\beta + \sqrt{\beta(\beta+3)} + \frac{3}{2}}}. \tag{6.77}$$

For fixed $\beta$, the perturbative series in (6.76) truncates, and it matches with the perturbative part of (6.68). For $\beta \to \infty$ we have

$$A(\beta)\sqrt{\lambda} = \frac{3\sqrt{2}\pi\sqrt{\kappa}}{\beta} + O\left(\frac{1}{\beta^2}\right), \tag{6.78}$$

and therefore the leading instanton action vanish in this limit causing the emergence of a new perturbative series as we discuss in the next section.

It is also natural to conjecture that there is another independent instanton action given by

$$B(\beta) = 2\pi \sqrt{1/\lambda^*(\beta)}, \tag{6.79}$$

where $\lambda^*(\beta)$ is the radius of convergence of the weak coupling expansion (6.74). This action naturally interpolates between the action at $\beta \to 0$ in (6.70), and the action at $\beta \to \infty$ in (6.84).

### 6.2.4 The $\beta \to \infty$ limit

We now consider the $\beta \to \infty$ limit of (6.73), and compare it with the expansions in subsubsection 6.2.1. More precisely we will consider the following two limits

① We first take $m, n \to \infty$ s.t. $\lambda, \kappa$ are fixed as in subsubsection 6.2.3. Then we take $\frac{n}{m} = \beta \to \infty$.

② We first take $n \to \infty$ s.t. $\kappa, m$ are fixed as in subsubsection 6.2.1. Then we take $m \to \infty$.

Let us first investigate these two limits in the weak 't Hooft coupling region, that is $|\kappa| < k^* = 1/8$ From (6.73) we get

$$\lim_{①} \Delta \mathcal{G}_{m,n} \simeq \sum_{k \geq 1} 48 \frac{(-1)^{k+1}\Gamma\left(\frac{3}{2} + k\right)^2}{\pi \Gamma(1+k)^2} \frac{1}{2k+1} \zeta(2k+1)(8\kappa)^k. \tag{6.80}$$

On the other hand, from (6.51), in the limit $m \to \infty$, we find

$$\lim_{\substack{\text{(2)}}} \Delta\mathcal{G}_{m,n} \simeq \sum_{k \geq 1} 48 \frac{(-1)^{k+1}\Gamma\left(\frac{3}{2}+k\right)^2}{\pi\Gamma(1+k)^2}\left(\lim_{m\to\infty} p_m(k)\right)\zeta(2k+1)(8\kappa)^k. \qquad (6.81)$$

Both in (6.80) and (6.81) we used that the series converge uniformly[20] in the region $|\kappa| < k^* = 1/8$ and therefore we can switch limits and sum. We have

$$\lim_{\substack{\text{(1)}}} \Delta\mathcal{G}_{m,n} \simeq \lim_{\substack{\text{(2)}}} \Delta\mathcal{G}_{m,n}, \quad |\kappa| < \kappa^*. \qquad (6.82)$$

Let us now investigate these limits in the strong $\kappa$ coupling region. Since the large $\kappa$ expansion is divergent we can not easily switch sum and limit, and we can only approach this regime using the limit ①. From (6.80) by performing Mellin-Barnes we get

$$\lim_{\substack{\text{(1)}}} \Delta\mathcal{G}_{m,n} \simeq 6\left(2 + 2\gamma_E + \log\frac{\kappa}{2}\right) + \frac{3\sqrt{2}}{\pi\sqrt{\kappa}} + 2\sum_{k \geq 1} \frac{\Gamma\left(k+\frac{1}{2}\right)^3 \pi^{-2k-\frac{5}{2}}}{\Gamma(k+1)}\zeta(2k+1)\frac{12k}{(8\kappa)^{k+\frac{1}{2}}}. \qquad (6.83)$$

The structure in (6.83) is similar to the one at finite $m$ in (6.57), except for the term $\frac{3\sqrt{2}}{\pi\sqrt{\kappa}}$, which interestingly does not appear at any finite $m$.

The resurgent analysis of (6.83) is completely analogous to the ones in (6.57). The leading instanton action is

$$4\pi\sqrt{2} \qquad (6.84)$$

which agrees with (6.62). For the discontinuity we find

$$\text{disc}^{(\infty)}(\kappa) = \mathrm{i}e^{-4\pi\sqrt{2\kappa}}\left(24 + \sum_{n \geq 1}\frac{c_n^{(\infty)}}{\kappa^{n/2}}\right) + \mathrm{i}\sum_{\ell \geq 2}e^{-4\pi\ell\sqrt{2\kappa}}\left(24 + \sum_{n \geq 1}\frac{c_n^{(\infty)}}{\ell^n \kappa^{n/2}}\right). \qquad (6.85)$$

where the $c_n^{(\infty)}$ can be easily derived and the list is available upon request. Notice that, the sums over $n$ on the r.h.s. are divergent. Hence to make sense of the r.h.s. we think of each sum over $n$ as its median Borel summation along the positive real axis.

## 7 Outlook

In this paper we studied extremal and integrated correlators in four dimensional $SU(3)$ SCFT in the regime where we have a large number of insertions, i.e. large $R$-charge. We found that a new description emerge involving a set of two coupled matrix models: a Wishart model and a Jacobi model. The size of the matrices in these models corresponds to the

---

[20]It is enough to note that coefficients of the series in (6.51) are all smaller than $48\zeta(3)(8|\kappa|)^k$ in absolute value. Since $\sum_k 48\zeta(3)(8|\kappa|)^k$ converges for $|\kappa| < 1/8$, uniform convergence follows from the Weierstrass M-Test.

maximal number of insertions for each of the two single trace operators: $\phi_2$ and $\Phi_2(x, y)$ insertions are controlled by the Wishart model, while $\phi_3$ and $\Phi_3(x, y)$ insertions are controlled by the Jacobi model. This description gives us a new analytic handle into these regimes and the corresponding non-perturbative effects. However, several open directions remain to be investigated. Let us list some of them.

- In this work we focus on $SU(3)$ gauge theories, however it should be possible to find a systematic generalization to all $SU(N)$ gauge groups. One simply needs to find the good change of variables which generalizes (2.41). It is natural to expect that the $\phi_2$ and $\Phi_2(x, y)$ insertions will always be controlled by a Wishart model, while the other insertions should be controlled by some Jacobi-like [21] models with possible multi-cuts phases starting at $N = 4$.

- Using our matrix models, we derive the behaviour of extremal and integrated correlators in the regime where we have a large number of insertions, including some non-perturbative effects. It would be very interesting to interpret our results from the point of view of an EFT and use what we learn in this simplified model to go beyond the realm of supersymmetric gauge theories. For example in the context of the $O(N)$ model, see [49–55] for some recent developments in this context.

- In this paper, we provided analytic predictions for some non-perturbative effects in the double scaling limits where $\text{Im}\tau$ scale with the number of insertions. In some cases (see subsection 5.1, subsection 5.3, subsubsection 6.2.1, subsubsection 6.2.2), we have a full prediction for these effects. However, this is not the case in general. For instance, in the large $m, n$ regime at fixed $\beta$, we know the instanton actions but we do not know the full non-perturbative structure. Additionally, if we work at fixed $\text{Im}\tau$, we also expect non-perturbative effects that are exponentially suppressed as $\sqrt{n \text{Im}\tau}$ and $\sqrt{m \text{Im}\tau}$. It would be interesting to extract the precise form of all these effects from the matrix models perspective and provide a detailed physical framework for them.

- For $\mathcal{N} = 2$ SQCD, we have successfully derived a matrix model description for extremal correlators in the regime where the number of $\phi_3$ insertions is large. This description remains valid regardless of whether the number of $\phi_2$ insertions is large or small. However, our technique requires a large number of $\phi_3$ insertions. Despite this, we expect that a matrix model description will still emerge even when the number of $\phi_3$ insertions is small, in particular given that the scaling limit (3.17) is well-defined. This is a peculiarity of the mixing structure in $\mathcal{N} = 2$ SQCD which it would be important to understand better. More generally, it would be interesting to understand at which extent the $\mathcal{N} = 2$ mixing structure gets modified at subleading orders in the 't Hooft expansion.

---

[21] Meaning that the eigenvalues only take value in a compact set.

- For integrated correlators in $\mathcal{N} = 4$ SYM, our matrix models are expected to encode all subleading effects in $1/n$ and $1/m$. This both in the double-scaling limit, where we scale $\mathrm{Im}\tau$, and in the "pure" large charge limit, where $\mathrm{Im}\tau$ remains fixed. Analyzing the latter, deriving an all-order expression for these subleading effects and finding an EFT interpretion would be very interesting.

- A beautiful feature of integrated correlators is their connection to modularity [41, 42, 56–58]. Exploring how this relationship manifests in matrix models and investigating the potential links to the holomorphic anomaly approach in [59] would be particularly insightful.

- Extremal correlators in rank 1 theories exhibit the integrable structure of a semi-infinite Toda chain [25, 27, 35, 36]. It would be interesting to explore whether our matrix model representation could help identify a more general integrable structure, if any, underlying higher rank correlators[22].

We hope to report on some of these questions in the near future.

## Acknowledgements

We would like to thank Matteo Beccaria, Francesco Galvagno, Zohar Komargodski, Marcos Mariño, Diego Rodriguez-Gomez, Raffaele Savelli, Luigi Tizzano, Sasha Zhiboedov and in particular João Caetano and Maximilian Schwick for many useful and interesting discussions. This work is partially supported by the Swiss National Science Foundation Grant No. 185723 and the NCCR SwissMAP.

## A  Matrix models

Here we can collect some details about the matrix models which are relevant for our analysis.

### A.1  Wishart-Laguerre model

Let $W$ be an hermitian non-negative $n \times n$ matrix. Schematically, a Wishart matrix model with potential $V$ is a matrix integral of the form

$$Z = \frac{1}{\mathrm{Vol}(U(n))} \int \mathrm{d}W \, \mathrm{e}^{V(W)}. \tag{A.1}$$

Here we are interested in a particular example of such matrix model where the potential is

$$V(W) = -n\mathrm{Tr}W - (3m+3)\mathrm{Tr}\log W \ . \tag{A.2}$$

---

[22]We would like to thank Zohar Komargodski for bringing this aspect to our attention.

After gauge-fixing, we can express the matrix integral as an ordinary multidimensional integral over the eigenvalues $z_i$ of $W$. More specifically we have the following $n$ dimensional integral

$$Z_{\text{MP}}^{(m)}(n) = \frac{1}{n!} \int_{\mathbb{R}_+^n} \mathrm{d}^n z \prod_{i<j} (z_i - z_j)^2 \prod_{i=1}^n \mathrm{e}^{-nz_i} z_i^{3m+3} . \tag{A.3}$$

This model is exactly solvable and is commonly known as the Wishart-Laguerre model, see for instance [60–63] for more details and references. We have

$$Z_{\text{MP}}^{(m)}(n) = \frac{1}{n!} \left(\frac{1}{n}\right)^{n(n+3m+3)} \prod_{j=0}^{n-1} \Gamma(j+2)\Gamma(j+3m+4) \tag{A.4}$$

For the propose of this paper, it is also useful to define

$$Z^{(m)}(n) = \frac{1}{n!} \int_{\mathbb{R}_+^n} \mathrm{d}^n y \prod_{i<j} (y_i - y_j)^2 \prod_{i=1}^n \mathrm{e}^{-2\pi\text{Im}\tau y_i} y_i^{3m+3} \tag{A.5}$$

which is simply related to (A.4) by a change of variables and we have

$$Z^{(m)}(n) = \left(\frac{2\pi\text{Im}\tau}{n}\right)^{-n(n+3m+3)} Z_{\text{MP}}^{(m)}(n) . \tag{A.6}$$

In the limit

$$n, m \to \infty \quad \frac{n}{m} = \beta \quad \text{fixed.} \tag{A.7}$$

the density of eigenvalues for (A.3) follows a Marčenko-Pastur distribution [64]

$$\rho_{\text{MP}}(z) = \frac{1}{2\pi z} \sqrt{(b-z)(z-a)}, \quad z \in [a, b] \tag{A.8}$$

where

$$a = 2 + 3\beta^{-1} - 2\sqrt{1 + 3\beta^{-1}}, \qquad b = 2 + 3\beta^{-1} + 2\sqrt{1 + 3\beta^{-1}}. \tag{A.9}$$

We also define expectation values in the model (A.3) as

$$\langle f(z, \cdot) \rangle_{\text{MP}}^{(n,m)} = \frac{\frac{1}{n!} \int_{\mathbb{R}_+} \mathrm{d}^n z \prod_{i<j} (z_i - z_j)^2 \prod_{i=1}^n \mathrm{e}^{-nz_i} z_i^{3m+3} f(z_i, \cdot)}{Z_{\text{MP}}^{(m)}(k)}$$
$$= \frac{\frac{1}{n!} \int_{\mathbb{R}_+} \mathrm{d}^n y \prod_{i<j} (y_i - y_j)^2 \prod_{i=1}^n \mathrm{e}^{-2\pi\text{Im}\tau y_i} y_i^{3m+3} f\left(\frac{2\pi\text{Im}\tau}{n} y_i, \cdot\right)}{Z^{(m)}(n)} . \tag{A.10}$$

If $\log f(z_i, \cdot)$ is subleading in the large $n$ limit (A.7), we have

$$\langle f(z, \cdot) \rangle_{\text{MP}}^{(n,m)} = \exp\left(n \int_a^b \rho_{\text{MP}}(y) \log(f(y, \cdot)) + \mathcal{O}(1)\right). \tag{A.11}$$

We note the momenta in the Wishart-Laguerre model (A.3) as

$$\tau_k^{(n,m)} = \frac{\frac{1}{n!} \int_{\mathbb{R}_+} \mathrm{d}^n z \prod_{i<j} (z_i - z_j)^2 \prod_{i=1}^n \mathrm{e}^{-nz_i} z_i^{3m+3} \sum_{i=1}^n z_i^k}{Z_{\text{MP}}^{(m)}(k)} \tag{A.12}$$

These can also be computed exactly and we have (see e.g. [65] and reference therein)

$$\tau_k^{(n,m)} = 2^k \sum_{\ell=0}^{n-1} \frac{\Gamma(\ell+1)}{\Gamma(\ell+1+3m+3)} Q(k+3m+3, \ell, 3m+3) \tag{A.13}$$

where

$$Q(r,\ell,\alpha) = \sum_{i,j=0}^{\ell} c_i(\ell,\alpha) c_j(\ell,\alpha) \Gamma(1+r+i+j) , \quad c_k(\ell,\alpha) = \frac{\Gamma(\ell+\alpha+1)(-\ell)_k}{\ell! k! \Gamma(\alpha+k+1)} . \tag{A.14}$$

We have

$$\tau_0^{(n,m)} = n, \quad \tau_1^{(n,m)} = 2n(n+3m-3), \quad \tau_2^{(n,m)} = 4n(3m+n+3)(3m+2n+3) \cdots \tag{A.15}$$

The analysis of the momenta is useful to compute the loop expansion of the correlators in $\mathcal{N} = 2$ SQCD. As an example let us look at the two loops expression of the $\mathcal{N} = 2$ correlators (2.72). This can be view as a momentum in the $\mathcal{N} = 4$ matrix model. So we just need to compute the moment (A.13) in the Wishart-Laugerre ensamble. The result for the two loop correction reads

$$\langle \Theta_n^m \Theta_n^m \rangle_{\mathbb{R}^4}^{\mathcal{N}=2} \Big|_{2\text{loops}} = \langle \mathcal{O}_n^m \mathcal{O}_n^m \rangle_{\mathbb{R}^4}^{\mathcal{N}=4} \left( 1 - \frac{3\zeta(3)}{(2\pi\text{Im}\tau)^2} \left( \tau_2^{n+1,m} - \tau_2^{n,m} \right) + \frac{15\zeta(3)}{\pi^2\text{Im}\tau^2} \right) \tag{A.16}$$

By making everything explicit we get

$$\langle \Theta_n^m \Theta_n^m \rangle_{\mathbb{R}^4}^{\mathcal{N}=2} \Big|_{2\text{loops}} = \frac{36^{\sigma_m} n! \Gamma(3m+n+4)}{3^{m+1} 2^{2m+1} (2\pi\text{Im}\tau)^{3m+2n}}$$
$$\left( 1 - \frac{3\zeta(3)}{(4\pi\text{Im}\tau)^2} \left( (24n(n+4)+80) + 36m(2n+3) + 36m^2 \right) + \frac{60\zeta(3)}{4\pi^2\text{Im}\tau^2} \right) \tag{A.17}$$

where the last term comes from expanding the normalization factor $Z_{S^4}^{-1}$.

## A.2 Jacobi model

In a Jacobi matrix model we integrate over $n \times n$ Hermitian matrices $J$, which are non-negative and bounded above by the identity. More specifically we have

$$\int dJ (\det J)^{c-1} (\det(1-J))^{d-1} e^{-V(J)} \tag{A.18}$$

where in general $c = n_2 - n + 1$ and $b = n_1 - n + 1$, $n_i \in \mathbb{N}$ and $V(J)$ is a potential[23]. We refer to [60, 65–68] for more details and references on these models. For our propose it is convenient to take $n_1 = n_2 = n$ and take a potential of the form

$$-V_s(J) = \frac{1}{2} \log \text{Tr}((1-J)) + \left( -\frac{1}{2} + s \right) \log \text{Tr}(J) , \quad s \geq 0 . \tag{A.19}$$

---

[23]The $n_i$ are reminiscent of the fact that Jacobi matrices are written using two Wishart matrices of size $n_i \times n$, $i = 1, 2$

After gauge-fixing, the matrix integral becomes an ordinary multidimensional integral over the eigenvalues $x_i \in [0, 1]$ of $J$, that is

$$Z_{\mathrm{J}}^{(s)}(n) = \int_{[0,1]^n} \mathrm{d}^n x_i \prod_{i<j}(x_i - x_j)^2 \prod_{i=1}^n (1 - x_i)^{1/2} x_i^{-\frac{1}{2}+s}, \qquad s \in \mathbb{R} . \tag{A.20}$$

This model is exactly solvable (via the Selberg integral formula)

$$Z_{\mathrm{J}}^{(s)}(n) = \frac{1}{n!} \prod_{j=0}^{n-1} \frac{\Gamma\left(j + \frac{3}{2}\right)\Gamma(j + 2)\Gamma\left(j + s + \frac{1}{2}\right)}{\Gamma(j + n + 1 + s)} . \tag{A.21}$$

At large $n$, for fixed $s$, the eigenvalues distribution is completely determined by the vandermonde interaction as the potential

$$\mathrm{e}^{-V_s(x_i)} = (x_i - 1)^{1/2} x_i^{-\frac{1}{2}+s} \tag{A.22}$$

is subleading in the large $n$ limit. The corresponding eigenvalue density is

$$\sigma_{\mathrm{J}}(x) = \frac{1}{\pi\sqrt{x(1-x)}}, \quad x \in [0, 1]. \tag{A.23}$$

We define expectation values in the model (A.20) as

$$\langle f(z, \cdot) \rangle_{\mathrm{J}}^{(n)} = \frac{\frac{1}{n!} \int_{[0,1]^n} \mathrm{d}^n x_i \prod_{i<j}(x_i - x_j)^2 (x_i - 1)^{1/2} x_i^{\frac{1}{2}+s} f(z_i, \cdot)}{Z_{\mathrm{J}}^{(s)}(k)} , \tag{A.24}$$

If $\log f(z_i, \cdot)$ is subleading in the large $n$ limit, we have

$$\langle f(z, \cdot) \rangle_{\mathrm{J}}^{(n)} = \exp\left(n \int_0^1 \sigma_{\mathrm{J}}(y) \log(f(y, \cdot)) + \mathcal{O}(1)\right). \tag{A.25}$$

# B  Non-perturbative effects: numerical study

In this section, we perform a numerical study of the leading exponentially small effects appearing in (5.41). Let us define

$$G^{\mathrm{np}}(\beta, \lambda) = \frac{9}{2\pi\mathrm{i}} \int_{\mathrm{i}\mathbb{R}+\epsilon} \mathrm{d}s \frac{2^{s+2}(3^s - 1)\zeta(2s + 1)\Gamma(-s)\Gamma\left(s + \frac{3}{2}\right) {}_2F_1\left(-s - 1, s + 2; 1; -\frac{\beta}{3}\right)(\lambda)^{s+1}}{\sqrt{\pi}(s + 1)^2}$$
$$- \left(-3(2\beta + 3)\lambda\log(3) + \log\left(\frac{\beta}{3} + 1\right) + 24\log(A) + \log(\lambda) - 2 - \log(2) - \frac{\log(3)}{2}\right) \tag{B.1}$$

where the first term is the Mellin-Barnes representation of $\log \Delta G_{\beta m}^m$, see (5.40), while the second line is the perturbative part as $\lambda \to \infty$, see (5.41). Following our discussion around (5.41), we expect

$$G^{\mathrm{np}}(\beta, \lambda) = \mathcal{O}\left(\mathrm{e}^{-A_1(\beta)\sqrt{\lambda}}\right). \tag{B.2}$$

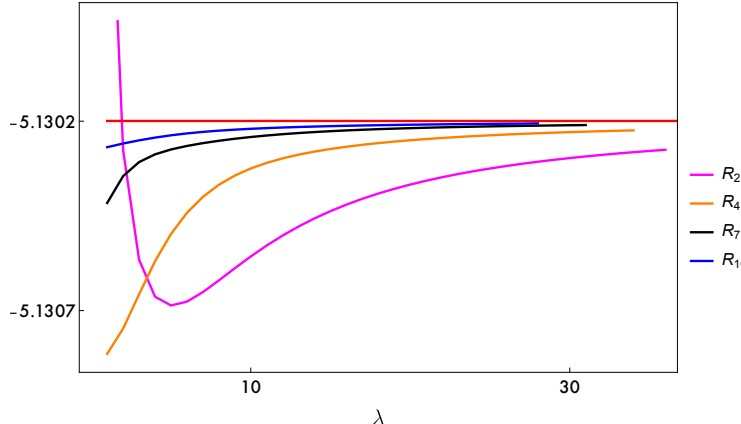

**Figure 3.** The Richardson transforms $R_N(1, \lambda)$ defined in (B.5), for $N = 2, 4, 7, 10$. The red line is $-2\pi\sqrt{6}/3$. We see a clear convergence pattern.

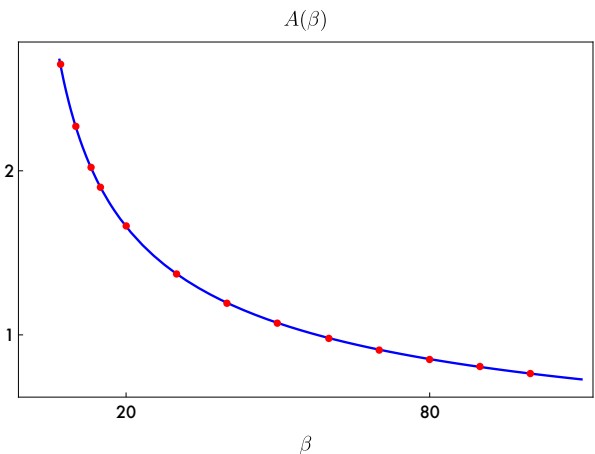

**Figure 4.** The red dots represent the action $A_1(\beta)$ computed numerically by applying the method of Richardson transforms. The blue line is the action (B.7).

We want to test (B.2) and extract the value of $A_1(\beta)$. For that it is convenient to define

$$F(\beta, \lambda) = 2\sqrt{\lambda}\left(\log G^{\mathrm{mp}}(\beta, \lambda + 1) - \log G^{\mathrm{mp}}(\beta, \lambda)\right) . \tag{B.3}$$

In this way we have

$$F(\beta, \lambda) = -A_1(\beta) + \mathcal{O}(\lambda^{-1}) . \tag{B.4}$$

We also define the $N^{\mathrm{th}}$ Ricardson transform of $F(\beta, \lambda)$ as

$$R_N(\beta, \lambda) = \sum_{k=0}^{N} \frac{(-1)^{k+N}(k+\lambda)^N F(\beta, k+\lambda)}{k!(N-k)!} = -A + \mathcal{O}(\lambda^{-1-N}). \tag{B.5}$$

The propose of $R_N$ is simply to accelerate the convergence. A graphical results for $\beta = 1$ is shown in Figure 3. We find numerically that

$$A_1(1) = 5.13020 \cdots = \frac{2\pi}{3}\sqrt{6}. \tag{B.6}$$

We repeat this procedure for several values of $\beta$, see Figure 4 and we find

$$A_1(\beta) = \frac{2\sqrt{6}\pi}{\sqrt{2\beta + 2\sqrt{\beta(\beta+3)} + 3}}. \tag{B.7}$$

## C   Resurgence at $\beta \to \infty$

### C.1   Borel summation

We are interested in studying the resurgence structure of (5.53), that is

$$F^{\mathrm{p}}(\kappa) = \sum_{n \geq 0} \kappa^{-n-\frac{1}{2}} a_n,$$

$$a_n = \frac{3 \, 2^{-3n-\frac{1}{2}}\left(3^{n+\frac{3}{2}} - 1\right)(n+1)\pi^{-2n-\frac{9}{2}}\zeta(2n+3)\Gamma\left(n+\frac{1}{2}\right)^3}{\Gamma(n+1)}. \tag{C.1}$$

Since $a_n \sim (2n)!$, we define its Borel transform as

$$B_F(\xi) = \sum_{n \geq 1} \frac{a_n}{(2n+1)!}\xi^n. \tag{C.2}$$

The new series (C.2) is now convergent but has a finite radius of convergence due to the presence of singularities. In our example these are located along the real axis at

$$\xi = \left(\sqrt{\frac{2}{3}}4\pi\right)^2. \tag{C.3}$$

see Fig. 5. In particular this means that (C.2) is not Borel summable on the real line. However we can define median Borel summation as

$$s_F(z) = \frac{1}{4z}\left(\int_{\mathbb{R}e^{i\epsilon}} e^{-\sqrt{\xi}/\sqrt{z}} B_F(\xi)\mathrm{d}\xi + \int_{\mathbb{R}e^{-i\epsilon}} e^{-\sqrt{\xi}/\sqrt{z}} B_F(\xi)\right)\mathrm{d}\xi, \quad z \in \mathbb{R}_+. \tag{C.4}$$

We performed the median summation (C.4) numerically by using the Padé-Borel method. We find that[24]

$$\frac{1}{\sqrt{\kappa}} s_F(1/\kappa) = -12 \int_0^\infty \frac{e^x}{x(e^x - 1)^2}\left(2 + J_0(x\sqrt{2\kappa})^2 - 3J_0\left(x\sqrt{\frac{2\kappa}{3}}\right)^2\right)$$

$$-24\log(A) + 6\kappa\log(3) - \log(\kappa) + 2 + \frac{\log(3)}{2} + \log(6), \tag{C.5}$$

where $A$ is the Glaisher constant.

---

[24]This can probably be derived analytically following [69, 70] and reference therein.

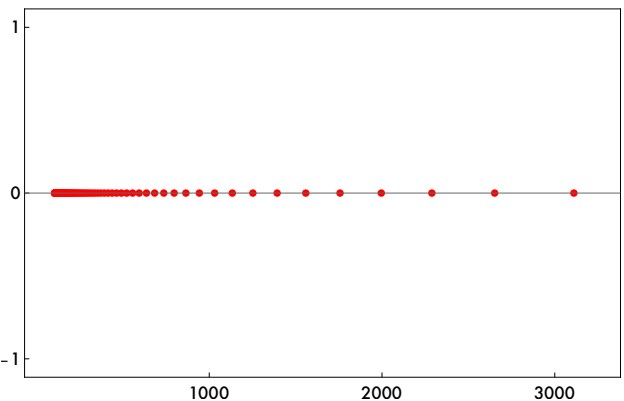

**Figure 5**. Singularities of the Borel transform (C.2) in the $\xi$-plane. These are obtained by using the Pade approximation of the function (C.2).

## C.2 Non-perturbative effects from large order behaviour

We follow [71], see also [72, 73] for a review and list of references. The large order behaviour of the coefficients $a_k$ can be easily read off from (C.1) and we have

$$
\begin{aligned}
a_k \sim &\frac{1}{\pi} \sum_{\ell \geq 1} (\ell A_1)^{-2k-1} \Gamma(2k+1) \ell^{-2} \left( c_0 + \sum_{n \geq 1} \frac{c_n (A_1)^n}{\prod_{i=1}^{n}(2k+1-i)} \right) \\
&+ \frac{1}{\pi} \sum_{\ell \geq 1} (\ell A_2)^{-2k-1} \Gamma(2k+1) \ell^{-2} \left( d_0 + \sum_{n \geq 1} \frac{d_n (A_2)^n}{\prod_{i=1}^{n}(2k+1-i)} \right)
\end{aligned}
\tag{C.6}
$$

where

$$
A_1 = \sqrt{\frac{2}{3}} 4\pi, \quad A_2 = 4\pi\sqrt{2}
\tag{C.7}
$$

are the instantons actions and $c_n, d_n$ are determined by the following equations

$$
\begin{aligned}
\frac{72(k+1)\Gamma\left(k+\frac{1}{2}\right)^2}{\pi^2 \Gamma(k+1)^2} &= \sum_{n \geq 0} \frac{\left(\sqrt{\frac{2}{3}}4\pi\right)^n c_n \Gamma(2k-n+1)}{\Gamma(2k+1)} \\
-\frac{24(k+1)\Gamma\left(k+\frac{1}{2}\right)^2}{\pi^2 \Gamma(k+1)^2} &= \sum_{n \geq 0} \frac{\left(\sqrt{2}4\pi\right)^n d_n \Gamma(2k-n+1)}{\Gamma(2k+1)} \ .
\end{aligned}
\tag{C.8}
$$

This implies that $d_n = -\frac{c_n}{3^{1+n/2}}$ and for the first few $c_n$ terms we have

$$
c_0 = \frac{72}{\pi^2}, \quad c_1 = \frac{27\sqrt{\frac{3}{2}}}{\pi^3}, \quad c_2 = -\frac{189}{32\pi^4}, \quad c_3 = \frac{513\sqrt{\frac{3}{2}}}{256\pi^5}, \quad c_4 = -\frac{26973}{16384\pi^6}, \quad \dots
\tag{C.9}
$$

These coefficients are also factorially divergent: $c_n \sim n!$. From (C.6) we can also read off the discontinuity of the Borel summation as we cross the real line. To see this we first write

$$
a_k = \frac{1}{2\pi\mathrm{i}} \oint \frac{F^{\mathrm{p}}(z^{-1})}{\sqrt{z}z^{k+1}} \mathrm{d}z \ .
\tag{C.10}
$$

Let us assume analyticity of $s_F(1/\sqrt{z})$ in the $z$ plane except on the real axis starting at the point where we have the poles of the Borel transform. We also assume dacay at infinity. Then we have

$$a_k = \frac{1}{\pi \mathrm{i}} \int_0^\infty \frac{\mathrm{disc}(1/z)}{z^{k+1}} \tag{C.11}$$

where $\mathrm{disc}(\kappa)$ is the discontinuity of the Borel summation across the cut on the real axis. From (C.11) and (C.6) it follows that

$$\begin{aligned}
\mathrm{disc}(\kappa) =& \frac{\mathrm{i}}{2} \mathrm{e}^{-A_1\sqrt{\kappa}} \sum_{n\geq 0} \frac{c_n}{\kappa^{n/2}} + \frac{\mathrm{i}}{2} \mathrm{e}^{-A_2\sqrt{\kappa}} \sum_{n\geq 0} \frac{d_n}{\kappa^{n/2}} \\
&+ \frac{\mathrm{i}}{2} \sum_{\ell\geq 2} \mathrm{e}^{-A_1\ell\sqrt{\kappa}} \sum_{n\geq 0} \frac{c_n}{\ell^{2+n}\kappa^{n/2}} + \frac{\mathrm{i}}{2} \sum_{\ell\geq 2} \mathrm{e}^{-A_2\ell\sqrt{\kappa}} \sum_{n\geq 0} \frac{d_n}{\ell^{2+n}\kappa^{n/2}}
\end{aligned} \tag{C.12}$$

where, to make sense of the divergent sums over $n$ on the right-hand side, we interpret each sum over $n$ using its median Borel summation along the positive real axis.

# D  Summary of conventions for the operators

Here, we provide a list of the various operators appearing in this paper. The distinctions between these operators are based on:

- The scalar product used in the Gram-Schmidt (GS) orthogonalization procedure.

- Whether we orthogonalize with respect to operators of the same dimension only, or if we also include operators of lower dimensions.

The list of operators we used in the text is the following.

1. Elementary Coulomb branch operators in a rank 2, four dimensional $\mathcal{N} = 2$ theory are denoted by
$$\phi_k^m = \left(\mathrm{Tr}\, \varphi^k\right)^m, \quad k = 2, 3, \quad m \geq 0. \tag{D.1}$$

   where $\varphi$ is the complex scalar in the $\mathcal{N} = 2$ vector multiplet. To compute the correlation functions of $(\phi_k)^m$ in the $\mathcal{N} = 2$ theory on $\mathbb{R}_4$, we need to go to $S^4$ and compute the correlation functions of a different class of operators denoted by

$$\left(\phi_k^m\right)' \quad k = 2, 3, \quad m \geq 0. \tag{D.2}$$

   These are defined starting from $\phi_k^m$ and by doing GS orthogonalization on the sphere, where we orthogonalize w.r.t. operators of lower dimension. The scalar product used in this GS procedure is the two-point function of $\mathcal{N} = 2$ SQCD on the $S^4$. An example is given in (2.67). Then we have

$$\langle \phi_n^m \overline{\phi_n^m} \rangle_{\mathbb{R}^4}^{\mathcal{N}=2} = \langle (\phi_n^m)' \overline{(\phi_n^m)'} \rangle_{S^4}^{\mathcal{N}=2}. \tag{D.3}$$

2. When working with $\mathcal{N} = 4\ SU(3)$ SYM is convenient to use another basis of operators, denoted $\mathcal{O}_n^m$, where

$$\mathcal{O}_n^m = \phi_2^n \mathcal{O}_0^m \quad m, n \geq 0 \ . \tag{D.4}$$

The $\mathcal{O}_0^m$ operators are constructed starting from $\phi_3^m$ by performing GS orthogonalization on the sphere, where we ortogonalize w.r.t. operators of the same dimension. In this construction the scalar product used in the GS procedure is the two point function in the $\mathcal{N} = 4$ theory on the sphere. See (2.51), some examples are given in (2.52). To compute the correlation functions of $\mathcal{O}_n^m$ on $\mathbb{R}_4$, we need to go on $S^4$ and compute the correlation functions of a different class of operators, denoted by

$$(\mathcal{O}_n^m)' \quad m, n \geq 0 \ , \tag{D.5}$$

with the convention that $(\mathcal{O}_0^m)' = \mathcal{O}_0^m$. The $(\mathcal{O}_n^m)'$ operators are constructed starting from $\mathcal{O}_n^m$ and by doing GS on the sphere, where we orthogonalize w.r.t. operators of the lower dimension but which have the same index $m$. Also in this case the scalar product used in the GS procedure is the two point function in the $\mathcal{N} = 4$ theory on the sphere. See (2.38). Then we have

$$\langle \mathcal{O}_n^m \overline{\mathcal{O}_n^m} \rangle_{\mathbb{R}^4}^{\mathcal{N}=4} = \langle (\mathcal{O}_n^m)' \overline{(\mathcal{O}_n^m)'} \rangle_{S^4}^{\mathcal{N}=4}. \tag{D.6}$$

Note that we can compute correlation functions of $\mathcal{O}_n^m$ in $\mathcal{N} = 2$ SQCD as well. However in this case (D.6) does not hold. We also note

$$(G_n^m)^{\mathcal{N}=4} = \langle \mathcal{O}_n^m \overline{\mathcal{O}_n^m} \rangle_{\mathbb{R}^4}^{\mathcal{N}=4} \ . \tag{D.7}$$

3. When working with $\mathcal{N} = 2\ SU(3)$ SQCD is convenient to use another basis of operators denoted by

$$\Theta_n^m \quad m, n \geq 0 \ . \tag{D.8}$$

The operator (D.8) is constructed starting from $\phi_2^n \phi_3^m$ and doing GS orthogonalization on $\mathbb{R}^4$ w.r.t. all operators of the form $\phi_2^i \phi_3^j$ such that $2i + 3j = 2n + 3m$ with $j < m$. In this construction the scalar product used in the GS procedure is the two point function in $\mathcal{N} = 2$ SQCD on $\mathbb{R}^4$. Some examples are given in (2.70). To compute the correlation functions of $\Theta_n^m$ on $\mathbb{R}^4$, we need to go on $S^4$ and compute the correlation functions of a different class of operators, denoted by

$$(\Theta_m^n)' \quad m, n \geq 0 \ . \tag{D.9}$$

These are constructed starting from $\phi_2^n \phi_3^m$ and then applying GS orthogonalization on $S^4$ w.r.t. all operators of the form $\phi_2^i \phi_3^j$ such that $2i + 3j \leq 2n + 3m$ with $j < m$. In this case the scalar product used in the GS procedure is the two point function of $\mathcal{N} = 2$ SQCD on $S^4$. Then we have

$$\langle \Theta_n^m \overline{\Theta_n^m} \rangle_{\mathbb{R}^4}^{\mathcal{N}=2} = \langle (\Theta_n^m)' \overline{(\Theta_n^m)'} \rangle_{S^4}^{\mathcal{N}=2}. \tag{D.10}$$

We also note

$$(G_n^m)^{\mathcal{N}=2} = \langle \Theta_n^m \overline{\Theta_n^m} \rangle_{\mathbb{R}^4}^{\mathcal{N}=4} \ . \tag{D.11}$$

4. To make contact with the matrix models in $\mathcal{N} = 2$ $SU(3)$ SQCD it is useful to introduce the operators

$$\widetilde{\Theta}_n^m \quad m, n \geq 0 . \tag{D.12}$$

This operator is defined starting from $\phi_2^n \phi_3^m$ and then applying GS orthogonalization on $S^4$ with respect to all operators of the form $\phi_2^i \phi_3^j$ such that $2i + 3j = 2n + 3m$ with $j < m$. In this case the scalar product used in the GS procedure is the two point function in the $\mathcal{N} = 2$ theory on $S^4$. These operators are similar to $(\Theta_n^m)'$, but now we only perform GS orthogonalization with operators of the same dimension, whereas in $(\Theta_n^m)'$ we also include operators of lower dimensions in the orthogonalization process. An important point is that in the 't Hooft limits (3.15), (3.16), (3.17) we have

$$\langle \widetilde{\Theta}_n^m \overline{\widetilde{\Theta}_n^m} \rangle_{S^4}^{\mathcal{N}=2} \simeq \langle (\Theta_n^m)' \overline{(\Theta_n^m)'} \rangle_{S^4}^{\mathcal{N}=2}, \tag{D.13}$$

as discussed in section 3.

5. When considering integrated correlators in $\mathcal{N} = 4$ SYM we also need to introduce the operators

$$\widetilde{\mathcal{O}}_n^m \quad m, n \geq 0 . \tag{D.14}$$

The operator $\widetilde{\mathcal{O}}_n^m$ is defined starting from $\phi_2^n \phi_3^m$ and then applying GS orthogonalization on $S^4$ with respect to all operators of the form $\phi_2^i \phi_3^j$ such that $2i + 3j = 2n + 3m$ with $j < m$. In this case the scalar product used in the GS procedure is the two point function in the $\mathcal{N} = 2^*$ theory on $S^4$. These are analogous to the operators $\widetilde{\Theta}_n^m$ but the scalar product now is taken w.r.t. the $\mathcal{N} = 2^*$. To compute the correlation functions of $\widetilde{\mathcal{O}}_n^m$ in $\mathcal{N} = 4$ on $\mathbb{R}_4$, we need to go on $S^4$ and compute the correlation functions of a different class of operators, denoted by

$$\left( \widetilde{\mathcal{O}}_n^m \right)', \tag{D.15}$$

with the convention that $\left( \widetilde{\mathcal{O}}_0^m \right)' = \mathcal{O}_0^m$. The $\left( \widetilde{\mathcal{O}}_n^m \right)'$ operators are constructed starting from $\widetilde{O}_n^m$ and by doing GS on the sphere, where we orthogonalize w.r.t. operators of the lower dimension but which have the same index $m$. Also in this case the scalar product used in the GS procedure is the two point function in the $\mathcal{N} = 2^*$ theory on the sphere, see (6.29).

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
