# Peer review of "Matrix models for extremal and integrated correlators of higher rank"

_SciPost Physics_

## Round 1 · Referee Report · Anonymous (Referee 1) · 2025-11-14

Strengths

  • detailed computations
  • cases that have not appeared in literature -new computational methods

Report

this paper deals with a problem which has received much attention in literature lately: the study of a class of field theories which do not admit a lagrangian. many new results are obtained which will be of interest for the research in this area

Requested changes

i suggest the authors to run a spell check since i found few typos

Recommendation

Publish (easily meets expectations and criteria for this Journal; among top 50%)

---

## Round 1 · Referee Report · Anonymous (Referee 2) · 2025-11-24

Strengths

The paper addresses timely and interesting questions concerning large-charge behaviour of correlators in supersymmetric theories with higher rank gauge group. (further details are in the report)

Weaknesses

Most issues are relatively minor. (see the report)

Report

This paper investigates the large-charge behaviour of extremal correlators in SU(3) SQCD and integrated correlators in SU(3) N=4 SYM. Unlike the rank-1 case, starting from SU(3) there exist many distinct operators with the same charges (equivalently, dimensions). Consequently, there are multiple different ways to take the large-charge limit, leading to a much richer dynamical structure than in rank-1 theories. The physical observables studied here are of important interest, and related topics have been an active research area.

These extremal correlators and integrated correlators can be computed exactly via supersymmetric localization. The authors analyse the resulting matrix models and show that, in a large-charge ’t Hooft limit, the results can be described in terms of certain dual effective matrix models. These dual matrix models allow one to efficiently extract various large-charge asymptotics and to explore the associated physical properties.

The paper is clearly written, and the results are new, interesting; and in some cases it helps clarify previous statements in the literature. I believe the paper is suitable for publication in SciPost.

Before publication, the authors may wish to consider the following points:

  1. Some proofreading may be helpful.

For example, in (2.61), I believe $Z_{\rm inst}$ should depend on $\tau_A$, due to the insertion of higher-dimensional operators. This of course does not affect any of the results, since instanton contribution was not considered in the paper.

In (5.49), it seems $dx$ is missing. In the paragraph after (5.53), the poles at $s=0, 1$ should be $s=0, -1$, and similarly $s= {2n+1 \over 2}$ should be $s=- {2n+1 \over 2}$.

Before (6.80), "... $|\kappa|<k^* = 1/8$ From (6.73) ... " should be "... $|\kappa|<k^* = 1/8$. From (6.73) ... ", namely a 'period' is missing.

A typo before (C.11), where "dacay" should be "decay", and $dz$ is missing in (C.11).

  1. Is the "$n$" that appears in the subscript of $\mathcal{O}_{n+3 \ell}^{(m-2\ell)}$ in (3.7) a typo? Should it be removed?

  2. The notations ${\rm disc}(1/z)$ in (C.11) and ${\rm disc}(\kappa)$ in (C.12) are slightly confusing. As far as I can tell, they are not describing the discontinuities of $1/z$ and $\kappa$.

  3. Finally, It’s worth mentioning that some of the large-charge properties of certain integrated correlators in N=4 SYM with SU(N) gauge group have been considered in the literature. In particular, the operators constructed in (2.51) (2.52) have also appeared and studied in 2407.02250 (namely ref. [45] of the paper).

Recommendation

Publish (easily meets expectations and criteria for this Journal; among top 50%)

---

## Round 1 · Referee Report · Anonymous (Referee 3) · 2025-12-12

Strengths

This paper addresses the important challenge of studying strongly coupled models.
It further extends the successful approach of working at large R-charge.
It uses a novel method, namely the one of expressing extremal correlators through matrix models.

Weaknesses

There are no scientific weaknesses. I would however recommend to go through the paper with a spellchecker and a grammar checker to address some typos and minor grammatical inconsistencies.

Report

The paper extends some interesting results in N=2 SQCD in the limit of large R-charge to higher rank theories, focusing on the SU(3) case by making use of the fact that extremal correlators can be expressed through Wishart matrix models where the size of the matrix corresponds to the R-charge.
This apporach leads to a number of interesting follow-up questions that can be addressed.
The paper uses novel methods and addresses interesting and timely questions. The results are carefully derived and clearly laid out. I recommend it for publication

Recommendation

Publish (easily meets expectations and criteria for this Journal; among top 50%)

---

## Round 1 · Referee Report · Anonymous (Referee 4) · 2025-12-15

Disclosure of Generative AI use

The referee discloses that the following generative AI tools have been used in the preparation of this report:

polishing of english grammar

Strengths

1) The paper addresses an interesting open problem, namely the construction of a matrix-model description of chiral ring correlators in the large-charge double-scaling limit for rank-2 theories. 2) The explicit results obtained for chiral ring operators are likely to be useful for future investigations using complementary or alternative techniques. 3) Obtains similarly interesting results for integrated correlators.

Weaknesses

1) While the technical analysis is thorough and elegant, the manuscript would benefit from additional discussion of the physical interpretation and significance of the results obtained.

Report

Recent literature has observed that chiral ring correlators in four-dimensional N=2 SCFTs admit interesting simplifications in a suitable large-charge double-scaling limit. Closely related limits were later identified also in non-supersymmetric settings. These developments led to the discovery of a matrix-model description of large-charge correlators in rank-1 SCFTs in ref. 17 by one of the authors. The present paper makes an important step forward by generalizing this framework to rank-2 theories, where progress had so far been hindered by significant technical challenges, most notably operator mixing. This is an important result and merits publication in SciPost. I only have a few minor comments, which the authors may wish to address in order to further improve the clarity of the manuscript.

Requested changes

1) I was unable to reproduce eq. (4.17) starting from eq. (4.15), and I assume this is due to a typo. Rewriting eq. (4.15) in terms of the total charge r=2n+3m, I instead find a structure of the form

exp[r log(r)+ g(beta) r+4log r +...]

This structure is remarkably similar to the rank-1 case. Although I understand that a detailed effective-field-theory analysis would likely deserve a separate work, it seems to me that this similarity warrants at least a brief comment. In particular, the r og r term is identical to the rank-1 case and likely follows from the simplicity of the large-charge EFT, which should effectively reduce to free fields on the Coulomb branch. Moreover, the coefficient of the log r term matches \alpha+1 where \alpha=3 in the notation of ref. 15 is the a-anomaly difference between the full theory and the generic point on the Coulomb branch. It might be worth commenting on this aspect.

2) Relatedly, does a similar structure persist in SQCD? In particular, some comments on the behavior of the r log r and log r terms in that case might be beneficial for readers.

3) It would be useful to further justify the conjectured form of the instanton actions given in eq. (5.45). Has this conjecture been explicitly verified numerically? Does it follow from expectations about the BPS spectrum on the Coulomb branch?

4) Some typos: - below eq. 4.9 "correlatos"-->"correlators" - below 5.40 "contanct"-->"contact"

Recommendation

Publish (surpasses expectations and criteria for this Journal; among top 10%)

---

## Editorial Decision

in_voting